# scSemiProfiler: Advancing large-scale single-cell studies through semi-profiling with deep generative models and active learning

Jingtao Wang[1,2], Gregory J. Fonseca [1,2,3] & Jun Ding [1,2,3,4,5] ✉

Single-cell sequencing is a crucial tool for dissecting the cellular intricacies of complex diseases. Its prohibitive cost, however, hampers its application in expansive biomedical studies. Traditional cellular deconvolution approaches can infer cell type proportions from more affordable bulk sequencing data, yet they fall short in providing the detailed resolution required for single-cell-level analyses. To overcome this challenge, we introduce "scSemiProfiler", an innovative computational framework that marries deep generative models with active learning strategies. This method adeptly infers single-cell profiles across large cohorts by fusing bulk sequencing data with targeted single-cell sequencing from a few rigorously chosen representatives. Extensive validation across heterogeneous datasets verifies the precision of our semi-profiling approach, aligning closely with true single-cell profiling data and empowering refined cellular analyses. Originally developed for extensive disease cohorts, "scSemiProfiler" is adaptable for broad applications. It provides a scalable, cost-effective solution for single-cell profiling, facilitating in-depth cellular investigation in various biological domains.

The advent of single-cell sequencing has dramatically reshaped the landscape of biological research, providing an unparalleled view into the cellular complexities of organisms[1,2]. This technique has unearthed the subtle distinctions among individual cells, enabling a richer understanding of cellular dynamics[3,4]. High-resolution data from such analyses are essential for delineating and characterizing the myriad of cellular subpopulations within patient samples, leading to transformative developments in biomarker discovery and personalized therapy strategies[5–7]. Cohort studies, which offer longitudinal insights by observing specific groups over time, are particularly poised to benefit from these advances[8–10]. However, the substantial financial cost associated with single-cell sequencing—such as the estimated $6000 required to sequence 20,000 cells as of 2023 (costpercell)—can often be a limiting factor for extensive research endeavors.

A range of computational strategies, particularly deconvolution methods, are available for dissecting bulk data[11] into distinct cell populations. These approaches enable a harmonious balance between affordability and data resolution. Prominent among these are deconvolution techniques such as CIBERSORTx[12,13], Bisque[14], DWLS[15], MuSiC[16], and NNLS[16], which have become particularly popular. Another method, EPIC[17], operates similarly but necessitates a pre-defined signature gene matrix. In addition to these classical deconvolution approaches, Scaden[18] and TAPE[19] present an emerging category of methods that employ more sophisticated deep neural networks to estimate cell type proportions using single-cell reference data and deconvolute bulk gene expression. These methods estimate the proportions of different cell types within bulk sequencing samples by utilizing signature profiles from single-cell reference datasets. While valuable for enhancing cohort study analyses, such

[1]Meakins-Christie Laboratories, Research Institute of McGill University Health Centre, 1001 Decarie Blvd, Montreal H4A 3J1 Quebec, Canada. [2]Department of Medicine, Division of Experimental Medicine, McGill University, 1001 Decarie Blvd, Montreal H4A 3J1 Quebec, Canada. [3]Quantitative Life Sciences, McGill University, 845 Rue Sherbrooke Ouest, Montreal H3A 0G4 Quebec, Canada. [4]School of Computer Science, McGill University, 3480 Rue University, Montreal H3A 2A7 Quebec, Canada. [5]Mila-Quebec AI Institute, 6666 Rue Saint-Urbain, Montreal H2S 3H1 Quebec, Canada. ✉e-mail: jun.ding@mcgill.ca

conventional bulk decomposition methods exhibit limitations in their resolution and accuracy[20,21]. Capable of dissecting bulk samples into different cell types and ascertaining their gene expression patterns, they offer beneficial insights at the cell type level. Yet, these methods often fail to deliver high deconvolution accuracy. The aforementioned conventional cell deconvolution methods often do not achieve high accuracy and perform substantially worse particularly when dealing with real bulk data. More importantly, conventional methods are constrained to cell type level resolution and are incapable of achieving the true single-cell resolution required to capture the significant variability within cell types. While cell type proportions offer a view of changes in cell fractions within bulk samples, and cell type signature matrices provide gene expression profiles at the cell type level, neither can delve into the heterogeneity within individual cell types. This single-cell level granularity is critical for deciphering the complexities of diseases and their response to therapies[22-26]. The limitations of traditional bulk decomposition methods restrict our ability to fully explore cellular dynamics and to appreciate the unique contributions of each cell to the overall sample profile. True single-cell resolution allows for in-depth analyses essential for understanding cellular heterogeneity, such as UMAP[27], pathway activation pattern analysis, biomarker discovery, gene functional enrichment, as well as cell-cell interaction[28] and pseudo-time trajectory analysis[29-32]. These analyses are further enhanced when combined with advanced machine learning techniques[33-37], which are particularly adept at decoding the subtleties of cellular heterogeneity and dynamics and typically necessitate single-cell datasets.

In response to the challenges previously highlighted and with the goal of offering a cost-effective approach for extensive single-cell sequencing, we introduce the single-cell Semi-profiler (*scSemiProfiler*). This deep generative computational tool is crafted to significantly improve the precision and depth of single-cell analysis. It stands out as a more economical and scalable option for single-cell sequencing, thereby facilitating advanced single-cell analysis with greater accessibility. This tool effectively integrates active learning techniques[38] with deep generative neural network algorithms[39], aiming to provide single-cell resolution data at a more affordable price. *scSemiProfiler* aims to simultaneously achieve two fundamental goals in the semi-profiling process. On one hand, *scSemiProfiler*'s active learning module integrates information from the deep learning model and bulk data, intelligently selecting the most informative samples for actual single-cell sequencing. On the other hand, *scSemiProfiler*'s deep generative model component[40-43] effectively merges single-cell data from representative samples with the bulk sequencing data of the cohort, computationally inferring the single-cell data for the remaining non-representative samples. This deep neural network approach leads to more detailed "deconvolution" of the target bulk data into precise single-cell level measurements. Consequently, with only the budget for bulk sequencing and single-cell sequencing of representatives, *scSemiProfiler* outputs single-cell data for all samples in the study. To the best of our knowledge, *scSemiProfiler* is the first of its kind designed for such intricate single-cell level computational decomposition from bulk sequencing data.

Through comprehensive evaluations across a variety of datasets, *scSemiProfiler* has consistently produced semi-profiled single-cell data that closely align with actual single-cell datasets and accurately reflects the results of downstream tasks. As a result, *scSemiProfiler* facilitates improved access to single-cell data for large-scale studies, including disease cohort investigations and beyond. By making large-scale single-cell studies more affordable, *scSemiProfiler* promises to catalyze the application of single-cell technologies in a wide array of expansive biomedical research. This advancement is set to expand the scope and enhance the depth of biological research globally.

## Results
### Method overview

The *scSemiProfiler* approach represents a deep neural network framework for decomposing broad-scope bulk sequencing data into detailed single-cell cohorts. It achieves this by conducting single-cell profiling on only a select few representative samples and then computationally inferring single-cell data for the remainder. This approach substantially lowers the costs associated with large-scale single-cell studies. As depicted in Fig. 1, our method is designed to deliver cost-effective, semi-profiled single-cell sequencing data, enabling deep exploration of cellular dynamics in large cohorts. In this context, "semi-profiling" refers to the generation of single-cell data for an entire cohort, achieved either through direct single-cell sequencing of selected representative samples or via in silico inference using a deep generative model. This in silico inference process combines actual single-cell sequencing data from a representative sample with bulk sequencing data, encompassing both the target and the representative sample. Thus, a "semi-profiled cohort" includes real single-cell data for representative samples and inferred data for the non-representative ones. This deep generative approach facilitates a thorough examination of individual cellular profiles within a larger dataset, seamlessly linking the extensive scope of bulk sequencing with the granularity of single-cell analysis.

Initially, the semi-profiling pipeline commences with bulk sequencing of each cohort member (Fig. 1a), laying the foundational data layer for all subsequent analyses. Following this foundational step, the methodology employs a clustering analysis to form $B$ (sample batch size) sample clusters utilizing the extensive data derived from the initial bulk sequencing and selecting a "representative" sample for each cluster (Fig. 1b). Single-cell profiling will be conducted on the selected representative samples in preparation for the following steps. The core of the *scSemiProfiler* involves a deep generative learning model (Fig. 1c). This model is engineered to intricately meld actual single-cell data profiles with the gathered bulk sequencing data, thereby capturing complex biological patterns and nuances. Specifically, it uses a VAE-GAN[44] architecture initially pretrained on single-cell sequencing data of selected representatives for self-reconstruction. Subsequently, the VAE-GAN is further pretrained with a representative reconstruction bulk loss, aligning pseudobulk estimations from the reconstructed single-cell data with real pseudobulk. Finally, the model undergoes fine-tuning with another target bulk loss tied to the real bulk sequencing data of the target samples, facilitating precise in silico inference of the targets' single-cell profiles. Once the in silico single-cell inference is finished for all non-representative samples in the cohort, an active learning module can be used for selecting the next batch of potentially most informative representatives for single-cell sequencing to further improve the semi-profiling performance (Fig. 1d). When studying a smaller dataset, or when more cells per sample are required, a smaller batch size, such as 2, may be preferred. However, a batch size of 4 is set as default to maximize the usage of a 10x genomics single-cell toolkit, which can typically capture up to 20,000 cells (4 samples if assuming 5000 cells each). This dynamic, iterative process is continuously augmented with newly acquired single-cell data, ensuring that the most informative samples are selected for real single-cell profiling, leading to more accurate in silico single-cell inference for non-representative samples. This iterative process concludes when the budgetary constraint is met or when a sufficient number of representatives have been chosen to ensure satisfactory semi-profiling performance.

When any of the stop criteria are met, the semi-profiled single-cell data can be used for a broad spectrum of downstream single-cell analyses (Fig. 1e), such as cell feature visualization, biomarker and function enrichment analysis, tracking cell type compositions in various tissues/conditions, cell-cell interaction analysis, and pseudotime analysis. Ultimately, *scSemiProfiler* offers a holistic sequencing

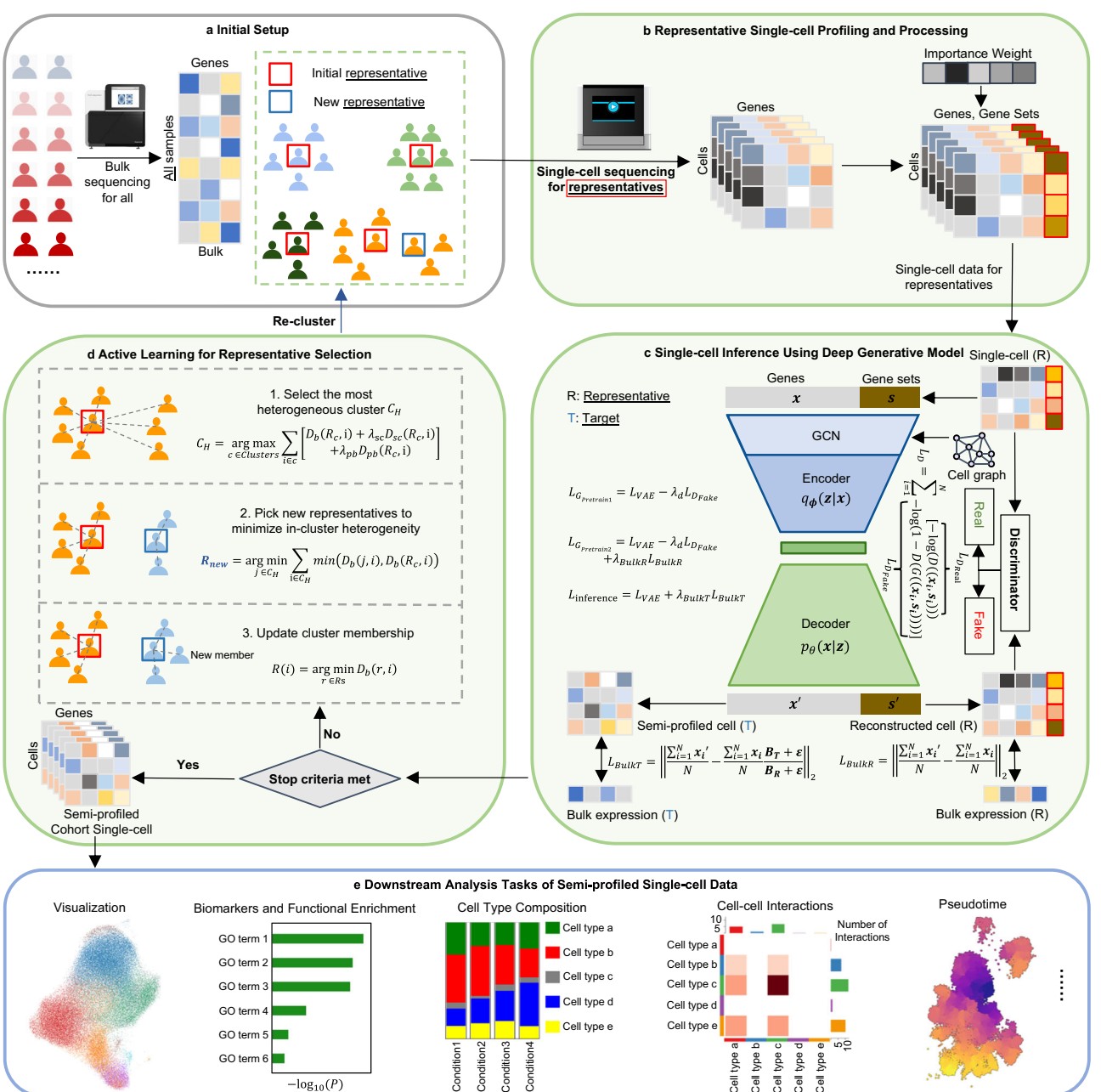

**Fig. 1 | Overview of the *scSemiProfiler* method. a** Initial Configuration: Bulk sequencing is first conducted on the entire cohort, followed by clustering analysis of this data. This analysis identifies representative samples, typically those closest to the cluster centroids. **b** Representative Profiling: The identified representatives are then subjected to single-cell sequencing. The data obtained from this sequencing is further processed to determine gene set scores and feature importance weights, enriching the subsequent analysis steps. **c** Deep Generative Inference: Utilizing a VAE-GAN-based model, the process integrates comprehensive bulk data from the cohort with the single-cell data derived from the representatives, and infers the single-cell data for the non-representative (target) samples. During the model's 3-stage training, the generator aims to optimize losses $L_{G_{Pretrain1}}$, $L_{G_{Pretrain2}}$, and $L_{inference}$, respectively, whereas the discriminator focuses on minimizing the discriminator loss $L_D$. In $L_D$, $G$ and $D$ are the generator and discriminator respectively. The term $D((\mathbf{x_i}, \mathbf{s_i}))$ represents the discriminator's predicted probability that

a given input cell is real, under the condition that it is indeed a real cell. Conversely, $D(G((\mathbf{x_i}, \mathbf{s_i})))$ denotes the discriminator's predicted probability that the input cell is real, when in fact it is a cell reconstructed by the generator. **d** Representative Selection Decision: Decisions on further representative selection are made, taking into account budget constraints and the effectiveness of the current representatives. An active learning algorithm, which draws on insights from the bulk data and the generative models, is employed to pinpoint additional optimal representatives. These newly selected representatives then undergo further single-cell sequencing (**b**) and serve as new reference points for the ongoing in silico inference process (**c**). This active learning step is optional if the user prefers the all-in-one "global mode". **e** Comprehensive Downstream Analyses: The final panel shows extensive downstream analyses enabled by the semi-profiled single-cell data. This is pivotal in demonstrating the model's capacity to provide deep and wide-ranging insights, showcasing the full potential and applicability of the semi-profiled single-cell data.

perspective, delivering nuanced single-cell insights from bulk sequencing data. The method exhibits acceptable performance in terms of runtime and memory usage (refer to Supplementary Fig. S1, making it suitable for scaling to large-scale datasets.

While active learning allows for selecting a minimal number of representatives to conserve budget, the necessity for multiple sequencing rounds can sometimes challenge many laboratories and introduce batch effects. To counter this, *scSemiProfiler* features a

"global mode" that enables all-in-one selection of representatives based on initial bulk data analysis. Furthermore, this "global mode" can be used in conjunction with active learning to enhance the selection process.

## The semi-profiled COVID-19 single-cell cohort exhibits significant similarity to its real counterpart

To test the performance of *scSemiProfiler* in generating semi-profiled single-cell data that resonates with the granularity and details of actual single-cell sequencing, we utilized a COVID-19 cohort single-cell sequencing dataset[45]. After quality controls (please refer to the section titled "Representative single-cell profiling and processing" in our Methods for more details.), this dataset includes 124 samples, including healthy controls and infected patients of different severity levels: asymptomatic, mild, moderate, severe, or critical. Here, we produced pseudobulk data by taking the average of the normalized count single-cell data. We then tested *scSemiProfiler*'s ability to regenerate the single-cell cohort from the pseudobulk data and real-profiled single-cell representatives using semi-profiling, as the actual bulk data for those samples in the COVID-19 cohort is absent. In generating our semi-profiled dataset, 28 representatives were selected in batches of 4 using our active learning algorithm for real single-cell profiling. Deep generative models then inferred the single-cell profiles for the remaining samples based on the representatives' real-profiled single-cell data and the bulk data of all samples. The estimated total cost of these bulk and single-cell sequencing is $62,640. This is only 33.7% of the estimated price, $186,000, for actually conducting single-cell sequencing for the entire cohort. Additionally, our approach offers the advantage of generating extra bulk data for the cohort, a benefit not provided by single-cell sequencing of the entire cohort. The price of bulk sequencing is estimated based on the cost at McGill Genome Centre in the year 2023. This estimation assumes the use of one NovaSeq 6000 S2 system (capable of sequencing up to 4 billion reads per run) at an approximate cost of $7000, plus an additional $110 for library preparation per bulk sample. The cost for single-cell sequencing is based on the tool (costpercell). This tool provides a cost estimate for capturing 0.8 billion total reads from 20,000 cells across four samples in one 10x lane, equating to $0.3 per cell. Consequently, the estimated cost for each sample (5000 cells) is $1500. In the following sections, we provide justification for *scSemiProfiler*'s capacity to match the analytical results of a fully profiled single-cell cohort. We compare the outcomes from the semi-profiled cohort, which includes real single-cell data for representative samples and inferred single-cell data for target samples, to those from the real-profiled cohort, the ground truth with single-cell data for all samples.

Using UMAP[27] visualizations, we show a significant alignment between the semi-profiled and the real-profiled in Fig. 2a and b. We also annotated the semi-profiled COVID-19 cohort using an unbiased approach. We trained a Multi-layer Perceptron (MLP) classifier using the annotated representatives' cells and used it to predict the cell types of the rest of the non-representative samples' cells generated by the deep learning model. As shown in Fig. 2b, cell clustering remains intact in the semi-profiled cohort, from the distinctive clusters of B cells, plasmablasts, and platelets to the nuanced similarities of the CD14 and CD16 cells. The semi-profiled dataset illustrates its high fidelity to the real-profiled version. This fidelity extends to capturing even the subtle batch effects, as observed in the twin CD4 clusters, further accentuating *scSemiProfiler*'s robust in silico inference capabilities. The UMAP plots in Fig. 2c weave together directly sequenced samples with the semi-profiled ones, showcasing the tool's finesse. The overlapping data points found in both sequencing techniques resonate with the transformative nature of *scSemiProfiler*—it harmonizes accuracy and cost-efficiency seamlessly. This comparison validates our tool's ability to approximate a fully profiled dataset. This prompts an inquiry: Does this alignment owe its credit to the active learning mechanism that identifies the most informative representatives, or does it also hail from the deep generative model's prowess in inferring target samples' cells with finesse? To answer this, Fig. 2d uses distinct colors to delineate real-profiled cells from representatives and generated cells for target samples. It shows that our representative selection strategy is able to select representatives such that their cells have a relatively good coverage of the overall cell distribution. This emphasizes our method's capacity to extend beyond the representatives, enriching the dataset. Meanwhile, the deep learning model managed to generate cells to complement the rest and make the overall semi-profiled cohort almost identical to the original real-profiled cohort. The effectiveness of the deep learning model in the semi-profiling process is further demonstrated in Fig. 2e and f and Supplementary Fig. S2. These figures illustrate the model's capability in reconstructing data from single representative samples and its proficiency in inferring data for individual target samples. Supplementary Figs. S3–S5 provides further details regarding the cell type distribution in each sample/sample cluster and *scSemiProfiler* is capable of generating similar sample clusters in terms of UMAP visualization and cell type distribution.

To test the fidelity of single-cell gene expression semi-profiling, we tested the interferon (IFN) pathway gene set—crucial for the innate immune response against COVID-19[46–49]. Through the prism of our semi-profiled dataset, Fig. 2g reveals IFN activation patterns that harmonize with the real-profiled dataset. The uniformity in IFN activation patterns across various key cell types and severity levels, as highlighted by similar heatmaps, confirms the effectiveness of the semi-profiling technique. This uniformity indicates that the critical disease-related pathways were effectively captured and maintained in the semi-profiled data.

Further, we explored the quantitative metrics of efficacy and cost-effectiveness in semi-profiling, Fig. 2h. Our analysis centered on understanding the relationship between the number of representative samples used for single-cell sequencing and the associated semi-profiling error. While an increase in representative samples intuitively raises the costs of single-cell sequencing, *scSemiProfiler* effectively leverages the single-cell data of these representatives for accurate inference of target samples. As more representatives are selected, our semi-profiling method effectively reduces the normalized error (see the Methods section for detailed error computation). We also compared our method to a selection-only method, which uses the same representatives chosen by *scSemiProfiler* using the active learning algorithm. For each target sample, it performs the single-cell data inference in a naive manner: merely copying the corresponding representative's data. The dashed line shows that our semi-profiling method has a huge lead over the selection-only method. The star symbol denotes the number of representatives selected for the specific semi-profiled cohort that was utilized for subsequent analyses, along with the associated error. The vertical dashed line underscores that by using the same representatives, our method achieves substantially lower errors compared to the selection-only method. Moreover, the horizontal dashed line demonstrates that the selection-only method requires a considerably larger pool of representatives to attain the same level of semi-profiling accuracy as our method. The comparative analysis between *scSemiProfiler* and the selection-only method highlights the deep learning model's efficacy in reducing costs and minimizing errors.

In further evaluating the effectiveness of semi-profiling, defined as a single-cell granularity bulk decomposition, we turned to the conventional realm of cell type proportion metrics, as anchored by Fig. 2i. Existing research[45] elucidates that PBMC cell type proportions undergo dynamic shifts with evolving disease conditions. True to this, our real-profiled dataset indicates a pronounced expansion of B cells and CD14 cells under aggravated conditions—a pattern mirrored in the semi-profiled dataset. The Pearson correlation coefficient[50] of cell type composition associated with different disease conditions

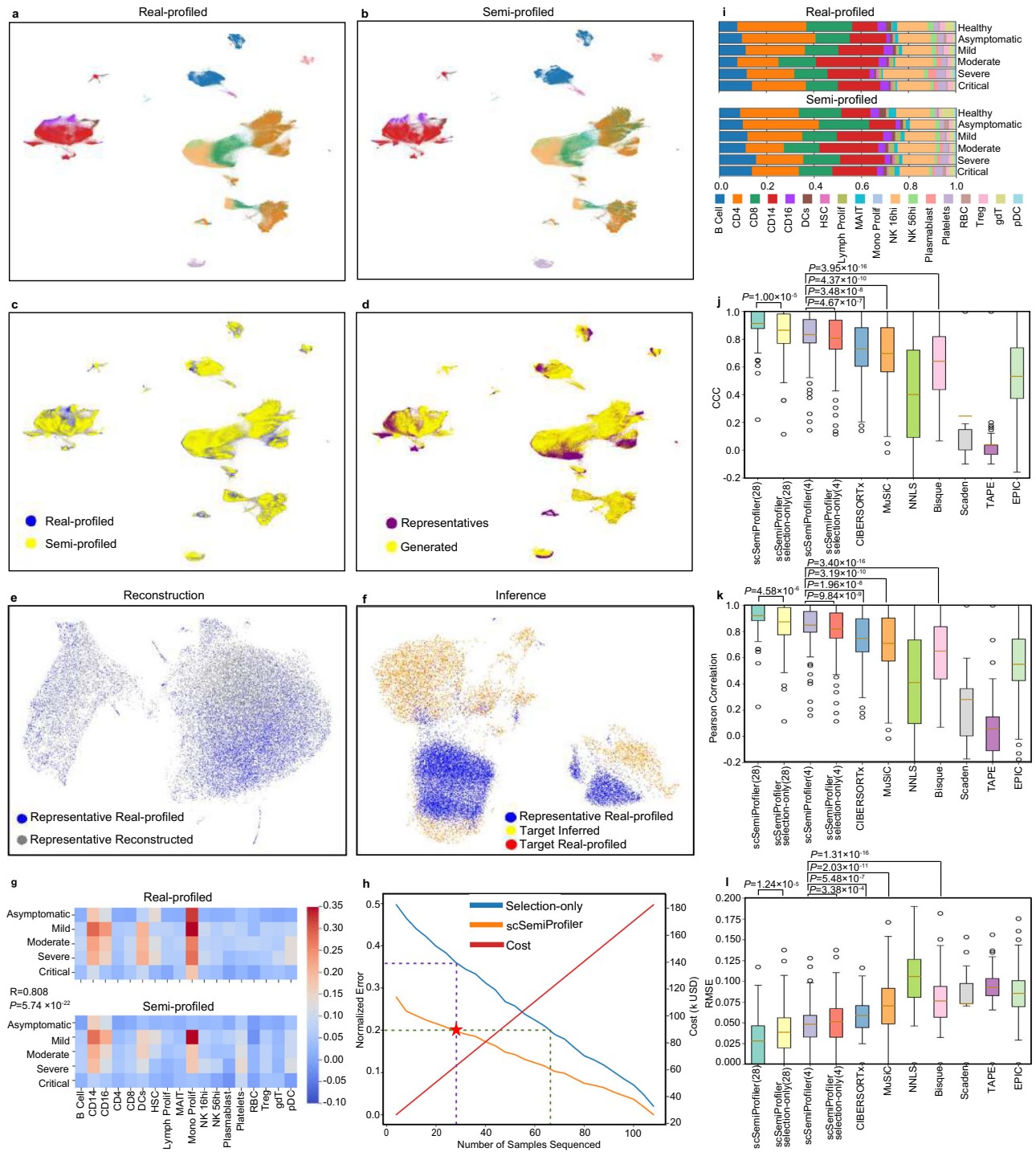

between the semi-profiled and the real single-cell datasets consistently surpasses 0.9.

In our comprehensive analysis of cell deconvolution methods, we meticulously compared *scSemiProfiler* with several leading-edge techniques, including CIBERSORTx[12,13], Bisque[14], Scaden[18], TAPE[19], EPIC[17], NNLS and MuSiC[16], as depicted in Fig. 2j–l. Each method was tested under identical conditions, using the same bulk data and single-cell reference data. A key challenge in this analysis was the memory constraints encountered by most methods, elaborated in Supplementary Fig. S1b, which hindered their ability to process the full set of 28 representative single-cell data. To ensure a fair comparison across all methods, we limited the single-cell reference to the same initial batch of four representative samples. Additionally, we demonstrated the

performance of our method using a set of 28 representatives that can not be fully exploited by other benchmarked tools. Our results unequivocally demonstrate the superior deconvolution performance of *scSemiProfiler* over all benchmarked methods. Its effectiveness is not only apparent when utilizing a smaller reference set of 4 samples but also becomes increasingly pronounced with a larger set of 28 samples. This distinct superiority of *scSemiProfiler* is a testament to its efficiency and versatility. Capable of excelling with both compact and extensive single-cell reference datasets, *scSemiProfiler* stands out as the most adept tool in cell type deconvolution, surpassing its contemporaries in handling diverse data scales with unparalleled precision and reliability. In Supplementary Fig. S6, we also show the high deconvolution accuracy using a side-by-side comparison between

**Fig. 2 | Overall comparisons of the semi-profiled and real-profiled COVID-19 dataset. a** UMAP visualization of the real-profiled data. Colors correspond to cell types and are consistent with (**g**). **b** UMAP visualization of the semi-profiled data. **c** UMAP visualization of semi-profiled data and real-profiled data together. The color differentiation signifies whether cells originate from the semi-profiled or the real-profiled dataset. Areas of overlap between the two indicate where the semi-profiled data closely resembles the real-profiled data. **d** UMAP visualization of the semi-profiled cohort, displaying different colors to distinguish cells produced by a deep generative model (labeled as "Generated") from the representative cells obtained through real-profiling (labeled as "Representatives"). **e**, **f** An illustrative case of in silico single-cell data inference for a target sample (AP1) using a representative sample (AP6) from the COVID-19 cohort is presented. **e** UMAP visualization compares the reconstructed single-cell data of the representative (in gray) against the real-profiled single-cell data of the representative (in blue). **f** UMAP illustrates the cell distribution of the representative sample (blue), the inferred target sample cells (yellow), and the real-profiled single-cell data of the target sample (red). The inferred cells exhibit a much higher resemblance to the real-profiled target sample data than to the representative's data. **g** Visualization illustrates the relative activation patterns of the interferon pathway. The comparison of these values between the semi-profiled and real-profiled matrices yields a Pearson correlation coefficient of 0.808 and a two-sided p-value of $5.74 \times 10^{-22}$. **h** Graph depicting the normalized error in semi-profiled data with an increasing number of

representatives. The terms "scSemiProfiler" and "Selection-only" represent our semi-profiling method and a method that only selects representatives using an active learning algorithm, respectively. It is important to note that actual costs may vary based on the sequencing technology and the specific number of cells sequenced. **i** Stacked bar plot illustrating the proportions of cell types across various disease conditions. The upper portion represents the real-profiled data, while the lower portion depicts the semi-profiled data. Pearson correlation coefficients comparing cell type proportions between the real-profiled and semi-profiled datasets are provided for different conditions: Healthy (0.987), Asymptomatic (0.970), Mild (0.996), Moderate (0.992), Severe (0.978), and Critical (0.989), indicating a high degree of similarity between the two datasets across these conditions. **j**–**l** Cell type deconvolution benchmarking. Results for all $N = 124$ samples in this COVID-19 dataset are shown. The boxes represent the interquartile ranges (IQRs), and the solid lines indicate the means. The whiskers extend to points within 1.5 IQRs of the lower and upper quartiles. P-values are from one-sided Wilcoxon tests without adjustment. **j** Figure displaying Lin's Concordance Correlation Coefficient (CCC) between actual (ground truth) cell type proportions and those estimated by various deconvolution methods. Except for the first two columns, all other columns' results are based on 4 representatives' single-cell data as reference. **k** Comparison of Pearson correlation across various deconvolution methods. **l** Comparison of Root Mean Square Error (RMSE) across various deconvolution methods. Source data are provided as a Source Data file.

each sample's predicted cell type proportion using 28 representatives with the ground truth.

Different from existing deconvolution methods, since our method is capable of "deconvoluting" the bulk samples into single-cell resolution, we can perform de novo cell type annotation for the generated single-cell data. To further substantiate this unique functionality, we applied de novo cell type identification to the semi-profiled COVID-19 dataset. We then compared the de novo cell type annotation results, derived from the single-cell data profiles generated by our model, with the ground truth. Briefly, de novo labeling of cell types can be achieved from our semi-profiled single-cell data using a biomarker-based strategy similar to those employed in other relevant studies[51,52] Specifically, we identified the top biomarkers associated with each cell cluster identified from the semi-profiled single-cell data. These biomarkers were then compared with known cell type markers from databases such as CellMarker[53,54] or Panglao DB[55] to annotate the cell types de novo. We compared the de novo cell type identification results with supervised cell type annotations and demonstrated the effectiveness of *scSemiprofiler* in de novo cell type identification (Supplementary Fig. S7).

Results presented above are based on selecting representatives using our active learning algorithm. To examine the effectiveness of *scSemiProfiler*'s global mode and compare it with results obtained under active learning for selecting representative samples, we employed the global mode to select the same number of representatives (28). We then used the deep generative model to infer the single-cell data for the remaining samples, thereby creating a semi-profiled dataset. This dataset underwent the same analytical procedures as those applied to the dataset generated under active learning. The results, which are detailed in Supplementary Fig. S8, indicate that although the global mode's performance is slightly inferior to that of the active learning mode, it remains quite comparable. However, it is important to note that, unlike active learning, the global mode cannot terminate before the predefined number of representatives has been profiled—even if fewer representatives could achieve similar performance. Therefore, this one-round global mode does not offer the same budget-minimizing flexibility as the multiple-round active learning mode.

### The semi-profiled COVID-19 single-cell cohort proves reliable for single-cell downstream analyses

We have previously demonstrated the capability of *scSemiProfiler* in accurately generating semi-profiled single-cell data that closely aligns with its real-profiled counterpart. Moving beyond basic cell type

proportion predictions, which is the primary focus of other methods, *scSemiProfiler* excels in predicting gene expression for each cell within a population, thereby more authentically mimicking true single-cell data. This advancement is crucial for more complex downstream single-cell analysis tasks.

To illustrate the effectiveness of semi-profiled data in standard downstream single-cell analyses, we conducted a series of evaluations. A key task in these analyses is the identification of biomarkers within distinct cell clusters, highlighting genes with distinct expression patterns. Utilizing the semi-profiled data generated by *scSemiProfiler*, we performed various single-cell level downstream analyses. The results from these analyses, as depicted in Fig. 3, demonstrate a significant consistency with outcomes derived from real-profiled data. For instance, we identified top cell type signature genes using the real-profiled cohort. When comparing their expression patterns in both real-profiled and semi-profiled datasets, the similarities were striking. The dot plots in Fig. 3a display these patterns, showcasing an almost indiscernible difference between the datasets. The semi-profiling at the single-cell level provides high-resolution expression data for marker genes within each cell population (cell type). Our approach reveals the distribution of marker gene expression across all cells within a specific type. While existing bulk deconvolution methods can offer average gene expression data for cell type markers, they fall short in depicting detailed gene expression distribution and variations among individual cells within the same population. This high level of cell type biomarker concordance underscores the *scSemiProfiler*'s robustness not only in replicating single-cell data but also in ensuring the fidelity of downstream analytical processes.

We further explored the similarities between biomarkers identified using semi-profiled data and those from real-profiled data. Biomarker discovery is possible with lower-resolution data, such as the average cell type gene expression data provided by current bulk deconvolution tools. However, these methods are less reliable compared to single-cell data. The limitation of average cell type gene expression data is its lack of replicates for each cell type, leading to reduced statistical power. The absence of replicates makes it challenging to estimate variance within a cell type, which is essential for standard differential expression tests. In contrast, our semi-profiled single-cell data supports robust biomarker discovery through rigorous statistical testing, as it includes multiple samples (i.e. cells) for each cell type. We demonstrate the similarity of biomarkers identified using real-profiled and semi-profiled datasets through a rank-rank hypergeometric overlap (RRHO) plot. An RRHO plot[56] visualizes the overlap

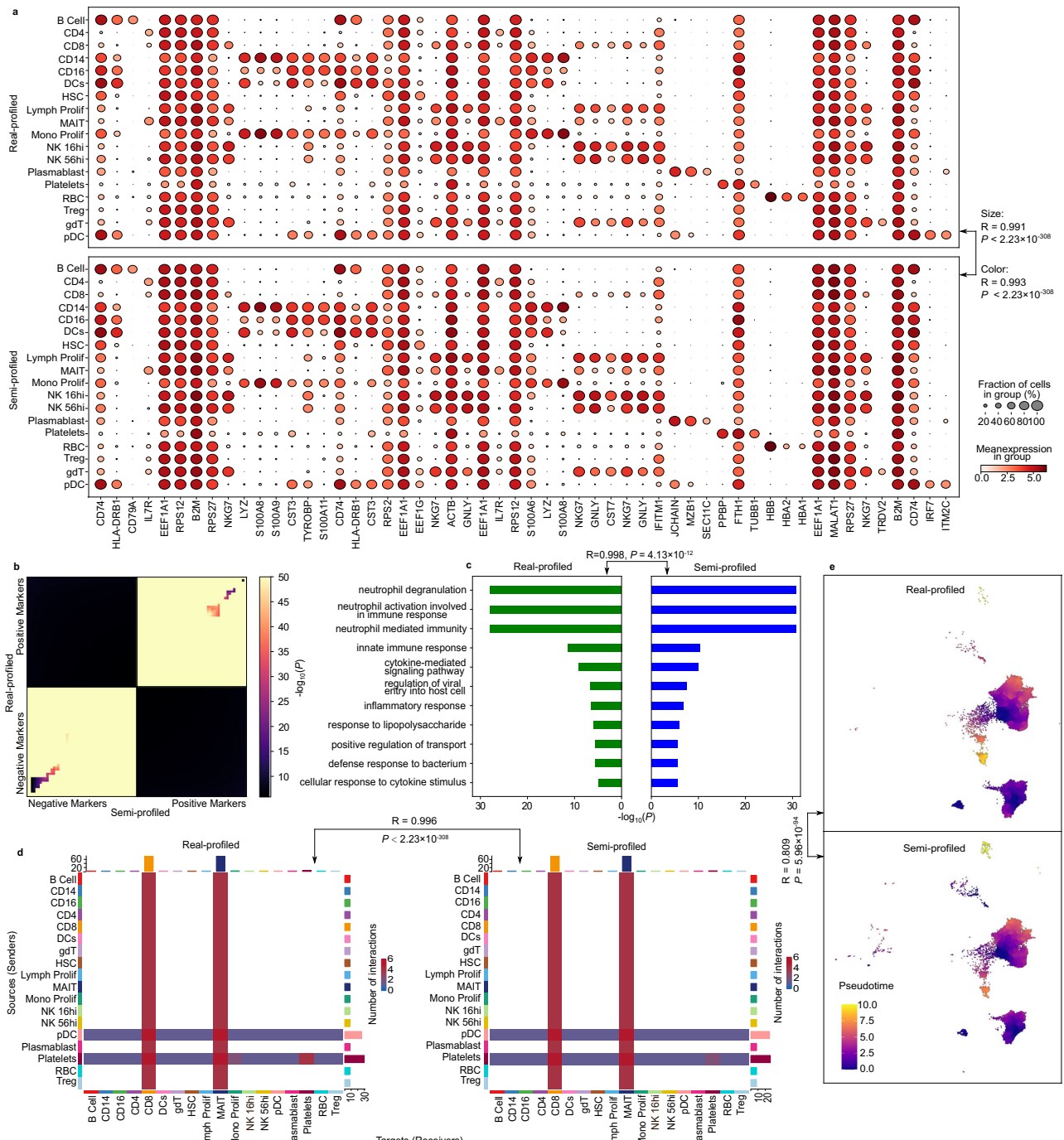

**Fig. 3 | Comparative analyses of single-cell level downstream analysis tasks using real-profiled and semi-profiled COVID-19 datasets. a** Dot plots elucidating the expression proportion and intensity of discerned cell type signature genes. The top half showcases the real-profiled dataset, while the bottom delineates the semi-profiled version. P-values are from two-sided Pearson correlation tests. **b** RRHO plot emphasizing the congruence between the CD4 positive and negative markers in both datasets. One-sided hypergeometric test p-values without adjustment are presented. **c** Visualization of the GO term enrichment outcomes rooted in CD4 signature genes from both dataset versions. The plot accentuates the union of the top 10 enriched terms, with the Pearson correlation coefficients of between the bar lengths, which is based on the corresponding p-value and therefore represents

the significant level. Benjamini-Hochberg (BH) adjusted one-sided hypergeometric test p-values are plotted. **d** A juxtaposition of cell-cell interaction analyses stemming from real-profiled and semi-profiled cells from moderate COVID-19 patients, underscoring the similarity in interaction types and counts. A two-sided Pearson correlation p-value is presented to show the similarity between the two interaction number matrices. **e** Comparative depiction of pseudotime trajectories for CD4 cells across both datasets, highlighting their striking similarity in reconstructing dynamic cellular processes. The p-value is from a two-sided Pearson correlation test. Source data are provided as a Source Data file.

between two ranked gene lists, highlighting the degree of similarity and the significance of the overlap between them (see the Methods section for more details). Leveraging the RRHO plots, we compared the top 50 positive and top 50 negative gene lists associated with different cell types (Fig. 3b for CD4 cells and Supplementary Fig. S9 for all other cell types. The plots show positive marker and negative marker lists from both datasets are highly similar. A marked dissimilarity was evident between the positive and negative marker lists, which is intuitively anticipated. By definition, positive markers are genes that are higher expressed in the corresponding cell types, and negative markers are the opposite—lower expressed. Therefore, they should have no overlap. The compelling concordance demonstrated in Fig. 3a, b bolsters our claim that the semi-profiled data from *scSemiProfiler* can viably supplant real-profiled data for the pivotal task of biomarker discovery.

Next, we used the biomarkers derived from the semi-profiled dataset and those from the real-profiled dataset for gene functional enrichment analysis, assessing whether the two versions of the analysis yield consistent results. Fig. 3c compares the Gene Ontology (GO)[57,58] enrichment[59,60] outcomes derived from real-profiled and semi-profiled datasets. The top 100 signature genes from both datasets are used for the enrichment analysis. We observed an overlap of 95 genes between the two lists, yielding a highly significant hypergeometric test p-value of $9.10 \times 10^{-196}$ (the population size of the hypergeometric test is the number of highly variable genes used for this dataset, 6030). A comparison of the top 10 overlapping terms from both versions reveals nearly identical significance (Pearson correlation coefficient of 0.998 with a p-value of $4.13 \times 10^{-12}$ for comparing the significant levels). The results for other cell types are in Supplementary Fig. S10. Reactome pathway[61] enrichment analysis results are in Supplementary Fig. S11. This further corroborates the reliability of semi-profiled data in downstream analyses.

Progressing to yet another pivotal single-cell level downstream analysis task, we evaluated the congruence in cell-cell interaction analyses derived from real-profiled and semi-profiled datasets. Given the paramount role of cell-cell interactions in orchestrating a myriad of multicellular processes, their analysis often unveils pivotal biological insights[62,63]. Fig. 3d juxtaposes cell-cell interaction analyses rooted in real-profiled and semi-profiled cells from moderate COVID-19 patients (see results for other severity levels in Supplementary Fig. S12). The evident concordance in types and counts of interactions in both renditions reinforces the reliability of our semi-profiled data ($R = 0.996$, $P < 2.23 \times 10^{-308}$). We also show a comparison of partition-based graph abstraction (PAGA) plots generated using real-profiled cohort and semi-profiled cohort in Supplementary Fig. S13, which demonstrates that the semi-profiled data can accurately capture the cellular trajectories and relationships between cell types. Given that such analyses intrinsically require single-cell data, *scSemiProfiler* emerges as the sole contender capable of producing data apt for this task from bulk sources.

Delving further into the capacity of semi-profiled data for other downstream single-cell level analysis tasks, we turned our attention to pseudotime analysis[29,30]. Pseudotime is a pivotal tool in reconstructing dynamic cellular processes, ranging from differentiation pathways to developmental timelines or disease trajectories. As depicted in Fig. 3e, the pseudotime trajectories derived from real-profiled and semi-profiled CD4 cells are strikingly similar (Consistent results for the pseudotime analysis of other cell types can be found in Supplementary Fig. S14). This similarity is supported by a high Pearson correlation of 0.809 (see the Methods section for details regarding pseudotime analysis and Pearson correlation test). Such compelling evidence underscores that the semi-profiled data retains its reliability even for intricate biological explorations like cell trajectory and differentiation analyses. The same downstream analysis has also been performed for the "global selection mode" of *scSemiProfiler* and is presented in Supplementary Fig. S15, which turns out to be only slightly worse than selecting representatives using active learning.

## In silico generated single-cell data offer additional insights beyond representative samples for understanding the studied cohort

Given the high similarity between the single-cell analysis results from the real-profiled and semi-profiled COVID-19 cohorts, a critical question arises: Does this similarity stem more from the accurate in silico inference of the deep generative learning model, or is it primarily due to effective representative selection through our active learning algorithm? To address this question, we divided the semi-profiled cohort into two components: the representative cells and the inferred cells, and conducted further comparative analyses. Our findings are presented from three perspectives: Analysis of the entire semi-profiled cohort is more informative than analyzing only the representative cells; the high similarity of the semi-profiled cohort with the real-profiled cohort, especially considering that most cells in the semi-profiled cohort are generated by the deep generative model, underscores the effectiveness of the in silico inference; comparisons between the analyses based on the real-profiled cohort and those using only inferred cells demonstrate the reliability and value of the inferred cells.

**At the cohort level:** The inclusion of generated cells significantly improves the overall data similarity to the real cohort and enhances analysis results compared to using only representative cells. The UMAP results, as indicated in Supplementary Fig. S16a−c, show that relying solely on cells from representatives fails to encompass many areas of the original cohort's UMAP. This leads to the omission of certain cell subtypes and a lack of intra-cluster variability. This observation is supported by a comparison between the semi-profiling and the "Representative-only" methods shown in Supplementary Fig. S16. These plots demonstrate that semi-profiling more accurately captures the real-profiled cohort, especially in terms of cell type proportions and overall gene expression, compared to using only representatives. While a sufficient number of representatives can achieve comparable results to the real cohort for straightforward tasks such as cell type marker identification (Supplementary Figs. S17 and S18), the semi-profiled cohort still presents advantages when conducting detailed single-cell analyses, such as pseudotime (Pearson correlation: 0.809 for semi-profiled vs. 0.545 for "representative-only", Supplementary Fig. S18d). It is also crucial to note that the effectiveness of representative data is enhanced by our selection strategies, which include bulk clustering and active learning.

**At the disease condition level:** Using only representative cells often fails to cover all disease conditions, which either precludes the study of specific conditions or leads to dramatically worse results compared to using the semi-profiled cohort. In the case of the COVID-19 cohort, we utilized 28 representatives out of 124 total samples for most analyses. These 28 representatives do not cover condition "Asymptomatic" condition, making it challenging to investigate this specific disease condition using single-cell data. For instance, the estimation of cell type proportions for the "Asymptomatic" condition cannot be achieved using only representatives, as there is no cells from this condition were profiled at the single-cell level. shown in Supplementary Fig. S16d. Moreover, even for conditions that are covered by the representatives, the analysis results are often less reliable due to a lack of statistical power or failure to capture the internal variety within each condition. For example, Supplementary Fig. S16d, g show that the semi-profiled cohort is significantly more similar to the real-profiled cohort than the "representative-only" version in terms of cell type proportion and pathway activation patterns. Notably, when investigating biomarkers for disease conditions, the semi-profiled dataset demonstrates a stronger similarity to the real-profiled dataset than the "representative-only" version, as illustrated in Supplementary Fig. S19.

For instance, in studying the "Critical" condition, the top 100 markers identified by the semi-profiled dataset have 68 overlaps ($P = 5.43 \times 10^{-109}$) with the real-profiled ground truth, whereas the "representative-only" dataset only has 30 overlaps($P = 4.31 \times 10^{-31}$). This disparity becomes even more pronounced when the number of representatives is reduced, often due to limited budget constraints in cohort studies. To illustrate this scenario with fewer representatives, we examined the COVID study using just 12 representatives and compared the findings with those from the semi-profiled results (utilizing the same 12 representatives). The results, presented in Supplementary Fig. S20 for the 12 representatives version, demonstrate that even with only 12 representatives, the semi-profiled dataset continues to maintain high similarity in terms of disease marker analysis, while the similarity of the "representative-only" results declines further.

**At the individual sample level:** The reliance on single-cell data from representatives alone for studying non-representative samples is not feasible without in silico inference, as these samples are not directly profiled. This limitation highlights two major concerns: Equity, Diversity, and Inclusion (EDI) and the potential for missed scientific discoveries. Excluding non-representative samples from analysis introduces biases in our understanding of the disease, as these samples are not represented, potentially leading to unfair representation and oversight of specific groups that may not be well represented by the selected representatives. Moreover, excluding non-representatives risks missing unique biological traits that representatives may not exhibit. However, *scSemiProfiler* enables the inclusion of these non-representative samples in single-cell analyses without additional costs. The generated single-cell data for individual samples closely resemble their corresponding target samples, as evidenced by UMAP visualizations in Supplementary Fig. S2.

Furthermore, representing non-representative samples directly using single-cell data from their corresponding representatives (as described in the "selection-only" method in our manuscript) can yield data too divergent from the ground truth to be meaningful. For example, Supplementary Fig. S21 shows a non-representative sample with a "Critical" severity level represented by its corresponding representative. This figure illustrates that the inferred single-cell data has cell type proportions dramatically more similar to the target sample than to the representative sample. In Supplementary Fig. S21b, we present 40 target sample biomarkers, differentially expressed genes (DEGs) of this sample against a healthy control. The left 20 DEGs show gene expression more similar to the target sample than the representatives, and the right 20 show where the representatives have more similar patterns than our inferred sample. The visualizations clearly indicate that the inferred sample is significantly more similar to the target than the representative. Even among the least similar 20 genes on the right-hand side, more genes show similar average expressions in the three samples. The Pearson correlations between the inferred sample and the target sample (0.904) are significantly higher than those between the representative and the target sample (0.568). Importantly, enrichment analysis in Supplementary Fig. S21c, d reveals that these DEGs are crucially relevant to the immune response, such as "regulation of immune response", "Immune System", "Innate Immune System", "Cytokine Signaling in Immune System", and several terms relevant to MHC[64], interferon pathway[49,65,66], indicating that using only the representative for analysis could likely lead to incorrect conclusions about the studied disease, since many of those key disease-associated immune terms would be missed. Additionally, extensive analyses have been conducted to further validate the effectiveness of the in silico generated cells. Supplementary Fig. S22a demonstrates that the deep generative learning model successfully generates the majority of cells, specifically those inferred for non-representative samples, across all datasets examined. The generally positive outcomes highlight the accuracy of these generated cells; any significant deviation from real-

profiled data could compromise the overall results of single-cell analyses. Moreover, the cell type proportions obtained using only the generated non-representative cells closely align with those from the real-profiled datasets, as shown in Supplementary Fig. S22b, further affirming the reliability of the inferred cells. Our model's detailed generative process is designed to maintain the integrity and variety of cell types, addressing concerns that replicating representatives might not accurately represent their associated target samples. Another comparative analysis between these semi-profiled non-representative cells and the original real-profiled cells within the COVID-19 dataset serves as a robust demonstration of our model's effectiveness. This comparison focuses on cell type distributions and pathway activations, revealing high concordance in cell type percentages and the preservation of key functional attributes, such as interferon activation pathways crucial for the immune response to COVID-19. The similarity in these critical biological aspects is illustrated in Supplementary Fig. S23. Comprehensive downstream analyses—including biomarker identification, pathway enrichment, cell-cell interaction studies, and pseudotime trajectory analyses-performed using the inferred cells further validate the integrity and biological relevance of our generated data. The results of these analyses, detailed in Supplementary Fig. S24, demonstrate the model's ability to produce data that is not only quantitatively robust but also qualitatively insightful, thus enhancing the dataset with a more representative spectrum of cellular states for advanced biological exploration.

## Semi-profiling maintains accuracy on a heterogeneous colorectal cancer dataset

We further tested the effectiveness of *scSemiProfiler* by validating it against a notably heterogeneous colorectal cancer dataset[67], which encompassed 112 single-cell sequencing samples that passed the quality control. This collection comprised 19 normal tissues, 86 tumor tissues (including colorectal cancer subtype iCMS2 and iCMS3), and 7 lymph node tissues. Considering the inherent diversity of this dataset, achieving accurate semi-profiling could ostensibly test the limits of *scSemiProfiler*. Again, this study does not include paired bulk sequencing data. Therefore, we have utilized the pseudobulk data derived from single-cell analysis as a surrogate for actual bulk sequencing. Nevertheless, following a consistent data processing and semi-profiling protocol, and selecting 36 representatives in batches of 4, the semi-profiled dataset mirrored its real-profiled counterpart. This congruence manifested not only in visual similarity but also in cell type distributions and subsequent analyses outcomes. Using the same estimation method as applied to the COVID-19 cohort, the total cost for both bulk and single-cell sequencing to obtain this highly similar semi-profiled single-cell cohort is approximately $73,320. This price also includes the cost of bulk data for the cohort and represents only 43.6% of the $168,000 estimated for conducting single-cell sequencing on the entire cohort.

Figure 4a, b graphically highlight the similarity between the semi-profiled and real-profiled data using UMAP visualizations. These visualizations, color-coded according to cell types, show a substantial alignment between the datasets. The semi-profiled data mirrors the real-profiled data in terms of the location and shape of each cell type cluster. Notably, both datasets effectively segregate cell types such as plasma B, enteric glial, Mast, and epithelial. Additionally, a nuanced connection between fibroblast and endothelial cells is evident in both versions. Immune-centric cells like McDC, T_NK, and B cells are also accurately positioned in close proximity in both datasets, underscoring the precision of the semi-profiled data. This similarity is further emphasized in Fig. 4c, which showcases the significant overlap between the two datasets. Moreover, Fig. 4d employs distinct color schemes to differentiate between cells generated by the deep generative learning model and those from real-profiled representative

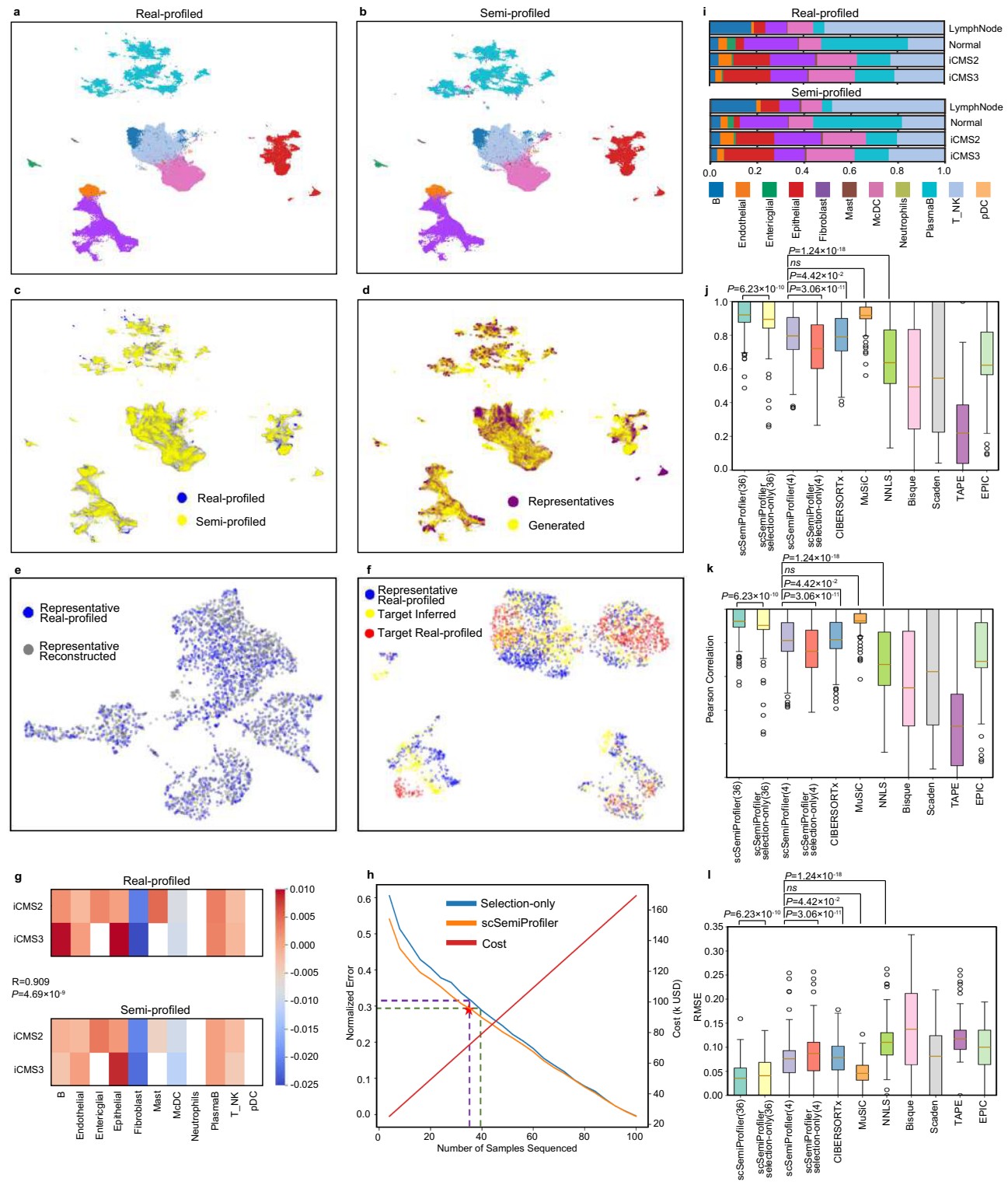

data. The cells from the representatives cover a substantial portion, indicating their well-chosen representative selection. However, numerous cells that fall outside the representatives' distribution are accurately generated by the deep learning model, highlighting the critical role of both active learning and the deep generative model in achieving effective semi-profiling. The accuracy of the deep learning model's generation can be further shown in Fig. 4e, f where we present the model's single-cell inference for an individual sample. More examples can be found in Supplementary Fig. S25. Supplementary Figs. S26–S28 provide further details regarding the cell type

distribution in each of the 36 sample clusters and individual samples and how *scSemiProfiler* performs in generating each of these sample clusters.

To further justify the semi-profiled gene expression values are accurate and can be used for biological analysis, we computed the gene set activation pattern of the GO term "activation of immune response" (GO:0002253) for the two tumor tissue types the same way as we did for the COVID-19 cohort (Fig. 4g). We chose this term because the immune response plays a significant role in the body's defense against cancer, and its activation or suppression can influence

**Fig. 4 | Detailed comparisons between the semi-profiled and real-profiled data in the heterogeneous colorectal cancer dataset. a** UMAP visualization of the real-profiled data, with colors denoting distinct cell types. Colors are consistent with (**g**). **b** UMAP visualization of the semi-profiled data, with colors denoting distinct cell types. **c** Joint UMAP visualization highlighting the close resemblance between the semi-profiled and real-profiled data. **d** UMAP plot of the semi-profiled dataset, with color-coding distinguishing cells from the actual sequenced representatives and the ones generated through semi-profiling. **e**, **f** UMAP visualization of the in silico inference of the target sample (MUX9009) using representative (EXT095). **e** UMAP showing the similarity of the reconstructed and the original representative. **f** UMAP showing the inferred target sample is more similar to the target ground truth than to the representative's cells. **g** "Activation of immune response" gene set relative activation pattern calculated for different tumor tissue types as compared to the "Normal" type in the real-profiled and semi-profiled datasets. Entries with fewer than 500 cells are left blank. The p-value is from a two-sided Pearson correlation test. **h**, Performance trajectory of the *scSemiProfiler* on the colorectal cancer

dataset, showcasing its superiority over the selection-only approach, with costs computed similarly to Fig. 2d. **i** Stacked bar plots comparing cell type compositions between the semi-profiled and real-profiled datasets across different tissues. The Pearson correlation coefficients between the real-profiled and semi-profiled tissues are LymphNode: 0.995, Normal: 0.993, iCMS2: 0.994, iCMS3: 0.988. **j**–**l** Cell type deconvolution benchmarking. Results for all $N = 112$ samples in the colorectal cancer dataset are shown. The interquartile ranges (IQRs) are depicted by the boxes, and the solid lines denote the mean values. Whiskers extend to points within 1.5 times the IQR from the first and third quartiles. P-values are derived from one-sided Wilcoxon tests without adjustment. **j** CCC between the actual cell type proportions and those estimated by various deconvolution methodologies. **k** The Same comparison using Pearson correlation coefficients. **l** Root Mean Square Error (RMSE) comparisons among different deconvolution techniques, highlighting the computational efficiency and accuracy of the *scSemiProfiler*, especially with an extended set of representatives. Source data are provided as a Source Data file.

cancer progression and patient outcomes[68,69]. The gene set activation scores are calculated and then adjusted by subtracting the score of the "Normal" tissue. The activation patterns in the real-profiled and semi-profiled datasets are highly similar, leading to a high Pearson correlation coefficient of 0.909 between them.

We also quantitatively examined the overall performance of *scSemiProfiler* as different numbers of representatives are selected. As shown in Fig. 4h, while our approach trumps the selection-only method, the gap narrows in comparison to results on the COVID-19 dataset-owing largely to the colorectal cancer dataset's inherent heterogeneity. Despite this, the deep generative model's efficacy remains conspicuous, ensuring cost-effective error reduction. Also, if we aim to achieve an error as low as the previous COVID-19 cohort, which leads to almost identical analysis results as the real data, only half of the samples need to be selected as representatives. This still reduces the cost significantly.

Diving deeper into the cell type proportions within the colorectal cancer cohort, one discerns variations across different tissue types-"Lymph Node", "Normal", "iCMS2", and "iCMS3". Fig. 4i illustrates these differences. For example, "Lymph Node" contains an expanded population of B cells compared with other tissue types, "Normal" is enriched with PlasmaB cells, and the two tumor subtypes have a pronounced epithelial presence. Remarkably, the semi-profiled dataset captures these nuances with precision, underlining its capability to replicate intricate analyses with fidelity.

Lastly, the benchmarking of deconvolution results presented in Fig. 4j–l positions *scSemiProfiler* at the forefront, significantly outperforming most existing methods such as Bisque, TAPE, and Scaden. While its performance with four representatives is on par with CIBERSORTx, *scSemiProfiler* excels in computational memory efficiency. Further extending the representatives to 36 dramatically boosts the deconvolution accuracy. In contrast, existing methods such as CIBERSORT falter when tasked with handling a large reference set like 36 representatives, mainly due to their computational inefficiencies. This distinction underscores the *scSemiProfiler*'s distinct advantage unshared by its peers. To provide a more detailed perspective on our deconvolution outcomes, Supplementary Fig. S29 showcases a comparative analysis. It displays our predicted cell type proportions for each individual sample alongside the ground truth, enabling a side-by-side evaluation.

### *scSemiProfiler* ensures consistent downstream analyses between semi-profiled and real single-cell data in heterogeneous colorectal cancer cohorts

In the context of a heterogeneous dataset like this colorectal cancer one, the semi-profiled dataset stands robust, offering downstream analysis results that mirror the real-profiled data. This close resemblance is consolidated in Fig. 5.

A notable observation is the accuracy in analyzing biomarker expression pattern and their intra-cluster variation using semi-profiled data. Fig. 5a showcases dot plots for top cell type signature genes derived from the real dataset. These plots reflect an identical pattern in both the real-profiled and semi-profiled datasets. Further affirmation comes from the strong Pearson correlation coefficient between the colors (0.994) and sizes (0.996) of the dots. Notably, these correlation coefficients even surpass those observed in the more homogeneous COVID-19 dataset. The semi-profiled dataset also reproduces biomarker discovery results, establishing its credibility as a suitable stand-in for the real-profiled data in such analyses. This consistency is exemplified by genes like KRT18 and KRT8, exclusive to epithelial cells, corroborated by existing literature[70,71]. Another illustration is the unique expression of CPA3 and TPSB2 in Mast across both datasets. Beyond these top cell type signature genes, a granular examination of epithelial cells—encompassing the top 50 positive and negative markers—reinforces the congruence. All markers were identified using data at single-cell resolution through thorough statistical testing, a process unachievable with decomposed cell type-level data from standard bulk decomposition methods. As depicted in Fig. 5b, the preponderance of highly significant entries, with many p-values lower than $10^{-50}$, strongly indicates a high degree of similarity in marker lists between the two datasets. RRHO plots for additional cell types can be found in Supplementary Fig. S30.

Diving deeper into the gene functional enrichment analysis, Fig. 5c offers further validation. Analyzing the top 100 signature genes across both datasets reveals a staggering 96 common markers, yielding a hypergeometric test p-value of $2.13 \times 10^{-183}$ (population size = 4053). GO terms from both datasets, along with their respective p-values, showcase pronounced similarity. Although the heterogeneity of the dataset leads to a relatively lower Pearson correlation coefficient of 0.593, the overall patterns in the two plots remain statistically similar ($P = 0.042$), leading to the same scientific conclusions. Moreover, when considering the union of the top 10 Gene Ontology (GO) terms from both the semi-profiled and actual datasets (comprising a total of 12 terms), there is a significant similarity in terms of enrichment p-values. Notably, the two versions of the top 10 GO terms have 9 overlap terms. More comprehensive GO term and pathway enrichment analysis results for other cell types' signature genes are also consistent for the real-profiled and semi-profiled versions (Supplementary Figs. S31 and S32).

Despite the increased heterogeneity of the dataset, the analysis of cell-cell interactions with the colorectal cancer semi-profiled cohort remains promising. As illustrated in Fig. 5d, the cell-cell interaction analysis, when executed on the real-profiled tumor tissue cells and the semi-profiled counterpart, reveals substantial consistency. Navigating the intricate interaction patterns characteristic of tumor tissues, the semi-profiled data astonishingly replicates the

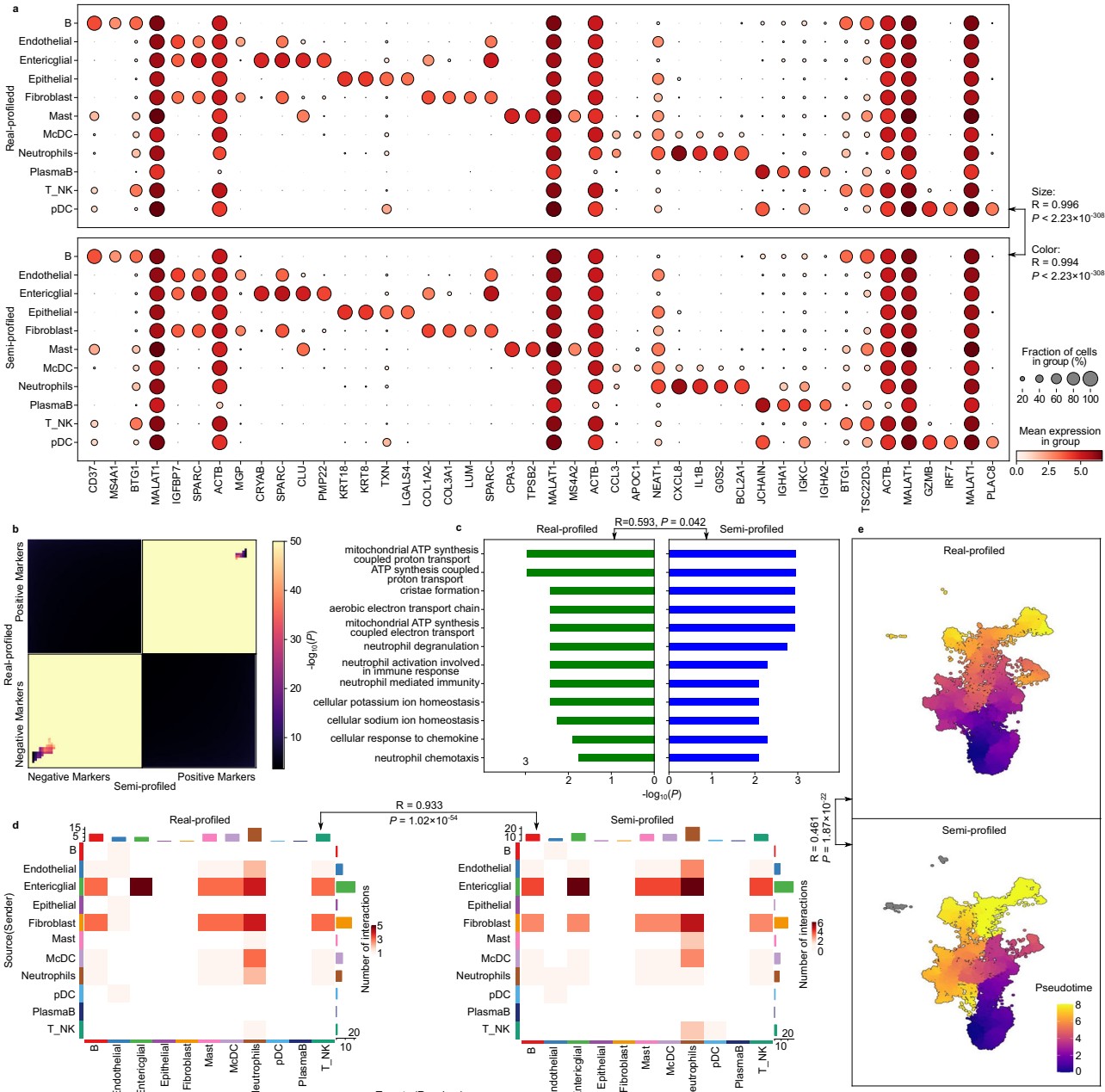

**Fig. 5 | Downstream analysis results comparisons for the colorectal cancer dataset. a** Dot plots visualizing the cell type signature genes. The p-values are from two-sided Pearson correlation tests. **b** RRHO plot visualizing the comparison between semi-profiled and real-profiled markers of epithelial cells. One-sided hypergeometric p-values are plotted. **c** Epithelial cell type signature genes GO enrichment analysis results comparison. Benjamini-Hochberg corrected one-sided hypergeometric p-values are plotted. The p-value on the top is from the two-sided Pearson correlation test. **d** Cell-cell interaction results comparison between the real-profiled tumor tissue cells and semi-profiled tumor tissue cells. A two-sided Pearson correlation p-value is plotted. **e** Pseudotime results comparison using the epithelial cells. A two-sided Pearson correlation p-value is plotted. Source data are provided as a Source Data file.

intricate layout with a robust Pearson correlation coefficient of 0.933. Both versions highlight enteric glial and fibroblast as the primary senders, while neutrophils emerge as the predominant receivers. Significantly, the most intense interactions identified across both sets involve the enteric glial with itself, the enteric glial with neutrophils, and the fibroblast with neutrophils. The cell-cell interaction results for other tissues also exhibit a high degree of similarity between the real-profiled and semi-profiled versions, as shown in Supplementary Fig. S33. Additionally, the strikingly similar PAGA plots presented in Supplementary Fig. S34 further demonstrate the utility of semi-profiled data in studying cellular trajectories and relationships between different cell types.

Shifting the focus to pseudotime analysis, we evaluated epithelial cells across all tissues as an example demonstration (Fig. 5e). Given the presence of cells from tumor tissues, this might introduce elevated heterogeneity within the cell type. Yet, the consistency between pseudotime analysis results from both versions is significant. Both versions discern lower pseudotime values concentrated at the base of the cluster, culminating in larger values towards the upper regions, with the pinnacle being the top-right quadrant. The statistical significance of the similarity is further validated by a Pearson correlation of 0.461 with a significant p-value of $1.87 \times 10^{-22}$ (see the Methods section for details about the Pearson correlation test). The pseudotime analyses for other cell types also

demonstrate a high degree of similarity, as evidenced in Supplementary Fig. S35. Such a finding underscores the capability of the semi-profiled data to adeptly capture intra-cluster nuances in detailed analyses.

## Semi-profiling with real bulk measurements yields a dataset nearly identical to the original single-cell data

To further illustrate the adaptability of *scSemiProfiler* in real-world applications, we directed our analyses towards the iMGL dataset[72], which uniquely profiles both single-cell and real bulk RNA-seq measurements for the human inducible pluripotent stem cell (iPSC)-derived microglia-like (iMGL) cells, differing from the pseudobulk datasets previously used. There are 25 samples having both single-cell data and bulk RNA sequencing data. Samples are of different conditions (grown in cell culture for 0-4 days and under various treatments). The availability of such datasets, which include both single-cell and bulk sequencing data on a large scale, remains very limited, partially due to the unnecessity of doing both sequencing for the same large-scale cohort and its prohibitive cost. Pseudobulk, created by averaging out the single-cell data, is often quite different from the bulk sequencing from the same sample[73]. This is because single-cell data is typically noisier, largely due to shallow sequencing and the presence of "dropouts" (genes not detected) in single-cell data[74–76].

To navigate this complexity, we devised a method to infer the target sample pseudobulk directly from its real bulk data, real bulk, and pseudobulk for the representative (refer to the Methods section "Fine-tune the deep generative learning model to infer the single-cell measurements for the target samples" for comprehensive details). This technique allows us to accurately estimate the pseudobulk data of the target samples. Supplementary Fig. S36 shows that the average Pearson correlation between the inferred pseudobulk and ground truth pseudo bulk data is 0.997, significantly higher than the Pearson correlation between the real bulk and ground truth pseudobulk, which is 0.880. This approach enables us to more effectively utilize our deep learning model in the pseudobulk data space, which often aligns more closely with single-cell data as it is computed by directly averaging the single-cell data without any extra bias. Through this method, despite the intricate challenges of the iMGL dataset, the results in Fig. 6 illustrate that our semi-profiled data parallels the real-profiled data quite closely. By selecting only eight representative samples out of a total of 25 samples, we could notably reduce the overall cost without compromising on accuracy. For this smaller dataset, we employed active learning to select 8 representative samples in batches of 2. Using the same estimation method as in the other two studies, the total cost for acquiring both bulk and semi-profiled single-cell data through our method is approximately $21,750. This amount is just 58% of the estimated $37,500 required for conducting single-cell sequencing across the entire cohort.

The UMAP visualization, as presented in Fig. 6a, b, solidifies our findings. Here, the two versions - semi-profiled and real-profiled - show a high level of consistency. The cell distributions of various iMGL subtypes (as delineated by the dataset provider) in the UMAP follow nearly identical patterns. The detailed observation showcases that clusters like "C2: Activated, immediate-early", "C4: Activated, non-immediate-early", and "C5: Activated, immediate-early" are interconnected and primarily located on the right-hand side. Likewise, "C3: Homeostatic, proliferative" finds its position at the upper left, with "C1: Homeostatic, non-proliferative" and "C6: Freshly thawed" lying at the bottom left. This consistency transcends to Fig. 6c, emphasizing a consistent cell distribution across the two versions. Furthermore, Fig. 6d underlines the precision of *scSemiProfiler*, where a majority of cells in the semi-profiled version were accurately generated. The efficient coverage of representative cells in this figure also highlights that our active learning strategy remains robust even when navigating the challenges of real bulk measurements.

UMAP visualizations in Fig. 6e, f and Supplementary Fig. S37 show *scSemiProfiler*'s capability of accurately reconstructing a representative sample and infer single-cell data of a target sample from its real bulk data. Despite the similarity, there are discernible differences in cell distributions between the target and representative samples, with the semi-profiled version showing more resemblance. We can still see the tendency of the generated cells (yellow), distributed closer to the target ground truth (red) than to the representatives' cells (blue). This indicates that the semi-profiled samples are either denser or sparser in the same areas as their counterparts. Importantly, these findings underscore *scSemiProfiler*'s sensitivity in detecting subtle differences between cell subtypes—a task that poses a challenge for other deconvolution methods that rely on cell type signature genes. Supplementary Figs. S38–S40 shows that the cell distribution varies in each sample cluster and sample. However, the semi-profiled version of the data is able to capture these nuance variances.

Figure 6g provides an in-depth look at the accuracy of semi-profiled gene expression values. Since microglia are the resident immune cells in the brain, we checked the activation pattern of GO term "activation of immune response" in each cell type under each treatment. The semi-profiled dataset presents a highly similar activation pattern as the real-profiled dataset.

Figure 6h offers further evidence of the *scSemiProfiler*'s effectiveness. Here, the performance curve of the semi-profiled approach significantly undercuts the selection-only one, demonstrating its capability to limit semi-profiling errors while optimizing on costs.

An intriguing observation from the iMGL dataset is the cell type proportion's dynamic shifts under various experimental conditions (Fig. 6i). The transitions from iMGL_D0 to iMGL_D4, for instance, reveal a progressive increase in the proportions of C1 and C4 cells. Contrastingly, "C2: Activated, immediate-early" cells peak at iMGL_D1 and then decrease steadily. Although the intricate effects of drugs iMGL_GW and iMGL_T need further investigation, preliminary data suggests that elevated doses result in a surge of "C1: Homeostatic, non-proliferative" cells. Impressively, these intricate variations are mirrored in the semi-profiled dataset.

A juxtaposition of our deconvolution method against others on the iMGL dataset offers illuminating insights. The performance of CIBERSORTx and MuSiC, which was previously high in other datasets, significantly declines, possibly due to the inherent challenges of real bulk data and the nuanced similarities among cell types. It has been previously reported that MuSiC fails to generate reasonable results on real bulk datasets, leading to negative CCC[19]. Other methods, such as Bisque, TAPE, and Scaden, are more robust to those challenges, showing decent deconvolution performance. Despite these challenges, *scSemiProfiler* showcases resilience and consistently outperforms all its peers except Bisque, a fact further corroborated by the Wilcoxon test[77] (see p-values in Fig. 6j–l). The marginal difference between our *scSemiProfiler* and the selection-only method is a testament to both approaches nearing optimal performance in this specific context. Supplementary Fig. S41 further presents our accurate deconvolution for individual samples. The superior performance is attributed to our pseudobulk inference strategy described in the Methods Section, which was designed to address the systematic differences between pseudobulk (the average expression) of aggregated cells from the target sample. As shown in Supplementary Fig. S36, the real bulk expression differs significantly from the ground-truth pseudobulk of the target single-cell data, while our inferred pseudobulk closely resembles the true pseudobulk.

To further demonstrate the effectiveness of *scSemiProfiler*'s ability to deal with real bulk data, we generated pseudobulk of the iMGL dataset and performed the semi-profiling using these pseudobulk data. We used the generated single-cell data to perform the same analysis and compared it with the real bulk version of semi-profiling. The pseudobulk version of semi-profiling analysis results were detailed

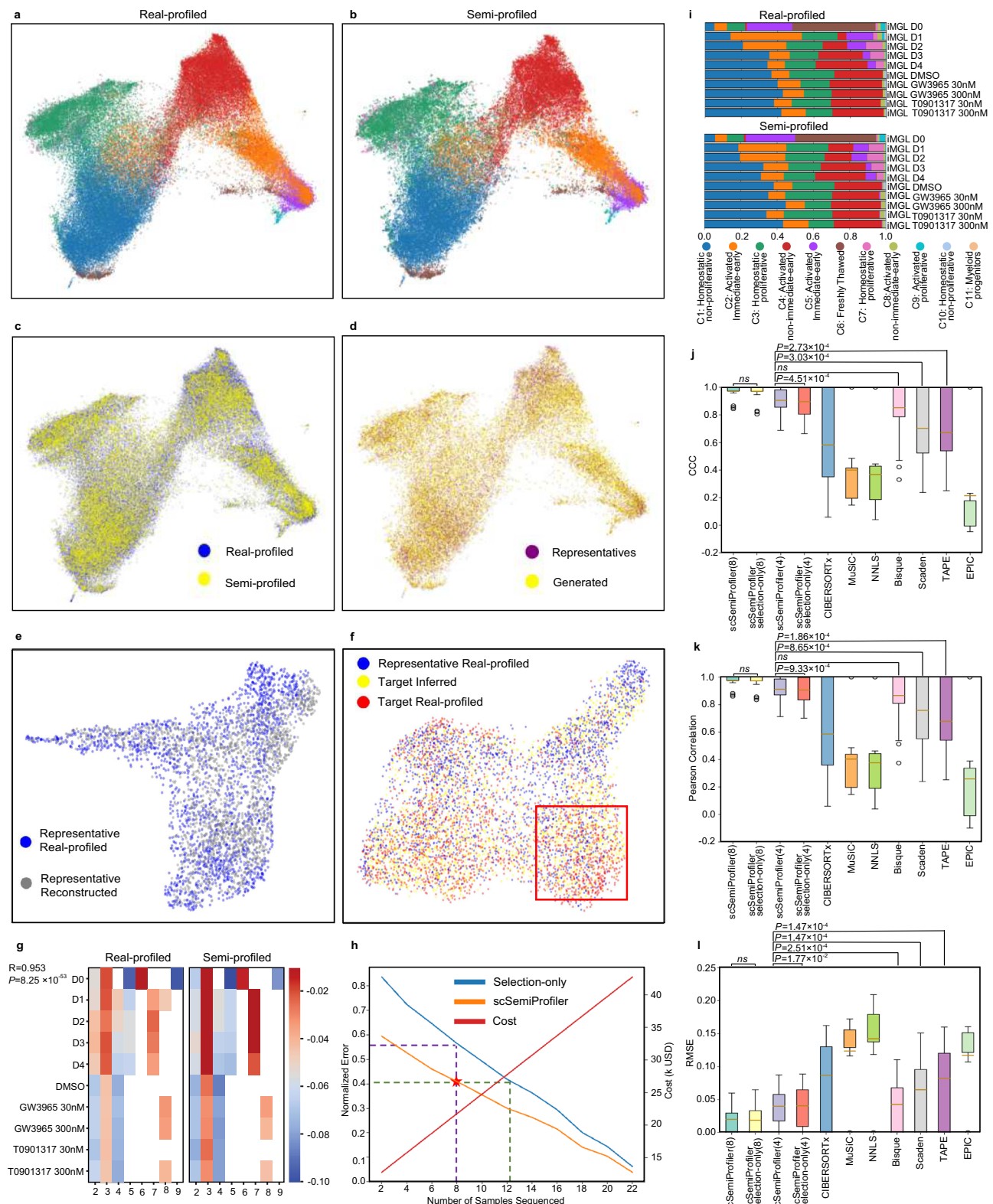

in Supplementary Fig. S42. Overall the results are very similar to the real bulk version, demonstrating that *scSemiProfiler* is able to handle the challenge brought by the real bulk data and maintain its high performance on pseudobulk data.

To further assess *scSemiProfiler*'s reliability with real bulk data and its adaptability to other species and tissues, we conducted an extensive search in the GEO database for datasets that include both bulk and single-cell data. We discovered a hamster lung dataset[78], which

represents a different species and tissue compared to the previously analyzed datasets. Our selection criteria focused on datasets with paired single-cell and bulk RNA-seq profiling of at least 15 samples, each averaging over 1000 cells. The average number of cells in each sample in the hamster dataset is 1,151.4, while in the colorectal cancer dataset is 1914.0, in the COVID-19 dataset is 5138.3, and in the iMGL dataset is 2974.9. Most samples in the hamster dataset could not pass our previous filtering criteria—samples with fewer than 1000 cells

**Fig. 6 | Comparative analyses between semi-profiled and real-profiled iMGL datasets. a** UMAP visualization of the real-profiled iMGL cohort. Different colors represent different cell types. Colors are consistent with (**g**). **b** UMAP visualization of the semi-profiled iMGL cohort. **c** Combined UMAP visualization showcasing the consistent cell distribution across both data versions. **d** UMAP visualization highlighting the representatives' cells alongside the semi-profiled cells within the semi-profiled dataset. **e, f** UMAP visualization of the process of generating target sample's (iMGL_D4_rep3) single-cell data using a representative (iMGL_D2_rep3) based on deep generative learning model. **e** UMAP visualization of the representative cells reconstruction. **f** UMAP visualization of inference. Compared to representative cells, inferred cells' distribution shows higher similarity with the target real-profiled cells (example region highlighted in red box). **g** Relative pathway activation pattern of the GO term "activation of immune response" calculated for cells of different treatments as compared to cell type "C1: Homeostatic non-proliferative" in the real-profiled and semi-profiled cohorts. Entries with fewer than 100 cells are left as

blank. The p-value is from a two-sided Pearson correlation test. **h** Performance evaluation of the *scSemiProfiler* on the iMGL dataset, emphasizing its efficiency in error reduction. Source data are provided as a Source Data file. **i** A comparative illustration of cell type proportions under varying experimental conditions, accentuating the similarity in patterns between datasets. The Pearson correlation coefficients between the real-profiled and semi-profiled versions of cell type proportions under different conditions are: iMGL_D0: 0.999, iMGL_D1: 0.871, iMGL_D2: 0.993, iMGL_D3: 0.989, iMGL_D4: 0.992, iMGL_DMSO: 0.998, iMGL_GW_30: 0.960, iMGL_GW_300: 0.999, iMGL_T_30: 0.987, iMGL_T_300: 0.999. **j**–**l** Deconvolution performance benchmarking using CCC, Pearson correlation and RMSE. The analysis covers all $N = 25$ samples in this iMGL dataset. The boxes illustrate the interquartile ranges (IQRs), with solid lines indicating the means. Whiskers extend to data points within 1.5 times the IQR from the lower and upper quartiles. P-values are obtained from one-sided Wilcoxon tests without adjustment.

should be filtered. A low number of cells significantly decreases the number of training samples for the deep generative learning model, making it hard for the model to learn the underlying data distribution. Despite its relatively lower data quality, characterized by fewer cells per sample and a smaller overall sample size, we still selected this dataset for analysis due to its unique composition and limited availability. We implemented the same analytical approach as with previous datasets, with results detailed in Supplementary Fig. S43. The close resemblance between the semi-profiled and real-profiled datasets, combined with *scSemiProfiler*'s significant advantage over other deconvolution methods (as evident in Supplementary Fig. S43f, h, i), confirms its effectiveness across diverse species and tissue types. By selecting only 2 out of 16 samples as representative (Supplementary Fig. S43d indicates these samples contained the majority of cells), *scSemiProfiler* successfully semi-profiled a single-cell dataset that closely mirrors the real-profiled ground truth. Additional outputs, including UMAP visualizations (Supplementary Fig. S43a−c) and precise cell type proportion estimates (Supplementary Fig. S43f−i), highlight its accuracy.

## Semi-profiling using real bulk measurement also leads to reliable downstream results

In this more realistic setting, where scSemiProfiler is tasked with semi-profiling using real bulk data for downstream analyses, the semi-profiled data consistently mirrored results from the real-profiled version. This is particularly remarkable given the unique challenges presented by the real bulk data (see the "Fine-tune the deep generative learning model to infer the single-cell measurements for the target samples" section in Methods for a detailed description of the challenges brought by real bulk dataset and our strategies for resolving them). We present these downstream analysis results in Fig. 7.

The markers identified using single-cell resolution data in the real-profiled dataset were almost identical in expression patterns to those in the semi-profiled dataset. Fig. 7a visually reinforces this, showing "C1: Homeostatic, non-proliferative" and "C10: Homeostatic, non-proliferative" with virtually indistinguishable expression patterns across both data types. Additionally, unique expressions in "C11: Myeloid, progenitors", such as GAPA2 and HPGDS, were consistently observed. The overarching similarities were further quantified with impressive Pearson correlation coefficients for both dot sizes and colors, clocking in at 0.980 and 0.989, respectively.

Further validating the congruency of our method, Fig.7b presented the RRHO plot of the top 50 positive and negative C3 markers from both single-cell datasets (see RRHO plots for other cell types in Supplementary Fig. S44). The degree of similarity is substantial, with most of the overlapping test entries showcasing p-values less than $10^{-50}$. Such findings strongly suggest that the *scSemiProfiler* is adept at producing reliable data for biomarker discovery.

Proceeding to more in-depth downstream analysis using the top 100 signature genes for GO enrichment, we observed an overlap of 90 genes between the semi-profiled and the real single-cell datasets (Fig. 7c). The hypergeometric test revealed a significant p-value of $1.81 \times 10^{-183}$ (population size of the hypergeometric test equals the total number of highly variable genes used, 6013). The enriched terms identified from both the semi-profiled and real datasets matched closely. The Pearson correlation coefficient between the two versions' significance is 0.995, underscoring the consistency in their analytical outcomes. Extended GO and Reactome enrichment analysis in Supplementary Figs. S45 and S46 further confirm the accuracy of finding signature genes using semi-profiled data.

In the case of cell-cell interactions, both real and semi-profiled single-cell data did not capture any significant interactions, probably due to the similarity between cell clusters in the iMGL data. Instead, the partition-based graph abstraction (PAGA) analysis[79] showcased in Fig.7d highlighted that major cell type links were consistent across datasets. Further cementing this was the strong Pearson correlation coefficient of 0.865 between the adjacency matrices of the two networks. Pseudotime analysis in Fig. 7e also affirmed the alignment between the two datasets. The topographical pseudotime alignment between them is almost congruent, justified by a Pearson correlation of 0.762 with a p-value $3.76 \times 10^{-77}$.

Additionally, as shown in Supplementary Fig. S47, the pseudobulk semi-profiling's downstream analysis results are similar to the real bulk version, further demonstrating *scSemiProfiler*'s ability to handle real bulk data. Furthermore, even in the challenging hamster dataset, all the downstream analysis as illustrated in Supplementary Fig. S48, demonstrates that the semi-profiled dataset supports robust downstream analytical tasks such as biomarker discovery, cell-cell interactions, and pseudotime analysis.

In conclusion, our findings robustly demonstrate that *scSemiProfiler* seamlessly adapts to real-world scenarios employing real bulk data. The downstream analytical outcomes derived from the semi-profiled data are significantly consistent with those based on real data.

## Active learning demonstrates its prowess in selecting the most informative samples for enhanced single-cell profiling

The crux of *scSemiProfiler*'s strategy revolves around judiciously selecting representative samples. The rationale is straightforward: the more informative the chosen representatives, the better the semi-profiling performance. This not only enhances the fidelity of the generated profiles but also provides a cost-effective approach by minimizing the number of necessary representatives.

Initially, our methodology bore similarities to uncertainty sampling[38,80,81], a heuristic active learning technique. Here, the intuition is to query samples with the most uncertainty, thereby maximizing the incremental information acquired. In our algorithm, we employed bulk data to pick out batches of samples that exhibited the

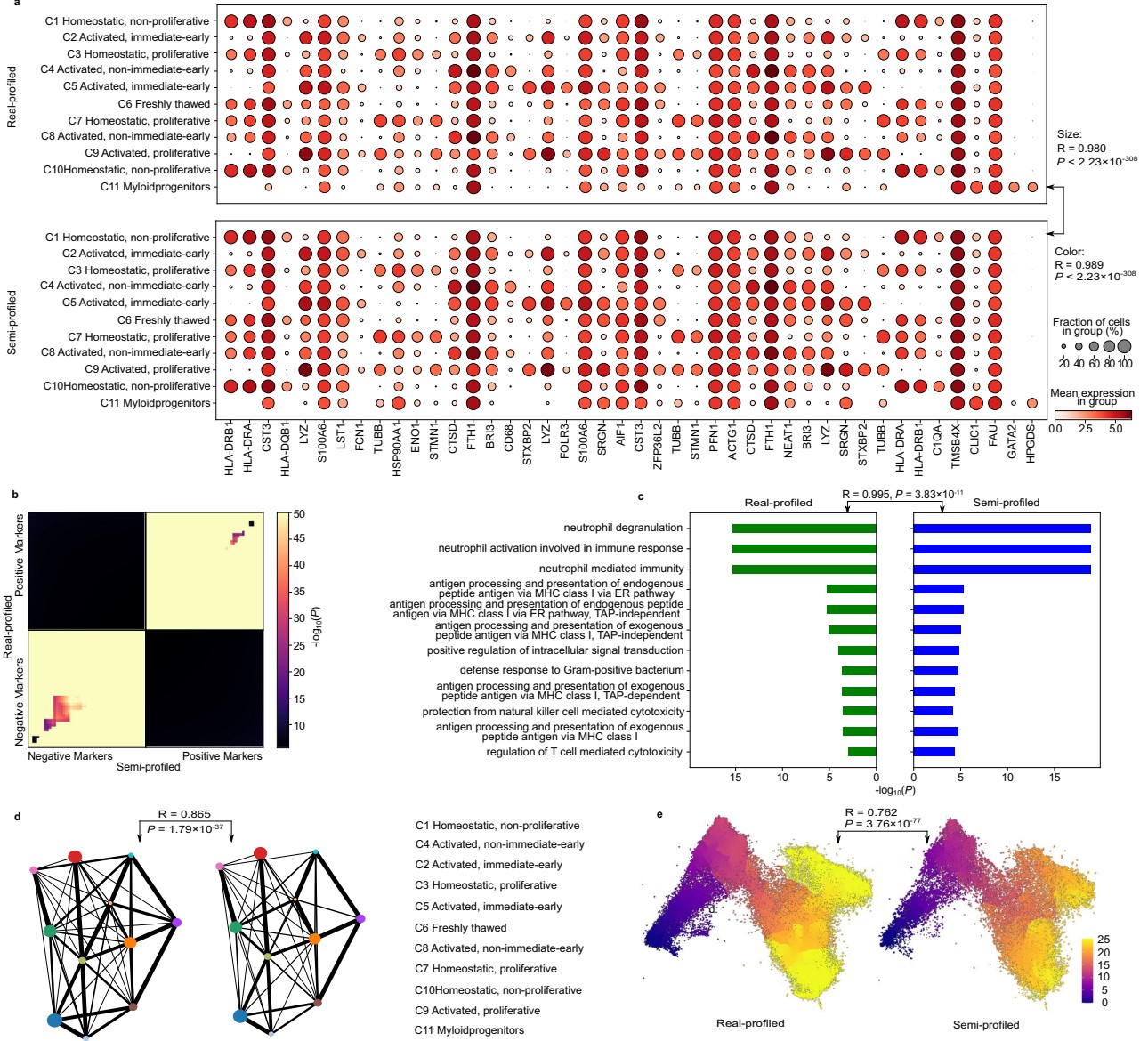

**Fig. 7 | Similarities in downstream single-cell analyses between real-profiled and semi-profiled data for the iMGL cohort. a** Dot plots visualizing the nearly identical expression patterns of cell type signature genes across both datasets. P-values are from two-sided Pearson correlation tests. **b** RRHO plots highlighting the striking similarities between the top 50 positive and negative C3 markers from both real and semi-profiled datasets. P-values are from one-sided hypergeometric tests. **c** Overlapping GO enrichment analysis results for the top C3 signature genes, emphasizing consistent analytical outcomes between the datasets. P-values plotted

are from one-sided hypergeometric tests and are Benjamini-Hochberg corrected. The p-value on the top is from a two-sided Pearson correlation test. **d** PAGA plots illustrating the consistent major cell type links observed in both datasets. The p-value is from a two-sided Pearson correlation test. **e** Pseudotime plots affirming the topographical alignment between the real-profiled and semi-profiled cohort. The two versions of datasets' pseudotime distributions show high similarity (Pearson correlation 0.762, two-sided p-value $3.76 \times 10^{-77}$). Source data are provided as a Source Data file.

most variance from their designated representatives. Since this method does not use information from the base learner (the deep generative model), it is still a passive learning algorithm.

To improve the representative selection, we then turn the algorithm into an active learning algorithm by incorporating the information from the deep generative models. The algorithm also utilizes the clustering information in the cohort and is thus a type II active learning algorithm[82]. Combining these two ideas, the algorithm aims to reduce the total heterogeneity of each sample cluster, ensuring that each target sample has a similar representative, thus optimizing semi-profiling performance.

As depicted in Fig. 8, we juxtaposed our advanced active learning algorithm against the rudimentary passive learning approach. Each

panel, from Fig. 8a–c, encapsulates the comparative analyses derived from distinct datasets. While the x-axis maps the representatives earmarked for single-cell sequencing, the y-axis portrays the single-cell in silico inference difficulty, which is quantified using the average single-cell-level difference (see Equation (17) in the Methods section) between target samples and their representatives. This premise holds that as the dissimilarity between the target and the representatives increases, the complexity of the in silico inference task also rises. This metric is crucial as it highlights the challenges encountered by the deep generative learning model in semi-profiling, thereby illuminating the effectiveness of the strategies used for selecting representatives.

The empirical evidence is resounding. Across all datasets, the active learning algorithm showcased its mettle by consistently

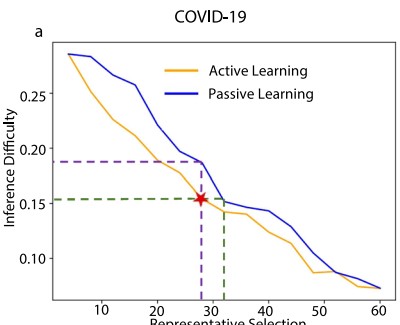
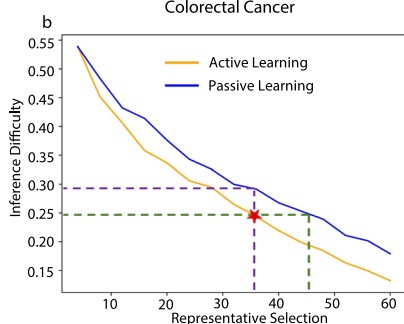
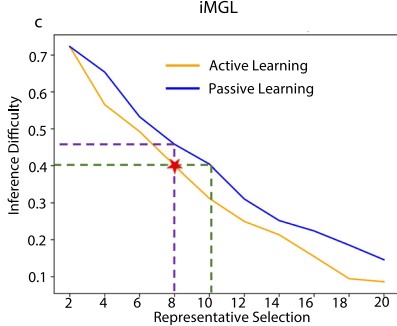

**Fig. 8 | Active learning demonstrates its prowess in selectively profiling the most informative samples at the single-cell level.** The x-axis represents the number of samples selected for single-cell profiling (representatives). The y-axis shows the single-cell in silico inference difficulty of the dataset, which is quantified by the average single-cell difference from each sample to its representative, showcasing the efficiency of representative selection strategies. The marked stars signify the iterations chosen for our methodology, with the generated data underpinning the analyses detailed in previous sections. **a** Results from the COVID-19 dataset. Active learning shows significantly better performance, especially in the beginning when a few representatives are selected. **b** Observations derived from the colorectal cancer dataset. Active learning continues to show significantly better performance even when more representatives are selected. **c** Insights from the iMGL dataset with real bulk measurements. Active learning still manages to outperform passive learning significantly. Source data are provided as a Source Data file.

pinpointing representatives that considerably reduced the total distance to other samples. This underscores the algorithm's capability to foster superior representative selection for semi-profiling. In Fig. 8a, when applying our method to the COVID-19 dataset, active learning shows better performance than passive learning, especially in the beginning iterations of the semi-profiling. Active learning reduces the inference difficulty when the same number of representatives are selected. Also, consider the marked point where 28 representatives are selected, which is consistent with the representatives we selected for our analyses in previous sections. To reach the same level of inference difficulty, passive learning needs to select 4 more representatives, i.e. the cost of 4 samples' single-cell sequencing experiment is saved. For the colorectal cancer dataset (Fig. 8b), active learning continues to perform better than passive learning, and at the point we selected, the cost for more than two batches (8) of representatives can be saved using active learning. Fig. 8c shows the results for the iMGL dataset, in which active learning is also significantly better than passive learning. In all experiments, active learning consistently outperforms passive learning in selecting representatives. With the same budget, active learning achieves lower inference difficulty. Furthermore, to reach a comparable level of inference difficulty, active learning requires a lower cost.

## Discussion

In the present work, we introduced *scSemiProfiler*, an innovative computational framework designed to produce affordable single-cell measurements for a given large cohort. The allure of *scSemiProfiler* stems from its unparalleled capacity to mirror the outcomes of conventional single-cell sequencing, yet with substantial cost-effectiveness. Marrying expansive sequencing with pinpoint profiling and optimizing data utilization, it charts a robust, intricate, and economical path for large-scale single-cell exploration. This tool is not merely a means of generating data but ensures the derived single-cell information remains dependable for an array of downstream analyses. The pipeline ingests bulk data, leverages an active learning module to judiciously select representatives for authentic single-cell sequencing, and employs a deep generative learning model, VAE-GAN, to infer the single-cell data of the remainder of the cohort. Through this approach, every sample in the cohort has access to single-cell data, either semi-profiled or real-profiled. We subjected *scSemiProfiler* to rigorous evaluation using three distinct cohorts, simulating a scenario devoid of single-cell data and then juxtaposing the output of our tool with the authentic single-cell datasets. The striking resemblance between semi-profiled and real-

profiled data in facets like UMAP visualization and cell type proportions ascertains the robustness of *scSemiProfiler*. Furthermore, both datasets align almost perfectly in subsequent analyses, encompassing biomarkers, enrichment analyses, cell-cell interactions, and pseudotime trajectories, offering a cost-effective means of integrating single-cell data in cohort studies without compromising on analytical precision.

The significance of scSemiProfiler's contributions lies in its defining properties: the precision of the method and its transformative capacity within the field of single-cell genomics.

**Precision-driven computational modeling:** Our study's primary contribution is the development and implementation of an iterative deep generative neural network framework with active learning, designed to enable single-cell level bulk decomposition. *scSemiProfiler* distinguishes itself from traditional cell deconvolution methods, which typically achieve only cell type level decomposition of bulk data. Our tool significantly advances this by actively deconvoluting bulk data into genuine single-cell data. This capability enables sophisticated analyses that delve into cellular heterogeneity, such as UMAP, pathway activation patterns, biomarker discovery, gene functional enrichment analysis, cell-cell interaction analysis, and pseudotime analysis-applications not possible with conventional deconvolution outputs. Notably, given the semi-profiled cohort, one can perform de novo cell type annotation for the single cells, which is another flexibility not owned by any other existing tools. Further setting *scSemiProfiler* apart, evaluations across four cohorts demonstrate its superior performance in traditional deconvolution tasks, particularly with real bulk datasets. This superior performance is attributed to its robust pseudobulk inference capabilities, which allow for more accurate and meaningful interpretations in practical settings, making *scSemiProfiler* a pioneering tool in the field. Furthermore, when placed under rigorous scrutiny, the semi-profiled data generated by our model consistently showcases significant similarity to real-profiled single-cell data, underscoring its reliability and potential to reshape the paradigm of single-cell analytics, particularly for large-scale cohort studies in which real single-cell sequencing cost would be prohibitive.

**Intelligent active learning mechanism:** Our second salient contribution hinges on the power of active learning. Instead of passively selecting representatives, our model embarks on an intelligent, iterative journey. By constantly evaluating the data and learning from previous rounds of selection, this module discerns which representatives would offer the most informative insights for semi-profiling. This isn't a mere addition; it's a reinvention of how representative selection operates. The synchronization of insights from the deep learning

model with this active learning module ensures that the selection process is not just data-driven, but also insight-driven. The implications of this are twofold: firstly, it enhances the quality and relevance of the selected representatives, ensuring that they truly resonate with the cohort's cellular composition. Secondly, it offers an economic advantage. By intelligently choosing representatives, the financial overheads associated with single-cell experiments can be minimized, ensuring maximum return on investment both in terms of data quality and budgetary constraints. Together, these innovations not only elevate the efficacy of *scSemiProfiler* but also pioneer a new direction in how single-cell data can be derived, analyzed, and utilized, holding profound implications for future biomedical research endeavors.

Future work on *scSemiProfiler* is focused on expanding its capabilities from single-cell RNA sequencing (scRNA-seq) to other biological modalities, such as proteomics. The adaptability of the tool's current methodology will require adjustments in algorithms to suit the unique data characteristics of each new modality, along with comprehensive performance evaluations benchmarked against established standards. A pivotal element of this development is the utilization of cross-modal information exchange, which aims to improve the accuracy of semi-profiling. This approach could significantly reduce the costs associated with single-cell multi-omics studies beyond the scope of the current study, by minimizing the need for extensive representative samples from each modality and leveraging the inherent similarities across various biological data types.

The *scSemiProfiler* represents a groundbreaking shift in single-cell analysis, particularly for large-scale and cohort studies, by offering a cost-effective yet comprehensive approach. It achieves this by selectively performing single-cell experiments on a few samples using active learning, coupled with deep generative models for in silico inference of single-cell profiles for the remaining samples, thereby creating similar semi-profiled cohorts at a substantially lower cost. This approach not only alleviates financial constraints in expansive biomedical research but also extends its utility beyond cohort studies. The tool's deep generative model enables "single-cell level deconvolution" across various studies, provided there is a single-cell reference. Initially centered on RNA sequencing datasets, *scSemiProfiler*'s adaptability to other single-cell and bulk data modalities opens new avenues in personalized medicine and can significantly enrich global data repositories. Significantly, scSemiProfiler is designed to complement, rather than compete with, large-scale single-cell sequencing technologies, such as WT Mega by Parse Biosciences. WT Mega is notable for its ability to process up to 1 million cells across 96 samples. When scSemiProfiler is integrated with these advanced sequencing platforms, it further reduces costs, enabling single-cell level profiling of thousands of samples in large-scale studies at a more manageable expense. This synergy significantly broadens the scope and depth of possible research, enhancing the efficiency and affordability of large-scale single-cell studies.

## Methods

In this section, we outline the methodological details for each module depicted in Fig. 1, presented in the following sequence: initial setup procedures, representative single-cell profiling and processing techniques, specifics of the deep learning model and the in silico inference process, the active learning algorithm for iterative representative selection, criteria for termination, and additional information concerning downstream analysis and evaluations.

### The initial setup of *scSemiProfiler* (INITIAL SETUP)
The initial setup of the scSemiProfiler method plays a pivotal role (depicted in Fig. 1a), serving as the foundation for subsequent semi-profiling iterations. Initially, each individual sample within the designated cohort undergoes bulk sequencing. Upon obtaining the raw count data from this sequencing, we perform library size normalization followed by a log1p transformation. From the processed data,

we identify and select highly variable genes and relevant markers for further analysis. In our experiments, we included more genes chosen by the default SCANPY's 'pp.highly_variable_genes' tool to avoid missing important markers. We also reviewed the original study generating the datasets to find important markers and manually added them to the highly variable genes. The highly variable genes are chosen using the 'pp.highly_variable_genes' tool in SCANPY. For the colorectal cancer dataset, due to its relatively sparser gene expression, we chose the top 4000 highly variable genes plus a few manually added markers. For the rest, we chose the top 6000 highly variable genes plus a few markers. The user can decide the number of highly variable genes based on their own dataset and GPU memory or stick with our default value 6000. Subsequently, we employ Principal Component Analysis (PCA)[83–85], implemented in the Python package Scikit-learn[86], to reduce the dimensionality of the processed bulk data. We employ a consistent approach to determine the number of principal components (PCs). Specifically, we use 100 PCs when there are more than 100 data samples to minimize information loss. For datasets with fewer than 100 samples, we use a number of PCs equal to the sample size minus one. We then determine the batch size of representatives ($B$) representing the number of selected representatives for actual single-cell profiling in each iteration. Once the representative batch size is defined, we cluster the dimensionality-reduced bulk data to select the initial batch of representatives. The dimensionality-reduced data is clustered into $B$ distinct clusters using the K-Means algorithm[87], the sample that is closest to the cluster centroid is designated as the representative for that cluster. This well-prepared initial setup provides the clustered bulk sequencing data and identified representatives for high-resolution single-cell sequencing.

### Representative single-cell profiling and processing
After the selection of representatives, the scSemiProfiler method proceeds to the single-cell profiling phase, focusing specifically on the chosen representatives (Fig. 1b). The raw count data obtained from single-cell profiling undergoes a sequence of preprocessing steps. These steps encompass traditional quality controls as well as advanced processing techniques aimed at enhancing the learning of the deep generative model.

To initiate, expression values that are extremely low (below 0.1% of the library size) are adjusted to zero. This adjustment is based on insights from prior studies[88,89], which have demonstrated that these minimal values are often representative of background noise and, therefore, should be excluded. The threshold for this processing is empirically determined by selecting the value that yields the most high-quality UMAP visualizations consistent with the original cell type annotations.

Following the noise removal, our preprocessing workflow encompasses several standard steps, including the removal of low-quality cells (expressing fewer than 200 genes, per SCANPY's default settings) and genes (expressed in fewer than three cells). We also remove dying cells identified by a high proportion of mitochondrial reads and doublets expressing an unusually high number of genes, both determined using quality control plots generated by SCANPY's 'pp.calculate_qc_metrics' tool. Subsequently, we normalize the data to a library size of 10,000 per cell and apply a log1p transformation. Low-quality samples with fewer than 1000 cells are removed, with an exception made for the hamster dataset to preserve an adequate sample size. The data matrix is then refined to include only highly variable genes identified from bulk data analysis. The COVID-19 dataset arrives already preprocessed as detailed by[45], including doublet removal with Scrublet[90], cell filtering, library normalization, and log1p transformation. In contrast, the colorectal cancer dataset, provided in raw counts, undergoes a comprehensive preprocessing regimen. We filter out cells expressing fewer than 200 genes and genes detected in fewer than three cells. Additionally, doublets expressing more than

6000 genes and dead cells with mitochondrial reads above 20% are removed. Samples with fewer than 1000 genes are also discarded. The dataset undergoes background noise removal, library size normalization, and log1p transformation, with a further selection of data columns based on highly variable genes. For the iMGL dataset, we process cells that passed the dataset provider's stringent quality controls, including thresholds of over 1500 genes detected per cell, fewer than 100,000 reads per cell, and mitochondrial RNA fractions below 20%. We also exclude any genes detected in less than three individual cells. Our preprocessing steps for this dataset include standard procedures like filtering low-quality samples, background noise removal, library size normalization, log1p transformation, and gene feature selection. The hamster dataset, characterized by its sparse single-cell data (averaging 1,151.4 cells per sample), bypasses cell, gene, and sample filtering to preserve data integrity, focusing solely on selecting genes common to single-cell and bulk data, removing background noise, normalizing library sizes, and selecting highly variable genes. This approach ensures we retain all samples for analysis despite the dataset's overall low quality, leveraging its unique pairing of single-cell and bulk data.

After the initial preprocessing, the single-cell data is further refined by integrating two types of prior knowledge to enhance cell representations. Firstly, gene set scores are calculated using all curated gene sets from MSigDB[59] and Ernst et al.[91], sourced from UNIFAN[92]. Gene set scores are determined by averaging the expression values for each gene set, followed by a log1p transformation, and then combined with the preprocessed single-cell gene expression matrix via concatenation. In the deep generative model, there are two separate encoders for encoding gene set scores and gene expression values, and two separate decoders for reconstructing their values. Both reconstructions will be involved in the reconstruction loss computation. The percentage of genes in the final features are COVID-19: 88.8%, colorectal cancer: 84.5%, iMGL: 89.8%, hamster: 90.0%. Secondly, we adopt a feature weighting method, giving more weight to features with higher variance in each cell's input when calculating reconstruction loss. More comprehensive details of this strategy are available in the section focusing on single-cell inference.

## Pretrain the deep generative learning model for reconstructing the single-cell data of the selected representatives (REPRESENTATIVE RECONSTRUCTION)

The next phase, depicted in Fig. 1c, involves training a deep generative learning model for the in silico single-cell data inference of a non-representative target sample using processed representative single-cell data and bulk data of both samples. This will be executed for all non-representative samples to ensure everyone in the study cohort will have single-cell data (real-profiled or semi-profiled) in the end. Firstly, our deep generative model aims to reconstruct the single-cell profiles of representatives. This reconstruction lays the foundation of the single-cell inference of the target samples, which can be viewed as a modified single-cell data reconstruction task. Specifically, we first pretrain a reconstruction model that's tailored to the representative of each cluster. We then fine-tune this model for each associated target sample, employing the target bulk loss to direct the transformation from the representative's profile to that of the target. This process is critical for ensuring the model's accuracy in reflecting the unique characteristics of each target sample (i.e., matching the bulk sequencing). When performing the single-cell inference after the reconstruction, the initial single-cell data reconstruction of the representative is modified by introducing the difference between the target sample and its representative based on their distinct bulk data profiling, aligning the synthetic single-cell data for the target sample closer to its actual single-cell profiling counterpart. The fundamental reasoning is that the single-cell data matrix of a specific target sample should bear a resemblance to the representatives chosen from the same study. Significant differences between them can be adjusted and guided by the disparity as delineated by the bulk sequencing data.

We designed a VAE-GAN-based model for the representative reconstruction and the target single-cell generation guided by their expression difference at the bulk level. It ingests gene expression data into an MLP encoder, which subsequently outputs parameters of a multivariate Gaussian. A random variable, $\mathbf{z}$, is sampled from this Gaussian and processed through an MLP decoder, resulting in parameters of a Zero-Inflated Negative Binomial (ZINB) distribution - optimal for modeling single-cell RNA-seq data[93–95]. Training the VAE parameters involves maximizing the data likelihood and minimizing the KL divergence[96] between the latent variable distribution and a standard Gaussian distribution, following the Evidence Lower BOund (ELBO) loss framework[40]. The geneset scores were concatenated with the gene expression data and factored into the loss calculations, essentially functioning as additional genes. A separate MLP in the encoder network is used for encoding gene set score values to the encoder's hidden layer, and a separate MLP in the decoder is used for generating gene set values from the hidden layer. However, their utilization is specifically tailored to improve the learning of the cell representation and they do not play a role in the computation of bulk tissue losses to guide in silico inference. To demonstrate the effectiveness of the gene set strategy, we compared the cell embedding quality generated with/without using gene set scores using an independent dataset[97] that is also employed in the SCVI study. We trained models using cells with only gene expression data and cells augmented with gene set scores separately. Then we used these two models to perform the reconstruction task for the dataset and found the one augmented with gene set score achieves lower MSE (19.33 vs. 25.48 Supplementary Fig. S49).

Innovating upon this foundational VAE structure, we integrated the following four techniques to enhance effective cell representation learning. **Gene set score inclusion:** As mentioned in the data preprocessing section, beyond mere gene expression reconstruction, we compute gene set scores for cells and concatenate them to the gene expression data. During the two stages of pretrain, we also compute the reconstruction loss for gene set scores. Such inclusion furnishes the model with an enriched biological context, fostering a more comprehensive learning of the input cells. Hence, input for each cell becomes the concatenation of gene expression and gene set scores: $(\mathbf{x_i}, \mathbf{s_i})$. **Feature importance weight:** We also compute a weight vector $\mathbf{w}$ to weight the contribution of each feature to the VAE's reconstruction loss based on the feature's variance. **Graph convolutional networks (GCN)[98,99]:** A GCN layer assimilates adjacent cells' information, mitigating dropout concerns and the inherent noise of single-cell sequencing. **Generative adversarial network (GAN)[41] dynamics:** We employ a discriminator network, resonating with GAN structures, which discerns between genuine and generated cell data. This guides the generator towards producing more authentic single-cell data. The resultant loss function for the generator discussed above is:

$$L_{G_{Pretrain1}} = L_{VAE} - \lambda_d L_{D_{Fake}} \tag{1}$$

$$L_{VAE} = \sum_{i=1}^{N} \left[ -\mathbf{w} \cdot \log(ZINB((\mathbf{x_i}, \mathbf{s_i})|\rho(\mathbf{z_i}), \pi(\mathbf{z_i}), \theta))) + KL(q(\mathbf{z}|(\mathbf{x_i}, \mathbf{s_i})) \| \mathcal{N}(\mathbf{0}, I)) \right] \tag{2}$$

$$L_{D_{Fake}} = \sum_{i=1}^{N} -\log(1 - D(G((\mathbf{x_i}, \mathbf{s_i})))) \tag{3}$$

$L_{G_{Pretrain1}}$ is the loss used to train the VAE-based generator network during the first pertrain stage. It has two components, a VAE loss $L_{VAE}$ and another loss $L_{D_{Fake}}$ including the feedback from the discriminator.

In $L_{VAE}$, $N$ is the number of cells in the representative's single-cell dataset. $\mathbf{x_i}$ is the vector representing the gene expression value of the $i$-th cell in the dataset and $\mathbf{s_i}$ is the corresponding gene set score. $\mathcal{N}(\mathbf{0},I)$ is the standard Gaussian distribution. $\mathbf{z_i}$ is a low-dimensional latent variable sampled from the Gaussian distribution $q(\mathbf{z}|(\mathbf{x_i},\mathbf{s_i}))$ whose parameters $\boldsymbol{\mu}_i$ and $\boldsymbol{\Sigma}_i$ are generated by the encoder network: $MLP_{Encoder}(GCN((\mathbf{x_i},\mathbf{s_i})))$. We follow the ZINB distribution design of SCVI[93] and generate ZINB parameter using the decoder network: $\boldsymbol{\rho}(\mathbf{z_i}) = MLP_{Decoder,\rho}(\mathbf{z_i})$, $\boldsymbol{\pi}(\mathbf{z_i}) = MLP_{Decoder,\pi}(\mathbf{z_i})$ and $\boldsymbol{\theta}$ is a free parameter vector that are learned in the training process. In $L_{D_{Fake}}$, $G$ represents the VAE-based generator so $G((\mathbf{x_i},\mathbf{s_i}))$ represents the $i$-th cell reconstructed by the generator. The reconstructed cell data is the mean of the generator's ZINB distribution. And $D$ is our discriminator network, so $D(G((\mathbf{x_i},\mathbf{s_i})))$ is the discriminator's predicted probability of the $i$-th reconstructed cell being a real-profiled cell. $\lambda_d$ denotes an empirically determined scaling factor. Meanwhile, the discriminator network will be trained using cross-entropy loss to reach higher classification performance.

$$L_D = \sum_{i=1}^{N} \left[ -\log(D((\mathbf{x_i},\mathbf{s_i}))) - \log(1 - D(G((\mathbf{x_i},\mathbf{s_i})))) \right] \quad (4)$$

Here, $D((\mathbf{x_i},\mathbf{s_i}))$ represents the discriminator's predicted probability of the $i$-th input cell being real when it is indeed real, and $D(G((\mathbf{x_i},\mathbf{s_i})))$, as mentioned in a previous paragraph, represents the discriminator's predicted probability of the $i$-th cell being real when it is, in fact, a cell reconstructed by the generator network. When trained together, the generator will first be trained until convergence, followed by alternating training with the discriminator every three epochs. Together, they form a GAN with a min-max objective:

$$\min_G \max_D \sum_{i=1}^{N} [\log(D((\mathbf{x_i},\mathbf{s_i}))) + \log(1 - D(G((\mathbf{x_i},\mathbf{s_i}))))] \quad (5)$$

In this context, the discriminator's objective is to maximize its accuracy, aiming for precise classification, while the generator's goal is to minimize discrepancies, generating realistic cells to challenge the discriminator.

Overall, our approach involves two pretrain stages and one fine-tune stage for performing single-cell inference. In the initial pretrain stage, we employ $L_{G_{Pretrain1}}$ and $L_D$ to train the generator $G$ and discriminator $D$, respectively, forming a GAN. The primary objective here is to train a generator capable of reconstructing genuine cell data from the representatives. Subsequently, the second pretraining stage closely resembles the first, without freezing any subnetwork, but it functions in full-batch mode. This modification enables the inclusion of an extra bulk loss term in the generator's loss function. Such an addition improves the model's capacity to leverage insights from bulk data and better aligns the single-cell data with its bulk counterpart. This enhancement is crucial for priming the model for the in silico inference of single-cell data from bulk data, while also strengthening the data reconstruction process.

$$L_{G_{Pretrain2}} = L_{VAE} - \lambda_d L_{D_{Fake}} + \lambda_{BulkR} L_{BulkR} \quad (6)$$

$$L_{BulkR} = \left\| \frac{\sum_{i=1}^{N} \mathbf{x}'_i}{N} - \frac{\sum_{i=1}^{N} \mathbf{x_i}}{N} \right\|_2 \quad (7)$$

The term $\lambda_{BulkR}$ represents another empirical hyperparameter for regularization weight. $\mathbf{x}'_i$ represents the reconstructed $i$-th cell. The bulk loss is defined as the disparity between the pseudobulk of the real representative dataset and that of the reconstructed representative dataset.

## Fine-tune the deep generative learning model to infer the single-cell measurements for the target samples blue (IN SILICO INFERENCE)

The in silico inference is achieved by performing reconstruction of the representative single-cell sample and then "transforming" it to match the pseudobulk (the average expression) of aggregated cells from the target sample. After successfully pretraining our deep generative model so that it can perform accurate single-cell data reconstruction, the subsequent stage of our method leverages this foundation and incorporates bulk data information to perform the "transformation". Intuitively, the incorporation is achieved by matching the bulk expression calculated from the generated cells and the bulk expression of the target sample. However, it will be inaccurate and even misleading to "guide" the transformation from the reference single-cell to the target single-cell if using the actual bulk measurement to calculate the "loss" for the in-silico inference, since the pseudobulk of the target single-cell sample is very different from its true bulk counterpart (Supplementary Fig. S36). This is because due to the difference in sequencing platform and technology, as well as the relatively higher level of dropouts and noise, the pseudobulk of the target single-cell sample is often quite different from the bulk sequencing from the same sample[73], due to the "dropouts" (gene not detected)[75,76] in single-cell data. As a result, it will be inaccurate and even misleading to "guide" the transformation from the reference single-cell to the target single-cell if using the real bulk measurement to calculate the "loss" for the in silico inference. To overcome this potential issue, here we employed the above strategy to estimate pseudobulk data for the representative and the target sample and keep the model running on pseudobulk space to steer away from modeling the difference between bulk and pseudobulk expression of the samples.

In this in silico inference phase, to perform the bulk data alignment described above, we introduce an extra term to the loss function that accounts for the difference between the bulk data of the representative and that of the target sample. This fine-tuning process refines the model's capabilities, enabling it to perform in silico generation of cells from the target sample while maintaining consistency with the observed bulk expression patterns. Consequently, our deep generative model transitions from being solely focused on reconstruction to generating single-cell data for the target sample. In this crucial stage, we retire the discriminator from our model architecture, as our focus shifts from generating cells that resemble the representatives' cells to adapting the model for the target samples. Consequently, the training process exclusively involves the generator. The loss function for the generator adopted in this stage bears resemblance to that of the second pretrain phase, with the distinction of the bulk loss transitioning from $L_{BulkR}$ to $L_{BulkT}$:

$$L_{Inference} = L_{VAE} + \lambda_{BulkT} L_{BulkT} \quad (8)$$

$$L_{BulkT} = \left\| \frac{\sum_{i=1}^{N} \mathbf{x}'_i}{N} - \frac{\sum_{i=1}^{N} \mathbf{x_i}}{N} \cdot \boldsymbol{\alpha} \right\|_2 \quad (9)$$

$$\boldsymbol{\alpha} = (\mathbf{B_T} + \boldsymbol{\epsilon}) \oslash (\mathbf{B_R} + \boldsymbol{\epsilon}) \quad (10)$$

The second term of equation (9), $\frac{\sum_{i=1}^{N} \mathbf{x_i}}{N} \cdot \boldsymbol{\alpha}$, functions as an estimation of the pseudobulk value for the target sample, derived from the actual bulk data. This term is incorporated to reduce the systematic discrepancy between the pseudobulk and the actual bulk data corresponding to the same sample. To derive this estimation, we first compute the conversion ratio $\boldsymbol{\alpha}$ as the element-wise Hadamard division between the expression level of the representative's genes and the expression level of the target sample's genes using the real bulk RNA-seq data. Here, $\mathbf{B_T}$ and $\mathbf{B_R}$ represent the real bulk data of the target and representative samples, respectively, while $\boldsymbol{\epsilon}$, whose elements are all

set to a small fixed scalar value (0.1 by default), is introduced as a pseudocount vector to avoid division by zero. We expect that the ratio, which reflects the relative gene expression differences between the representative and target samples, will remain consistent whether it is derived from pseudobulk or true bulk expressions. This consistency underlines the robustness of our method in processing expression data across various bulk analyses. Following this, we compute the target sample's pseudobulk by multiplying the representative's pseudobulk from single-cell data (already known) with this conversion ratio $\alpha$, derived from the true bulk RNA-seq data. This process significantly boosts the robustness of our method, as the expression ratio between samples tends to remain stable, irrespective of the bulk sequencing technology and normalization methods applied. In contrast, the absolute differences between real bulk RNA-seq and pseudobulk data can vary based on their respective sequencing and normalization techniques, making it challenging to model them accurately. Our method's focus on relative rather than absolute differences allows for more consistent and reliable analyses across varied datasets. Supplementary Fig. S36 shows that in contrast to the real bulk data, the inferred pseudobulk from our method closely resembles the true pseudobulk observed from the profiled target single-cell sample, which suggests that our pseudobulk inference method can mitigate the technical difference between pseudobulk and real bulk and convert the real bulk into the corresponding pseudobulk that can better guide the in silico inference of the target single-cell sample. In cases such as the COVID-19 and colorectal cancer dataset, where the pseudobulk is already employed and no need to acquire it via estimation, the entire second term simplifies to the target's pseudobulk. Furthermore, as the bulk loss $L_{BulkT}$ inherently propagates more gradient to higher gene expression values (given the target bulk loss is in a squared error form), we implement a strategy to gradually reduce the gradient received by larger values. See details in the "Deep Generative Learning Model Training Details" section. Concurrently, we increase the weight of the bulk loss in the loss function over the course of training. This refined approach ensures that, after adequately training for larger values, the model can fine-tune smaller values, resulting in a comprehensive and precise single-cell data generation for the target samples. To counteract the influence of potentially noisy gene expression with minuscule values, we introduced a neuron activation threshold. This threshold converts output gene expression values that are exceedingly small, specifically those below 0.1% of the input library size (a threshold consistent with that used for background noise removal in preprocessing), to zero. In all of these stages, please note that none of the weights or subnetworks are frozen, except in the inference the discriminator network is no longer involved as mentioned in the paragraph describing inference. This is because we want the model to perform reconstruction and consider the reconstruction loss all the time so that the model does not deviate too aggressively from the representative's data distribution. Supplementary Fig. S50 shows the cells generated using models trained with different training stages, demonstrating the effectiveness and necessity of these training stages.

## Deep generative learning model architecture

In this study, our model integrates a Variational Autoencoder (VAE) as the generator and a Multi-Layer Perceptron (MLP) as the discriminator. We developed both components using `PyTorch`[100]. A concise overview of the model architecture is provided below, and for an in-depth understanding, we encourage consulting the complete details available in our GitHub repository (refer to the code availability section).

Before sending the cell data into the model, we performed some precomputation for the cells, which is part of a Graph Convolutional Network (GCN) layer. We aggregated information from the K (15 by default) nearest neighboring cells for each input cell. The neighbor finding for each cell was based on the 'neighbors.kneighbors_graph' tool in the Scikit-learn package. We preprocessed this message-passing step on the log1p transformed cell expression data, subsequently inputting the aggregated cell data into the deep learning model. The encoder of the VAE then employs linear layers and activation functions, leading to a hidden layer comprising 256 neurons. The message passing, the first linear layer, and the first activation function in the encoder form a GCN layer together. The design of the rest of the VAE structure, inclusive of both the encoder and decoder, was adapted from SCVI[93]. ReLU[101] activation is used for both the encoder and decoder. Weight decay[102,103] is used for avoiding overfitting. We chose 32 for the latent space dimension and 256 for the decoder's hidden layer size. Our model's VAE generator also utilizes a standard Gaussian prior for the latent space distribution, serving as a regularization mechanism to avert over-adaptation of the model parameters to the training data. For the discriminator, we constructed a 4-layer MLP, with Leaky ReLU activations[104] between linear layers. The hidden layers are configured with 256, 128, and 10 neurons, respectively. The output layer employs a Sigmoid activation to produce the prediction probability of the input cell being real. Weight initialization in the model utilizes the Kaiming method[105].

## Deep generative learning model training details

Following the detailed outline of the VAE-GAN model structure, we next describe the detailed training settings, which were designed to optimize the model's performance. The training process was structured in a three-stage sequence to ensure comprehensive learning and fine-tuning of the model parameters.

Initially, in the first pretraining stage (Pretrain 1), we focus on training the deep generative learning model to learn the distribution of the data and making it capable of reconstructing the representative single-cell data. This pretraining stage can further be split into two substages: in Pretrain 1.1, the VAE generator is trained independently until convergence (setting 100 epochs as a default value). Then, in Pretrain 1.2 the generator and discriminator are trained jointly until the generator cannot further reduce its loss. When trained together, the generator and the discriminator are trained alternatively for 3 epochs each, and this will go for 100 iterations by default. During this phase, we employed the default SCVI learning rate of $1 \times 10^{-3}$.

The second pretraining stage (Pretrain 2) mirrors the first in its basic approach but incorporates notable modifications. It is executed in full-batch mode, with an additional representative bulk loss integrated into the training process. Here, in Pretrain 2.1, similar to Pretrain 1.1, the generator is again trained until convergence (set to 50 epochs by default), and then trained jointly with the discriminator in Pretrain 2.2 until convergence. The default setting is training the two networks for 50 iterations, where each network is trained for 3 epochs in each iteration. To facilitate more refined adjustments during this stage, the learning rate is reduced to $1 \times 10^{-4}$.

In the final stage of training, the single-cell inference fine-tuning stage, the discriminator is set aside, and training is executed through a series of mini-stages. Each mini-stage involves setting the gradients for highly expressed genes above specific thresholds (each mini-stage corresponds to a different threshold) to zero, a strategy aimed at ensuring equitable optimization for genes with smaller expression values. These thresholds are hyperparameters estimated relative to the peak expression value (max_expr) from the representative's normalized count matrix, set at No threshold, $\frac{1}{2}$max_expr, $\frac{1}{4}$max_expr, $\frac{1}{8}$max_expr, and $\frac{1}{8\sqrt{2}}$max_expr (The last threshold should not be too close to 0 to avoid emphasizing too much on background noise). Training is continued until convergence in each mini-stage, typically not exceeding 150 epochs, with a learning rate of $2 \times 10^{-4}$. As the thresholds are lowered, some generated gene expressions are not involved in the loss computation, so the weight for the target bulk loss in the loss function is progressively quadrupled (i.e. increased by four times), ensuring a consistent total magnitude for this loss component. Upon completion of all mini-stages, the trained model is then utilized

to generate single-cell data for the target sample. The rationale behind this mini-stage strategy is to guide the model to equally tune the different levels of gene expression values it generates. When no threshold is applied since our bulk loss has the form of a square error, larger gradient values are passed to larger gene expression values. In other words, the training will focus too much on tuning the larger gene expression values and lack attention to the smaller ones. Therefore, we apply this mini-stage strategy to progressively fine-tune the model, encouraging it to adjust the generated gene expressions from the larger ones to the lower ones. Supplementary Fig. S50 shows that the cells generated using models trained after each mini-stage become progressively closer to the ground truth (target sample's real-profiled data), justifying the effectiveness of this mini-stage training strategy.

### Incrementally select representatives using active learning to improve single-cell inference (ACTIVE LEARNING)

Our strategy to experimentally select representative samples prioritized by active learning for sequencing is detailed as follows. Our approach to selecting representative samples for single-cell sequencing employs an iterative active learning process. Initially, an algorithm identifies a set of samples that provide a diverse and informative snapshot of the biological system under study, prioritizing them for the first round of sequencing (as described in the "The initial setup of *scSemiProfiler*" section in Methods). After sequencing, the gathered data is used to update and refine our semi-profiling model. If the accuracy of the resulting semi-profiles does not meet our standards (the inferred data is different from the sequenced data from the previous round), the active learning algorithm is re-engaged. It then selects samples as new representatives for subsequent rounds of sequencing based on their expected contribution to improving the model's accuracy. This iterative method ensures that the process does not exhaust the pool of informative samples in the initial round. Instead, it strategically selects a portion of samples at each step, preserving ample candidates for future sequencing. Our active learning design is meant to make efficient, incremental use of the sample pool, improving the model progressively with each round of sequencing.

Upon performing the initial round of representative selection and executing the deep generative learning models for in silico inference, every sample in the cohort possesses single-cell data, either real-profiled or semi-profiled. While researchers can opt to conclude here and proceed with downstream analysis using the single-cell data, when budget permits, our active learning module can further refine the process. By pinpointing additional informative representatives for single-cell sequencing, some target samples will be assigned a more similar representative, leading to enhanced single-cell inference performance.

Our initial idea of representative selection strategy is based on the belief that selecting the most "uncertain" samples as the representatives results in the most inference improvement and yields the richest insights for our model. Consequently, we select the sample with the largest bulk data difference compared to its representative to serve as the new representative:

$$R_{\text{new}} = \arg\max_{i \in \text{Cohort}} D_b(R(i), i) \tag{11}$$

Where $R(i)$ is the representative of $i$, and $D_b$ is the Euclidean distance between the PCA-reduced bulk data of two samples. This process is executed $B$ (representative batch size) times. Upon choosing new representatives, we update the membership for all samples. This ensures that, if a sample aligns more closely with a new representative than their previous one, their affiliation shifts:

$$R(i) = \arg\min_{r \in \text{Rs}} D_b(r, i) \tag{12}$$

where $R(i)$ is the representative of the sample $i$ and Rs is the set of representatives. Conceptually, this approach is similar to the uncertainty sampling[80] algorithm, a subtype of active learning. However, this method is passive learning since it does not utilize the knowledge acquired by the base learners, our deep generative learning models.

To improve the representative selection, we developed our active learning algorithm by merging the information gained by the deep generative model base learners into the selection of new representatives. To commence, we pinpoint the $B$ sample clusters exhibiting the highest in-cluster heterogeneity, which is the aggregate distance from each sample to the cluster's representative. The heterogeneity for a cluster $c$ is expressed as: $\sum_{i \in c} \left[ D_b(R_c, i) + \lambda_{sc} D_{sc}(R_c, i) + \lambda_{pb} D_{pb}(R_c, i) \right]$, where $i$ can be any sample in the cluster and $R_c$ is the representative. Essentially, a sample's heterogeneity is the combined distance-across three distance metrics-between the sample and its representative. The total cluster heterogeneity is derived from the summation of all sample heterogeneity. $D_b$ denotes the Euclidean distance between the PCA-reduced bulk data of two samples. The term $D_{sc}(R_c, i)$ represents the difficulty of transforming the representative single-cell data to that of a target. This is quantified by averaging the Euclidean distances between the representative cells and their K nearest neighbors in the target sample, all within a PCA-reduced space. Formally:

$$D_{sc}(a,b) = \frac{1}{N_a} \sum_{i=1}^{N_a} \sum_{j \in \text{KNN}(i)} \text{ED}(\mathbf{v_{a,i}}, \mathbf{v_{b,j}}) \tag{13}$$

with $a, b$ being two samples in the semi-profiled cohort, $v_a, v_b$ being PCA reduced single-cell data matrices for the two samples, $\text{ED}(x, y)$ computing the Euclidean distance between $x$ and $y$, and $\text{KNN}(i)$ identifying the cell's $K$ (1 by default) nearest neighbor in another sample's data matrix. Subscript represents rows, e.g. $\mathbf{v_{a,i}}$ is the $i$th row ($i$th cell) of the PCA reduced data of sample $a$. $D_{pb}$ is the Euclidean distance of the log1p transformed pseudobulk data of two samples, which is positively related to the amount of bulk loss and thus quantifies the deep learning model's difficulty in performing the in silico inference. Finally, $\lambda_{sc}$ and $\lambda_{pb}$ are empirical scaling factors (both are 1 by default).

The most heterogeneous cluster $C_H$ is chosen as:

$$C_H = \arg\max_{c \in \text{clusters}} \sum_{i \in c} [D_b(R_c, i) + \lambda_{sc} D_{sc}(R_c, i) + \lambda_{pb} D_{pb}(R_c, i)] \tag{14}$$

After selecting the $B$ most heterogeneous clusters to split, the subsequent step involves selecting a new representative for each of these $B$ clusters to minimize total in-cluster heterogeneity. Within each cluster, every non-representative sample is a potential new representative, each representing a different way of splitting the cluster. In the cluster, samples that are closer in bulk distance $D_b$ to the potential new representative than to the original representative can be reassigned to the new representative, splitting the cluster into two. The total heterogeneity of these two new clusters is then calculated based on the bulk distance. For each of the $B$ most heterogeneous clusters, we select the sample whose corresponding split results in the minimal total in-cluster heterogeneity to be the new representative $R_{new}$, as shown in the equation below.

$$R_{\text{new}} = \arg\min_{j \in C_H} \sum_{i \in C_H} \min(D_b(j, i), D_b(R_c, i)) \tag{15}$$

Where $R_c$ denotes the original representative of this cluster. Finally, the cluster membership updating will be executed to make sure each sample $i$ is assigned to the closest representative $R(i)$.

$$R(i) = \arg\min_{r \in \text{Rs}} D_b(r, i) \tag{16}$$

With the identification of new representatives, they are subjected to real single-cell data profiling and appended to the existing collection. In subsequent semi-profiling iterations, certain target samples will be assigned with more analogous representatives, enhancing in silico single-cell inference accuracy. This iterative procedure persists until either the budget is exhausted or a sufficient number of representatives are ascertained, ensuring optimal semi-profiling results.

## Semi-profiling pipeline stop criteria

For our semi-profiling pipeline (Supplementary Fig. S51) that iteratively selects representatives and performs in silico single-cell inference, we recommend two types of stop criteria for the users to choose according to their own needs. First, in the case that the user knows their budget and aims to achieve the best possible semi-profiling performance, they can run the pipeline and keep choosing representatives until they run out of budget. Second, in the case that the user aims to achieve an acceptable semi-profiling performance with the least amount of budget, the iterative cycle persists until an acceptable performance is reached. The performance is measured by comparing the representatives' actual single-cell data and the inferred data for them in previous iterations. Based on the user's needs, we recommend three levels of criteria. First, when the user is mainly studying the cell type proportion, we recommend stopping when the Pearson correlation between the ground truth and the estimated proportion reaches 0.85. Second, when the user's study includes more detailed analysis, such as biomarker discovery, we recommend stopping when the Pearson correlations between the biomarker expression patterns (dot colors and sizes in the dot plots) discovered in real-profiled data and those in semi-profiled data reach 0.85. Finally, for studies requiring more intricate single-cell level analysis, it is advised to halt when the Pearson correlation between data matrices, generated from advanced analysis tasks like trajectory inference and cell-cell interaction, for semi-profiled and real-profiled data, reaches 0.85. These guidelines are merely suggestions, and users have the flexibility to modify the stopping criteria to suit their specific requirements. The effectiveness of the semi-profiling process can be consistently evaluated by comparing the semi-profiled single-cell data (from the previous round) with the actual single-cell profiling (in the subsequent round). If the actual single-cell profiling aligns with the semi-profiling generated by our model from the previous round, this indicates a robust semi-profiling model has been effectively learned, and the iterative process can thus be concluded.

## Global mode representative selection

Multiple rounds of representative selection could be challenging for some labs and could potentially lead to batch effects. To accommodate users who prefer all-in-one-batch representative profiling, *scSemiProfiler* offers a "global mode". This mode enables users to assess data heterogeneity using Silhouette scores and bulk data visualization, helping to determine the approximate number of representative clusters required for their dataset. Once the number of clusters is established, users can select these representatives based on bulk data clustering and sequence all representatives in one batch to minimize potential batch effects. When using *scSemiProfiler* in real-world scenarios, users are advised to choose between the two modes—global and active learning-based on their specific needs. If budget conservation is paramount, then multiple-round selection using active learning is recommended. However, if the goal is to minimize batch effects and maximize convenience, then the global mode is more suitable. Even when the global mode is preferred, active learning remains crucial for evaluating the quality of in silico inference (semi-profiling). Unlike the active learning mode, where in silico inference from the previous round can be compared with actual single-cell profiling in the next round to assess semi-profiling accuracy, the accuracy of in silico inference in global mode cannot be directly

examined. To overcome this limitation, a portion of the samples selected in global mode (e.g., N = 28 split into 24 training + 4 test samples) can be used. This method involves using part of the samples (e.g., 24 samples) to train the model and performing in silico inference on the remaining samples (e.g., 4 samples). By comparing the in silico inferred single-cell data with the true single-cell measurements, the semi-profiling accuracy using N samples can be assessed. If the results are not satisfactory, the active learning module can then be used to select the next batch of representatives for improved semi-profiling in subsequent rounds. Therefore, a combination of the two modes can be employed in practical applications.

## Semi-profiling performance evaluation

**Cell type annotation for semi-profiled data:** Utilizing the annotated single-cell data from the representatives, we trained an MLP classifier available in the Scikit-learn package[86]. This classifier was then employed to categorize the cell types for cells in the semi-profiled data. In addition, beyond the supervised cell type annotation, *scSemiProfer* is capable of performing de novo cell type labeling for the generated cells according to the cell type marker expression using existing tools, such as Cellar[52], CellMarker[53,54], or Panglao DB[55].

**Dimensionality reduction:** In order to juxtapose the real-profiled and semi-profiled cohorts, we merged them and executed Principal Component Analysis (PCA) to reduce it to 100 principal components. This PCA-processed data serves as the foundation for computing single-cell level difference metrics. For visualization purposes, we first used SCANPY[106] to compute the neighbor graph using the PCA-reduced data, with the local neighborhood size parameter set to 50. Subsequently, the Uniform Manifold Approximation and Projection (UMAP) functionality within SCANPY was used for the visualization process.

**Semi-profiling error quantification:** To assess the accuracy of the semi-profiled cohort, we performed PCA to reduce its dimensionality, in conjunction with the real-profiled data, to 100 dimensions. For each sample within the semi-profiled dataset, we calculated a single-cell difference metric in comparison to the real-profiled single-cell data corresponding to that sample. This involves determining the Euclidean distance of each cell to its K-nearest neighbors in another dataset within the PCA-reduced space, followed by averaging these distances across all cells. The formula below outlines the difference computation between single-cell matrices $\mathbf{m_a}$ and $\mathbf{m_b}$.

$$\text{Dif}_{\text{sc}}(\mathbf{m_a}, \mathbf{m_b}) = \frac{1}{N_a + N_b}\left(\sum_{i=1}^{N_a}\sum_{j \in \text{KNN}(i)} \text{ED}(\mathbf{v_{a,i}}, \mathbf{v_{b,j}}) + \sum_{i=1}^{N_b}\sum_{j \in \text{KNN}(i)} \text{ED}(\mathbf{v_{b,i}}, \mathbf{v_{a,j}})\right) \tag{17}$$

Where $\text{Dif}_{\text{sc}}$ denotes single-cell level difference. $N_a$ and $N_b$ signify the cell counts in matrices $\mathbf{m_a}$ and $\mathbf{m_b}$ respectively. The $\mathbf{v}$s represents the PCA-reduced matrices. The terms $\mathbf{v_{a,i}}$, $\mathbf{v_{b,j}}$ represent the $i$th and $j$th cell (rows) in the PCA-reduced matrices of sample $a$ and $b$ respectively. The function KNN identifies the $K$ nearest neighbors within the alternate sample's PCA-reduced data matrix. We chose $K = 1$ for our experiments. ED computes the Euclidean distance between two vectors.

Referring to the comprehensive performance curves depicted in (Figs. 2f, 4f, 6f, the normalized error $E_N$ displayed on the left y-axis represents the mean single-cell difference between each sample's real-profiled version and semi-profiled version. This error is normalized using a theoretical upper bound (UB) and a lower bound (LB) of an expected single-cell difference between samples in the study.

$$E_N = \frac{1}{\text{UB} - \text{LB}}\left(\frac{1}{P}\sum_{i=1}^{P} \text{Dif}_{\text{sc}}(\mathbf{m_i}, \tilde{\mathbf{m}_i}) - \text{LB}\right) \tag{18}$$

and if $i$ itself is a representative, the single-cell difference equals the lower bound.

$$\text{Dif}_{sc}(\mathbf{m_i}, \tilde{\mathbf{m_i}}) = LB, \quad \text{if } i \in Rs$$

where $P$ stands for the total count of samples within the cohort and $\tilde{\mathbf{m_i}}$ refers to the semi-profiled version of $\mathbf{m_i}$. Rs denotes the set of representatives. The upper bound UB was derived by averaging the single-cell difference of randomly paired samples based on their PCA-reduced real-profiled data. This represents the performance of the worst possible representative selection, which is random selection. For computing the lower bound LB, we first divided the comprehensive PCA matrix, which contains all single-cell samples, into two random halves. Due to the randomness of the split, these two halves have the same data probabilistic distribution. Furthermore, given the large sample size, these halves are expected to have approximately the same actual cell distribution and thus can be regarded as two replicates of the same study. The difference between the replicates should be regarded as the lower bound, as it represents the best performance that an optimal selection can achieve[107].

**Deconvolution benchmarking:** We benchmarked the accuracy of our deconvolution method by comparing its performance against established methods: CIBERSORTx, Bisque, TAPE, Scaden, MuSiC, NNLS, and EPIC. We ran CIBERSORTx using the official web portal. Bisque, TAPE, MuSiC, and EPIC were installed according to the official GitHub repositories provided in the corresponding publications. We used the NNLS included in the MuSiC package. We used the PyTorch version of Scaden implemented in TAPE's GitHub repository for its testing. First, we annotated our cells generated by the deep learning model using the MLP classifier, which was trained on the representatives' annotated data, as discussed previously. Using this annotated data, we computed the cell type proportion. Subsequently, we employed the CCC, Pearson correlation coefficient (calculated using the Python package SciPy[108]) and RMSE to assess the consistency with the ground truth. However, when applying the aforementioned methods to the three datasets, we encountered both time and space complexity constraints for many benchmarked methods. Specifically, CIBERSORTx and Bisque were unable to utilize as many representative samples for single-cell references as our approach. Scaden and TAPE always only sample 5000 cells for training their models, thereby also failing to fully exploit all available representatives. To ensure a fair comparison, we also offered an alternative version of our results for these datasets. In this version, the single-cell reference is confined to just the first 4 representatives, aligning with the capacity of all other benchmarked methods. To statistically ascertain if one method notably surpasses another in performance, we calculated the p-values using a one-sided Wilcoxon test[77], as implemented in SciPy. This test enables us to determine whether the metric of one method is significantly higher or lower compared to that of another.

**Gene set activation scores computation and heatmap plotting:** We computed the interferon pathway activation scores following the same procedure in the study[45] from which we acquired the COVID-19 dataset. To compute the activation scores depicted in the heatmaps (refer to Fig. 2c), we first computed the average expression for each cell type and severity combination using the code from the COVID-19 dataset provider, which uses the 'tl.score_genes' tool in SCANPY. Then, for each cell type, we computed the activation score of each severity level as the fold changes from the healthy condition to it. For the "activation of immune response" activation pattern in the colorectal cancer dataset depicted in Fig. 4c, we first collected the genes of this GO term (GO:0002253) and used the same SCANPY tool to compute the average expressions for each cell type and tissue combination. For each cell type, we calculated the activation scores by determining the fold changes in comparison to the "Normal" tissue type across all other tissue types. For the "activation of immune response" activation

pattern in the iMGL dataset depicted in Fig. 6c, we first computed the average gene expression for each cell type and tissue combination similarly. The activation score was determined by calculating the fold change for each cell type, using "C1: Homeostatic non-proliferative" as the reference background. Considering the relatively smaller size of the colorectal cancer and iMGL dataset and the fact that some entries have very few cells, the values were only computed for entries with more than 500 cells for the colorectal cancer dataset and entries with more than 100 cells for the iMGL dataset.

**Biomarker discovery and enrichment analysis:** For making the RRHO plots, we used SCANPY to identify each cell type's top 50 positive cell type markers and top 50 negative cell type markers for the real-profiled and semi-profiled datasets. RRHO plots are used for visualizing the overlap between two ranked gene lists. Each entry in the plot corresponds to a negative logged p-value of the hypergeometric test between two marker lists. The bottom left quadrant corresponds to the comparison between two positive marker lists. In this quadrant, the plot entry in the $i$-th row from bottom to top and $j$-th column from left to right corresponds to the comparison between the top $i$ genes in the real-profiled version top negative markers and top $j$ gene in the semi-profiled version top negative markers. Other quadrants are plotted similarly, with the marker lists always starting from the respective corners of the plot. Strict p-values are adopted in RRHO analysis, focusing on overlaps greater than K (the number of overlap), which can help identify points of particularly high significance that stand out more starkly against the background, providing a clear indication of where the most robust patterns lie, albeit potentially at the cost of missing smaller yet significant patterns.

For the GO and Reactome enrichment analysis, we identified the top 100 cell type signature genes for both real-profiled and semi-profiled datasets using SCANPY. The identified markers were subsequently employed for enrichment analysis with the Python package GSEApy[109]. For enrichment analysis visualization in Fig. 3c, Fig. 5c, Fig. 7c, we employed the adjusted p-value from GSEApy's output to indicate significance.

**Cell-cell interaction analysis:** We conducted cell-cell interaction analyses on both real-profiled and semi-profiled datasets using the R package CellChat[28]. For the COVID-19 dataset, we analyzed interactions of cells from patients with each disease severity. In the colorectal cancer dataset, our focus was on interactions of cells originating from different tissues, and in the iMGL dataset, we evaluated cells across various condition groups.

**Pseudotime analysis:** We carried out pseudotime analysis utilizing the Monocle3 R package[110]. The UMAP coordinates, previously generated in our UMAP comparison between real-profiled and semi-profiled datasets, served as the input for pseudotime computation. To ensure a fair comparison between the real-profiled and semi-profiled data, we consistently selected the roots at the same positions within the UMAP space. To compare the similarity between the pseudotime values computed for the two datasets, taking UMAP coordinates into consideration, we split the UMAP plot (the area within max/min x/y coordinates) into 20 by 20 grids, compute the average pseudotime value in each grid, and finally compute the Pearson correlation between the values from the two datasets.

**Cell trajectory analysis using partition-based graph abstraction (PAGA):** The PAGA plots were generated using SCANPY. First, PCA was applied independently to each dataset to reduce them to 100 principal components. Subsequently, the PCA-reduced data was utilized to compute the neighbor graphs, with the size of the local neighborhood set to 50, aligning with our UMAP settings. Based on the neighbor graphs, we generated the PAGA plots using SCANPY.

## Statistics & reproducibility

The study does not involve biological experiments. No statistical method was used to predetermine sample size. No data were excluded

from the analyses. The experiments were not randomized. The Investigators were not blinded to allocation during experiments and outcome assessment.

## Reporting summary

Further information on research design is available in the Nature Portfolio Reporting Summary linked to this article.

## Data availability

Three datasets are used for evaluating our method. The preprocessed COVID-19 dataset is from Stephenson et al.'s study[45] and can be downloaded from Array Express under accession number E-MTAB-10026 [https://www.ebi.ac.uk/biostudies/arrayexpress/studies/E-MTAB-10026]. The colorectal cancer dataset is from Joanito et al.'s study[67]. The count expression matrices are available through Synapse under the accession codes syn26844071 [https://www.synapse.org/#!Synapse:syn26844071/wiki/615389]. The iMGL dataset is from Ramaswami et al.'s study[72]. The raw count iMGL bulk and single-cell data can be downloaded from the Gene Expression Omnibus (GEO) repository under accession number GSE226081 [https://www.ncbi.nlm.nih.gov/geo/query/acc.cgi?acc=GSE226081]. The hamster bulk and single-cell data can be downloaded from the GEO repository under accession number GSE200596 [https://www.ncbi.nlm.nih.gov/geo/query/acc.cgi?acc=GSE200596]. Source data are provided with this paper.

## Code availability

Code for the models and results reproduction is publically available on GitHub: https://github.com/mcgilldinglab/scSemiProfiler [111].

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

## Acknowledgements

This work was funded in part by grants awarded to [JD]. We gratefully acknowledge the support from the Canadian Institutes of Health Research (CIHR) under Grant Nos. PJT-180505; the Funds de recherche du Québec - Santé (FRQS) under Grant Nos. 295298 and 295299; the Natural Sciences and Engineering Research Council of Canada (NSERC) under Grant No. RGPIN2022-04399; and the Meakins-Christie Chair in Respiratory Research. We also extend our gratitude to Yanshuo Chen, the lead author of TAPE, for his indispensable support in the deconvolution benchmarking experiments. This work is part of the HCA publication bundle (HCA-8).

## Author contributions

J.D. conceived and designed the study, developed the methodology, and planned the experiments. J.W. implemented the methodology and conducted the experiments. Both J.D. and J.W., along with G.J.F., participated in data collection and analysis. All authors (J.D., J.W., and G.J.F.) contributed to writing and revising the manuscript. Each author has read and approved the final manuscript for publication.

## Competing interests

The authors declare no competing interests.
