## [Peer Review File · Nature Communications]

scSemiProfiler: Advancing Large-scale Single-cell Studies through Semi-profiling with Deep Generative Models and Active LearningReviewer #1 (Remarks to the Author):

This deep learning approach develops a "semi-profiled" single-cell seq dataset that contains both real single-cell data for representative samples of (real or pseudo) bulk sequencing and inferred single-cell data for the non-representative samples, which compares significantly well with a real dataset. Thus, the goal of this project is to offer a cost-effective alternative for single cell profiling.

A few comments for the authors:

- * Missing latest reference on CIBERSORTx (<https://pubmed.ncbi.nlm.nih.gov/31960376/>).
- * MuSiC, semi-related work, utilizes cross-subject scRNA-seq to estimate cell type proportions in bulk RNA-seq. How do these methods compare (the former starts from sc, the latter from bulk, what else?) (<https://www.ncbi.nlm.nih.gov/pmc/articles/PMC6342984/>)?
- * The methods compared to scSemiProfiler later in the manuscript (Scanded, TAPE, DWLS) should be discussed in the related work paragraph, together with the CIBERSORTx and Bisque.
- * Missing references on the related work paragraph in page 2. E.g.,
 1. "Conventional bulk decomposition methods, while valuable for enhancing cohort study analyses, exhibit limitations in their resolution and accuracy" [A reference would be appreciated here to justify the resolution and accuracy limitations. If the authors believe this is justified by the analysis presented in the manuscript, it should be noted].
 2. Similar comment as above for "Yet, these methods fall short in delivering true single-cell resolution. Another significant challenge they face is the precise decomposition of cell types and accurate inference of gene expression. Additionally, these methods often overlook the considerable variability that exists within individual cell types." Do the authors believe that these arguments are further supported by the analysis they present later in the manuscript? If so, it should be clearly noted.
- * Page 4, "After quality controls,": the authors should explain the quality control performed on the COVID-19 cohort sc-seq dataset.
- * Page 13, can the authors explain in more detail what is the reason they had to infer pseudobulk from real RNA bulk data?
- * Page 13, same as above, can the authors explain in more detail why the pseudobulk data space "often aligns more closely with single-cell data"?
- * Page 15, "This is particularly remarkable given the unique challenges presented by the real bulk data." Can the authors explain these challenges in relation to the deep learning method developed?
- * The alignment of the directly sequenced and semi-profiled samples is indeed remarkable. It would be appreciated if the authors explain how overfitting has been avoided.^[1]_[SEP]

Reviewer #1 (Remarks on code availability):

1. Installation of semiprofiler was seamless.
2. The <https://scsemiprofiler.readthedocs.io/en/latest/environment.txt> link is invalid
3. The README file had enough information.
4. I am not familiar with Jupyter notebook so I was not able to run the code (although I did have Jupyter notebook installed).

Reviewer #2 (Remarks to the Author):

I co-reviewed this manuscript with one of the reviewers who provided the listed reports

Reviewer #3 (Remarks to the Author):

This paper describes a method for deconvolution of bulk RAN-seq data that is capable not only of extracting percentages of cell types (as usual), but also of generating data at the single cell level. To the best of my knowledge, this is a novel feature. As with other deconvolution algorithms, it requires training data consisting of bulk samples and corresponding scRNA-seq, but an important

point (not emphasized enough by the authors) is that it appears to work without the need for previously labeled training data. This means that one could theoretically generate single cell data from bulk data and work on de novo labeling of cell types.

The algorithmic approach to achieve the goal, although based on SCVI, seems appropriate and adds value to the study on a methodological level; however, the methods section should be improved in terms of structure and level of detail in the descriptions. The code is available to the scientific community via github.

Despite all the positives, the work needs to be improved in two main areas: a) to further show that the method is able to generate profiles of cells other than those used in training; b) to demonstrate that, after the proposed active learning process, the single cell data generated from the samples with only bulk data add relevant and necessary information to reach better results or conclusions than if only the real single cell data of the selected samples are used.

General:

- Can we gain more insight or extract new or different information by adding the semi-profiled cells of samples without sc data together with the real sc data obtained for the selected representatives? The number of samples is important when performing sample-based statistics, such as comparing percentages of cell types, but in this scenario traditional deconvolution methods are sufficient and single-cell resolution is not necessary. In downstream analyses with the three datasets, where is it really important to generate more single cell data within cohorts? This needs to be discussed, also commenting on the inherent increase in data size for downstream sc analysis when adding semi-profiled data, and the obvious increase in resource requirements for processing and management. This is not shown in this paper.
- What is the significance of the gene sets (scores) in this framework? What is the percentage of final features that are genes and not gene sets? A comparison between using only gene expression (without gene sets), using only gene sets, and using both is needed to understand these points.
- The difference between Fig. 2c and 2d (and equivalent in other figures) is not clear. 2c shows only data from representatives? What is shown is better understood in Supp Fig 2...
- Plot umaps colored by real/semi-profiled (Fig. 2c), but separately for each of the B clusters (of samples, with 1 representative each). Also, umap plots colored or split by sample should be included in the supplement to check if the segregation of cell types depends on the sample/sample_cluster and if there are always some cells from representatives.
- For evaluation and benchmarking, it is recommended to report CCC (Concordance Correlation Coefficient) as it is a more appropriate metric than RMSE or Pearson correlation alone.
- Were the geneset scores concatenated with gene expression data also used to calculate losses? Are they treated exactly the same as the genes (like "extra genes")?
- Clarify if the training (or part of the training) is independent for each cluster of samples (representative and associated target samples). What about fine tuning?
- Do plots and results comparing real-profiled vs. semi-profiled represent all available samples? Only representatives for real-profiled and only non-representatives for semi-profiled? Clarification is important in any case. Results should be reported (at least in the Supp) separately for representatives (sc data used for training) and non-representatives (only (pseudo) bulk data was used) in order to assess proper evaluation. Also, in plots where different samples are represented, the ones that were used as representatives should be indicated. For example Fig.S3
- The results of the downstream analyses are compared between real-profiled and semi-profiled sc data. All samples in the cohort were used (original sc data and all semi-profiled from the corresponding (pseudo) bulk data)? In any case, this comparison shows that semi-profiled cells are close to real sc data. But it would be very relevant to compare all these sc-based results between: a) using only real sc data from the representatives; b) using sc from the representatives together with the all sc semi-profiled data. In this way, we can mimic a real scenario where there is a limited budget, so we cannot run as many sc experiments as we would like, and we decided to use scSemiProfiler to fill the gap. Are there more insights when adding the semi-profiled cells to the study? Also comment on the increase in data for downstream sc analysis when adding semi-profiled data, and the obvious increase in resource requirements for processing and management. This is linked to first point.

- PAGA analysis in Fig. S8 (and S16, and 7d). In order to properly use graph plots for visual inspection to check the similarity of output graphs, clusters for real and semi-profiled datasets should be placed in the same locations.
- Fig. S10 (more than S18), especially b), which shows examples of representative real, target real, and target semi-profiled, may be placed somewhere in the main text and figures due to its importance. It shows that semi-profiling is not a "noisy replication of representatives", but is able to correctly generate cells that are similar but different from the corresponding representative, based on the differences between the two (pseudo) masses. This figure could be completed and improved by adding the semi-profiled results after the different training stages (and not only the last one), showing why the following stages improve the results.
- In Figure 8, the authors show the difference in performance when using an active or a passive learning strategy, through an iterative process to gradually increase the number of representatives. But what if the selection of N representatives was done by Kmeans from the beginning (independent of the results of scSemiProfiler and based only on the differences in the bulk data)? Are these results still worse than the active iterative selection, which has to quantify the differences in bulkPCA(D_b) + bulkGenes(D_pb) + scPCA(D_sc)? Discuss this
- The iMGL dataset is the only one with real bulk data paired with sc data. So results there are more informative and it should be emphasized in the text.
- It is not clear what was used for training and inference in the iMGL dataset. There are real bulk and real single cell data available, but it seems that the (unnecessary) pseudobulks obtained from single cell data were used during the process. A better description is needed, and a discussion of the reasons for doing this instead of using only real data.
- How similar are the expression distributions between real bulk data and pseudobulks generated from sc data? This can be quantified in the iMGL dataset.
- Authors can use other approaches to generate pseudobulks, and check if the distributions are closer to the real ones? Will the results of scSemiProfiler be better on real samples if better pseudobulks are used?
- Fig. S18 for the iMGL dataset would be more interesting if it showed a clear target real-profiled sample that differs more from the representative real-profiled data, as in Fig. S10. In this way, the authors can show that in real data, scSemiProfiler is able to exploit the differences between bulk samples to generate correct cells that are different from those learned from the corresponding representative.

METHODS:

- Structure of the section must be improved to facilitate comprehension.
- The initial setup of scSemiProfiler. Selection of the number of representatives using Kmeans to perform the clustering. How many PCs were used to do it?
- Representative single cell profiling and processing. Number of highly variable genes and method to select them must be specified as this is a critical starting point. Numbers for "high proportion of mitochondrial reads" are missing. Same for the method to find doublets.
- Fine-tune the deep generative learning model to infer the single cell. Are some weights or subnetworks frozen for pretrain2 with respect to pretrain1? And what about inference (fine-tuning for bulk data)?
- In cases such as the COVID-19 and colorectal cancer datasets, where real bulk data is not available, the entire second term simplifies to the pseudobulk of the target. It should not be like this, it should use the target pseudobulk as BT to work in a realistic way. Explain better or discuss
- Deep generative learning model architecture. More details are needed. For example: type of activation function for encoder and decoder (ReLU, I think), minibatch size in each training stage, etc. Reference to SCVI is missing.
- Deep Generative Learning Model Training Details. Pretrain1 is described as 2-stage training, first only the AE (generator), and later generator+discriminator. Is this correct? In this case it should be Pretrain1.1, Pretrain1.2 or similar. Same for Pretrain2
- The explanation of ministage training in fine-tuning for inference is confusing and incomplete. Each ministage corresponds to a threshold option? How are they ordered? Why is the last threshold not 1/16? What exactly does "target bulk loss in loss function is progressively quadrupled" mean, something to do with lambda_BulkT? In general, why minibatches are needed or recommended; what is the point of this strategy?

- Incrementally select representatives using active learning to improve single-cell inference. Missing values (weights) for λ_{sc} and λ_{pb} .
- Semi-profiling performance evaluation. "Referring to the comprehensive performance curves shown in (Fig.2d, Fig.4d, Fig.6d, the normalized error..." should be Fig.2f, Fig.4f, Fig.6f.

Reviewer #4 (Remarks to the Author):

This manuscript introduced scSemiProfiler, a novel computational framework that integrated deep generative model with active learning strategy to infer single-cell profiles across large cohorts. The authors carefully evaluated its performance and robustness using three distinct datasets, obtaining consistent results compared to real-profiled single-cell RNA sequencing data, validated by UMAP visualization and downstream analyses. Through these comprehensive evaluations, the authors demonstrated the method's capability to select representative samples for authentic single-cell sequencing, using an active learning module and successfully inferred the remaining samples' single-cell data using a deep generative learning model. Overall, this work is interesting and presents a potentially useful method in facilitating single-cell profiling. I do not have any major concerns, and the manuscript should undergo a minor but essential revision prior to acceptance.

1. In the overview of their method, the authors outlined the scSemiProfiler pipeline. It looks to me that multiple rounds of single-cell sequencing might be needed to obtain the final perfect results. However, such a procedure might bring a heavy burden to typical labs which budget is always limited, and in multiple rounds of sequencing, batch effects will occur and be difficult to clear. Also, the authors should describe how to experimentally select representative samples prioritized by active learning for sequencing. In my thinking, these representative cells have been used up for the first-round sequencing. Additionally, if only the deep generative module was used, the users would lose the superiority of active learning module. The authors should carefully address these concerns.

2. Although the authors successfully validated the performance of scSemiProfiler on three distinct cohorts, showing a considerable consistency between semi-profiled and real-profiled data, I suggest that the authors must perform additional comparisons using more cohorts of different species or tissues to enhance the reliability and transparency of the algorithm.

3. From the results, it can be found that the performance of scSemiProfiler is not markedly higher than Bisque on the data set of iMGL, compared to other two data sets. Thus, I am curious that whether the results might be attributed to the process of pseudo-bulk data generation. The authors should describe that whether the method of averaging normalized count single-cell data to generate bulk data is a gold-standard method. Furthermore, the authors should clarify that whether only the characteristics of pseudo-bulk data generation from single-cell data has been learnt, in the training step using the COVID-19 and Colorectal Cancer datasets learns. The authors should include more cohorts with real bulk sequencing data and paired single-cell sequencing data for further validation. Considering the data limitation, the authors can also choose to analyze the pseudo-bulk data of iMGL, and then compare it with the real bulk data.

4. Although the authors compared the performance of scSemiProfiler with CIBERSORTx, Bisque, TAPE, and Scaden, it would be valuable to include more tools for comparisons on deconvolution of bulk data to single-cell data analysis, such as MuSiC, EPIC, and so on. Additionally, analyzing the profiles generated by other tools would highlight the necessity and superiority of scSemiProfiler.

5. Existing tools such as CIBERSORTx offer web and software versions for accessibility, making it easier to use for scientists. The authors only provide a Python package for applications. Is there any possibility that the authors can develop a user-friendly web or software interface to facilitate broader use for biologists?

6. Although annotations were provided for each color in Figures 2g, 4g, and 6g, the authors should

provide full annotations of their clustering results in Figures 2a, 4a, and 6a.

Taken together, this study provided an interesting method, which might be useful for facilitating single-cell profiling and data analysis.

Reviewer #4 (Remarks on code availability):

If possible, an online service should be implemented for a convenient use of their method.

Dear ,

We would like to express our gratitude to the reviewers and editorial team for the constructive comments and suggestions offered on our manuscript, titled "**scSemiProfiler: Advancing Large-scale Single-cell Studies through Semi-profiling with Deep Generative Models and Active Learning**," with manuscript ID NCOMMS-23-58261-T. The detailed feedback has substantially improved the manuscript's quality and thoroughness. We also thank the editor for their expedient and careful handling of our submission. Following the guidance provided, we have addressed each comment in detail. Enclosed are our point-by-point responses to the issues raised and the amendments applied to the manuscript (all changes were tracked in blue).

Reviewer #1 (Remarks to the Author):

This deep learning approach develops a “semi-profiled” single-cell seq dataset that contains both real single-cell data for representative samples of (real or pseudo) bulk sequencing and inferred single-cell data for the non-representative samples, which compares significantly well with a real dataset. Thus, the goal of this project is to offer a cost-effective alternative for single cell profiling.

Response: We sincerely appreciate your thoughtful review and accurate understanding of our tool's contribution. Your recognition of the cost-effective alternative our tool offers for single-cell profiling is particularly encouraging. We are also grateful for your acknowledgment of how well the inferred single-cell data for non-representative samples resembles actual single-cell data.

A few comments for the authors:

Comment 1: * Missing latest reference on CIBERSORTx (<https://pubmed.ncbi.nlm.nih.gov/31960376/>).

Response: We appreciate the reviewer highlighting the absence of the latest CIBERSORTx reference. We have now added this reference (Steen, Liu et al. 2020) to the introduction section of our manuscript that discusses the CIBERSORTx tool.

Comment 2: * MuSiC, semi-related work, utilizes cross-subject scRNA-seq to estimate cell type proportions in bulk RNA-seq. How do these methods compare (the former starts from sc, the latter from bulk, what else?) (<https://www.ncbi.nlm.nih.gov/pmc/articles/PMC6342984/>)?

Response: Thank you for raising an issue that was insufficiently detailed in our initial submission. We recognize the importance of a detailed discussion on how other deconvolution methods compare with *scSemiProfiler*. To address this, we have enriched the manuscript with a more comprehensive comparison, clearly outlining both the similarities and differences. This enhancement will assist readers in understanding the distinct scenarios each tool addresses, as

well as the unique features of *scSemiProfiler*. These comparisons have been integrated into the introduction, benchmarking results, and discussion sections of the manuscript for better clarity.

MuSiC and many other deconvolution tools are similar to *scSemiProfiler* in the way that they all take scRNA-seq reference as input and can output cell type proportions, which are crucial for many biological studies, of the target bulk RNA-seq samples. Nevertheless, *scSemiProfiler* is different in the following ways: First and foremost, *scSemiProfiler* provides finer-grained output compared to MuSiC and other deconvolution methods. *scSemiProfiler* does not only output cell type proportion but also outputs single-cell level gene expression data of each target sample, which enables single-cell level analysis that cannot be performed using outputs from other deconvolution methods. Secondly, *scSemiProfiler* has components that are designed for investigating large cohorts and these features are not owned by MuSiC and other deconvolution methods: (1) As mentioned by the reviewer, our method starts from the bulk data of the cohort to gain information about sample similarities and the mapping between representatives and target samples. MuSiC and other deconvolution methods do not have this step and do not well leverage this sample similarity information. (2) Various kinds of sample similarity information gained during the semi-profiling process inform the representative selection using active learning, which is not considered by other deconvolution tools. Representative selection (acquiring single-cell reference) is beyond the scope of the traditional deconvolution task. That said, a sub-routine of *scSemiProfiler* – *in silico* inference (Fig.1 c) is more similar to MuSiC and can be applied to any scenario that MuSiC can be applied, except the output is single-cell level. Lastly, *scSemiProfiler* has significantly better deconvolution performance than other deconvolution tools including MuSiC, which now has been added to our benchmarking in the revision (see Fig.2 j,k,l, Fig.4 j,k,l, Fig.6 j,k,l, and Supplementary Fig. S43 f,h,i). Utilizing the sample similarity information in the cohort, the capability of learning and transformation data distribution of our deep generative model, and the power of active learning, *scSemiProfiler* shows superior performance and unique functionalities (i.e., true single-cell level decomposition) over other methods including MuSiC. Remarkably, *scSemiProfiler* performs deconvolution accurately on real bulk datasets (iMGL and hamster) while most other deconvolution tools' performance, especially MuSiC's, drops dramatically. It has been previously reported that MuSiC struggles with real bulk data, often yielding negative values for Lin's concordance correlation coefficient (CCC) (Lawrence and Lin 1989) as noted in recent studies (Menden, Marouf et al. 2020, Chen, Wang et al. 2022). In contrast, comparative analysis using pseudobulk and real bulk data from the iMGL dataset indicates that *scSemiProfiler* exhibits very similar performance on both pseudobulk and real bulk data (Reviewer #4 minor comment 3).

Comment 3: * The methods compared to *scSemiProfiler* later in the manuscript (Scanded, TAPE, DWLS) should be discussed in the related work paragraph, together with the CIBERSORTx and Bisque.

Response: We thank the reviewer for the comment. Acknowledging the importance of providing a comprehensive background, we have expanded the introduction section to include detailed discussions of all methods, including Scaden, TAPE, and DWLS, against which

scSemiProfiler is benchmarked. This enhancement aims to provide readers with a clearer understanding of the research context and the specific challenges our tool is designed to address.

Comment 4: * Missing references on the related work paragraph in page 2. E.g.,

1. “Conventional bulk decomposition methods, while valuable for enhancing cohort study analyses, exhibit limitations in their resolution and accuracy” [A reference would be appreciated here to justify the resolution and accuracy limitations. If the authors believe this is justified by the analysis presented in the manuscript, it should be noted].

2, Similar comment as above for “Yet, these methods fall short in delivering true single-cell resolution. Another significant challenge they face is the precise decomposition of cell types and accurate inference of gene expression. Additionally, these methods often overlook the considerable variability that exists within individual cell types.” Do the authors believe that these arguments are further supported by the analysis they present later in the manuscript? If so, it should be clearly noted.

Response: Thank you for your suggestion to bolster our statements about the limitations in resolution and accuracy of conventional bulk decomposition methods with relevant references. We have now included references (Maden, Kwon et al. 2023, Momeni, Ghorbian et al. 2023) in the Introduction section. These sources elaborate on various technical and biological factors that contribute to the limitations in the accuracy and reliability of conventional deconvolution methods.

Furthermore, this assertion is further reinforced by our analysis. The benchmarking of deconvolution across the four datasets in our study illustrates that most benchmarked methods do not achieve high accuracy and perform substantially worse than *scSemiProfiler*, especially when analyzing real bulk data. Some methods even yield predictions with negative CCC values compared to the ground truth. The limitations in resolution are also underscored by the inability of conventional deconvolution results to support single-cell level analysis and visualization, including UMAP, pathway activation patterns, biomarker discovery, and gene functional enrichment analysis (specifically for those samples or conditions not profiled at the single-cell level), as well as cell-cell interaction analysis and pseudotime analysis. In response to your suggestion, we have added this explanation to the introduction section of the revised manuscript.

Regarding the second point, yes, these arguments are indeed supported by our analysis results. In response to your feedback, we made the following modifications to the discussion section of our manuscript: Explicitly mentioning that conventional deconvolution does not provide single-cell resolution data and thus cannot perform all the single-cell level tasks that our generated data can perform. Additionally, the cell type gene expression signature provided by conventional deconvolution methods represents the gene expression at cell type level and thus cannot be used for investigating the heterogeneity within each individual cell type, e.g. visualization of the gene expression of individual cells of the cell type or pseudotime analysis. We believe that such modifications can help the reader better understand the extra power of *scSemiProfiler* over conventional deconvolution tools.

Comment 5: * Page 4, “After quality controls,”: the authors should explain the quality control performed on the COVID-19 cohort sc-seq dataset.

Response: Thanks for pointing this potential lack of clarity, we have now clarified the specific quality control we performed on the COVID cohort single-cell dataset. Specifically, it's important to note that the COVID-19 dataset, along with all other datasets included in our study, underwent quality control processes using the same exact standardized pipeline. This data preprocessing (including quality controls) pipeline is thoroughly described in the section "Representative single-cell profiling and processing" of our Methods. Briefly, background noise of the single-cell data is removed, low quality cells and gene are filtered using standard Scanpy 1.9 (Wolf, Angerer and Theis 2018) preprocessing pipeline. Then, library size is normalized, data is log_{1p} transformed, highly variable genes are selected.

Comment 6: * Page 13, can the authors explain in more detail what is the reason they had to infer pseudobulk from real RNA bulk data?

Response: We are grateful to the reviewer for highlighting the absence of a detailed explanation for a key technique employed in our study, which was pivotal in enhancing the performance of single-cell semi-profiling using real bulk RNA-seq data. A comprehensive description of this technique is provided below.

Firstly, our methodology is intricately designed around the *in silico* inference process, which involves reconstructing a representative sample. The approach of inferring pseudobulk from real bulk as employed in our method aligns well with the mechanism of our *in silico* inference process. The inference is achieved by performing reconstruction of the representative single-cell sample and then “transforming” it to match the pseudobulk (the average expression) of aggregated cells from the target sample. Because of the difference in sequencing platform and technology, as well as the relatively higher level of dropouts and noise, the pseudobulk of the target single-cell sample is often quite different from the bulk sequencing from the same sample (Hou, Ji et al. 2020), due to the “dropouts” (gene not detected) and noise (Kharchenko, Silberstein and Scadden 2014, Hicks, Townes et al. 2018) in single-cell data. As a result, it will be inaccurate and even misleading to “guide” the transformation from the reference single-cell to the target single-cell if directly using the real bulk measurement to calculate the “loss” for the *in silico* inference, since the pseudobulk of the target single-cell sample is different from its true bulk counterpart (Supplementary Fig. S36). In contrast, the inferred pseudobulk from our method closely resembles the true pseudobulk that we observed from the profiled target single-cell sample, which suggests that our pseudobulk inference method can mitigate the technical difference between pseudobulk and real bulk and convert the real bulk into the corresponding pseudobulk that can better guide the *in silico* inference of the target single-cell sample.

Secondly, our approach of inferring pseudobulk from real bulk also provides robustness to our model when dealing with the technical difference between pseudobulk and real bulk, and the biases introduced by different real bulk sequencing technologies. To infer the target sample's pseudobulk, we first calculate the conversion ratio $\alpha = \frac{(B_T + \epsilon)}{(B_R + \epsilon)}$, where B_T is target real bulk, B_R is representative real bulk, ϵ is pseudocount to prevent zero division error, between the expression level of the representative's genes and the expression level of the target sample's genes using the real bulk RNA-seq data. We expect that the ratio, which reflects the relative gene expression differences between the representative and target samples, will remain consistent whether it is derived from pseudobulk or true bulk expressions. This consistency underlines the robustness of our method in processing expression data across various bulk analyses. Following this, we infer the target sample's pseudobulk by multiplying the representative's pseudobulk from single-cell data (already known) with this conversion ratio α , derived from the true bulk RNA-seq data. This process significantly boosts the robustness of our method, as the expression ratio of pseudobulks between samples tends to remain stable, irrespective of the bulk sequencing technology and normalization methods applied. In contrast, the absolute differences between real bulk RNA-seq and pseudobulk data can vary based on their respective sequencing and normalization techniques, making it challenging to model them accurately.

We have added the above explanation into the "Fine-tune the deep generative learning model to infer the single-cell measurements for the target samples" section in the Methods part of our manuscript, and we believe this can significantly help the reader understand our techniques used for dealing with real bulk data and their robustness, which is a major challenge for other deconvolution methods.

Supplementary Fig. S36: Comparative analysis of similarity of inferred pseudobulk and real bulk against ground truth pseudobulk. This figure contrasts the ground truth pseudobulk data against both real bulk data and inferred pseudobulk data using our strategy across two datasets featuring paired single-cell and real bulk data. The results illustrate a significantly higher similarity of our inferred pseudobulk data (based on the real bulk) to the ground truth pseudobulk (of the target single-cell data) compared to the real bulk data (direct usage), as confirmed by p-values derived from Wilcoxon tests. **a**, iMGL cohort results comparison between inferred pseudobulk and real bulk in terms of their concordance with ground truth pseudobulk. **b**, Hamster cohort results comparison between inferred pseudobulk and real bulk in terms of their concordance with ground truth pseudobulk.

Comment 7: * Page 13, same as above, can the authors explain in more detail why the pseudobulk data space “often aligns more closely with single-cell data”?

Response: Certainly. This topic is addressed in our response to Reviewer #1 comment 6, and is further elaborated in the revised manuscript. In brief, pseudobulk data is generated by averaging the expression of cells within a single-cell data, making it inherently more aligned with single-cell data characteristics as they come from the same data source. Conversely, real bulk data may differ from average single-cell expression, as single-cell sequencing technologies typically exhibit higher noise levels and more technical dropouts compared to bulk sequencing technologies.

To reinforce the above point, we further demonstrate the advantages of using pseudobulk expression with our strategy over directly employing bulk expression in the loss function to guide the *in silico* inference of a target sample. This is done by comparing the accuracy of representation between inferred pseudobulk (based on the real bulk) and real bulk (direct usage) against the ground truth pseudobulk derived from the real single-cell data of the target sample (Supplementary Fig. S36). The pseudobulk acts as a summary, averaging the expression of all cells within a single-cell dataset to guide deconvolution; the goal is that accurately deconvoluted single cells will have expressions that match this average, or pseudobulk. By comparing the pseudobulk inferred by our model to the actual pseudobulk of a target sample, we've shown our method can closely replicate the actual data, validating our strategy of training the deep learning model on pseudobulk differences. These differences are drawn from the variations found in the real bulk data between the representative and target samples, providing a reliable basis for the model's training process.

Comment 8: * Page 15, “This is particularly remarkable given the unique challenges presented by the real bulk data.” Can the authors explain these challenges in relation to the deep learning method developed?

Response: Thank you for highlighting a major challenge that was not sufficiently detailed in our initial discussion. We have updated the “Fine-tune the deep generative learning model to infer the single-cell measurements for the target samples” section in the Methods of our manuscript (colored in blue) to clarify that real bulk data differs substantially from the pseudobulk of the single-cell (i.e., average gene expression derived from single-cell data), which is critical to our model's fine-tuning for *in silico* inference of the target sample. This fundamental discrepancy between real bulk and pseudobulk will induce bias in accuracy of the semi-profiling, which poses a significant challenge for various deconvolution methods, including those that are semi-related to ours. Different from existing approaches, here we addressed this challenge using a designated robust pseudobulk inference technique, as detailed in our response to Reviewer #1 comment 6.

Comment 9: * The alignment of the directly sequenced and semi-profiled samples is indeed remarkable. It would be appreciated if the authors explain how overfitting has been avoided.

Response: We are grateful for your positive remarks on the performance of our method. We explain below how we prevent overfitting in our semi-profiling pipeline. The semi-profiling single-cell generation is performed by first reconstructing the representatives' data and then inferring the targets' data by introducing 'target bulk loss'. We have strategies for avoiding overfitting at both stages.

First, the reconstruction phase is an unsupervised learning task, where the main concern regarding overfitting is the potential for the model to adapt to technical noise rather than the true underlying data distribution. We addressed overfitting in this "reconstruction" stage using the following techniques: (1) Weight decay: We incorporated weight decay (Krogh and Hertz 1991, Loshchilov and Hutter 2017), a widely recognized method for reducing overfitting, into our deep generative learning model; (2) Latent space prior: Our model's VAE generator utilizes a standard Gaussian prior for the latent space distribution, serving as a regularization mechanism to avert over-adaptation of the model parameters to the training data. These details have been added to the "Deep generative learning model architecture" section of the Methods part of the revised manuscript.

Second, to prevent overfitting during the inference stage after learning the data distribution of the representatives, we employ the following strategies: (1) We incorporate information from bulk data through the "target bulk loss" to prevent the model from overfitting to the seen representative's single-cell data and failing to produce accurate data for the unseen target sample. This requirement compels the model to adjust the data distribution it generates from the representative sample towards the target sample. This shift is illustrated in Supplementary Figs. S2, S25, S37, and S50, where the generated data progressively aligns closer to the ground truth; (2) The strategies for avoiding model overfitting during the reconstruction, described above, are also applied during the inference stage. Finally, our strategy of estimating pseudobulk from real bulk data, as detailed in response to Reviewer #1 comment 6, further prevents the model from overfitting to the technical noise introduced by sequencing technologies and various data processing steps. This comprehensive approach ensures that our model robustly reflects the underlying biological variations rather than the artifacts of data handling. Relevant information can be found in the "Fine-tune the deep generative learning model to infer the single-cell measurements for the target samples" section of the Methods in the revised manuscript.

Reviewer #1 (Remarks on code availability):

Code availability comment 1. Installation of semiprofiler was seamless.

Response: Thank you very much for your positive feedback regarding the installation.

Code availability comment 2. The

<https://scsemiprofiler.readthedocs.io/en/latest/environment.txt> link is invalid

Response: Thanks for the comment. The link to our readthedocs documentation is <https://scsemiprofiler.readthedocs.io/en/latest/>. Also, feel free to use the hyperlink in our GitHub repository to reach the readthedocs. The environment.txt file is in our GitHub repository, instead of the readthedocs page.

Code availability comment 3. The README file had enough information.

Response: Thanks for the positive feedback. We will keep it updated whenever new functionalities are added.

Code availability comment 4. I am not familiar with Jupyter notebook so I was not able to run the code (although I did have Jupyter notebook installed).

Response: Thank you for the comment. We have now included instructions to assist users in running the Jupyter notebook. It's important to note that this Jupyter notebook scripts that we provided serve merely as examples to facilitate users in leveraging our tool more effectively. Users are at liberty to employ our fully-encapsulated scSemiProfiler Python package in any manner they prefer.

Reviewer #2 (Remarks to the Author):

I co-reviewed this manuscript with one of the reviewers who provided the listed reports

Response: We appreciate the collaborative review effort and the time you have invested in providing feedback on our manuscript. Thank you!

Reviewer #3 (Remarks to the Author):

General comment: This paper describes a method for deconvolution of bulk RNA-seq data that is capable not only of extracting percentages of cell types (as usual), but also of generating data at the single cell level. To the best of my knowledge, this is a novel feature. As with other deconvolution algorithms, it requires training data consisting of bulk samples and corresponding scRNA-seq, but an important point (not emphasized enough by the authors) is that it appears to work without the need for previously labeled training data. This means that one could theoretically generate single cell data from bulk data and work on de novo labeling of cell types.

The algorithmic approach to achieve the goal, although based on SCVI, seems appropriate and adds value to the study on a methodological level; however, the methods section should be improved in terms of structure and level of detail in the descriptions. The code is available to the scientific community via github.

Despite all the positives, the work needs to be improved in two main areas: a) to further show that the method is able to generate profiles of cells other than those used in training; b) to demonstrate that, after the proposed active learning process, the single cell data generated from the samples with only bulk data add relevant and necessary information to reach better results or conclusions than if only the real single cell data of the selected samples are used.

Response: We appreciate the reviewer's recognition of our tool's unique capability for single-cell level deconvolution, which distinguishes it from existing alternatives. We are also grateful for the insightful observations and thorough understanding demonstrated in the review, particularly regarding the scSemiProfiler's capacity for de novo cell type labeling, which we did not sufficiently emphasize in our manuscript.

Yes, here we confirm that our tool possesses the capacity for true single-cell resolution reconstruction from bulk data, where individual cells are generated from bulk data and a reference single-cell sample. To further substantiate this unique functionality, we applied de novo cell type identification to the semi-profiled COVID-19 dataset. We then compared the de novo cell type annotation results, derived from the single-cell data profiles generated by our model, with the ground truth. Briefly, de novo labeling of cell types can be achieved from our semi-profiled single-cell data using a biomarker-based strategy similar to those employed in other relevant studies (Hao, Hao et al. 2021, Hasanaj, Wang et al. 2022). Specifically, we identified the top biomarkers associated with each cell cluster identified from the semi-profiled single-cell data. These biomarkers were then compared with known cell type markers from databases such as CellMarker (Zhang, Lan et al. 2019, Hu, Li et al. 2023) or Panglao DB (Franzén, Gan and Björkegren 2019) to annotate the cell types de novo. We compared the de novo cell type identification results with supervised cell type annotations and demonstrated the effectiveness of scSemiprofiler in de novo cell type identification (Supplementary Fig. S7). In response to your comment, we have expanded our discussion of this topic in the “The semi-profiled COVID-19 single-cell cohort exhibits significant similarity to its real counterpart” section in the Results part in our manuscript to emphasize this important feature of the tool as per the reviewers' suggestion.

Supplementary Fig. S7: Side-by-side comparison of the real-profiled data and semi-profiled data annotated by de novo annotation. The semi-profiled cohort annotated via supervised cell type annotation can be found in Fig 2b.

In response to the reviewer's comment on our algorithms and code availability, we have revised the corresponding parts to enhance the following aspects: (1) the structure and overall readability; (2) the details concerning the techniques used, such as pseudobulk inference for handling real bulk data, the structure and parameters of neural network models, and data feature processing. All of these changes are detailed in the updated Methods section of our manuscript.

We also thank the reviewer for their insightful comments on the necessity of demonstrating that our model can generate new cell profiles beyond those used in training, and that these newly generated cells provide valuable biological insights. In response, we performed additional experiments, along with those in the original manuscript, to show that our method can generate new cell profiles that closely resemble the ground truth (see Reviewer # 3 comment 11). The results further demonstrate that the generated cells add significant value to single-cell analysis and are crucial for enhancing the similarity to the real cohort, thus enabling more accurate conclusions and deeper biological insights. Please refer to our response to Reviewer # 3 comment 1 below for more details.

Comment 1: - Can we gain more insight or extract new or different information by adding the semi-profiled cells of samples without sc data together with the real sc data obtained for the selected representatives?

Response: We very much appreciate the reviewer's critical insight on the importance of illustrating the added value of semi-profiled (generated) single-cell data alongside real single-cell data obtained from selected representatives in the downstream biological analysis of the cohort. To demonstrate how semi-profiled single-cell data can provide insights that are not possible with representative single-cell data alone, we conducted a comprehensive analysis from multiple perspectives below. Specifically, we compared analyses derived from the semi-profiled cohort to those obtained from representative single cells, using the COVID-19 dataset with the most abundant number of samples and the highest sample quality as a demonstration. These findings highlight the crucial role of *in silico* generated single-cell data in enabling a deeper understanding of the cohort and facilitating the exploration of various disease conditions and individual sample characteristics.

At the cohort level, the inclusion of generated cells significantly improves the overall data similarity to the real cohort and enhances analysis results compared to using only representative cells. The UMAP results, as indicated in Supplementary Fig S16a, b, c, e, f show that relying solely on cells from representatives fails to encompass many areas of the original cohort's UMAP. This leads to the omission of certain cell subtypes and a lack of intra-cluster variability. This observation is supported by a comparison between the semi-profiling and the "representative-only" methods shown in Supplementary Fig. S16. These plots demonstrate that semi-profiling more accurately captures the real-profiled cohort, especially in terms of cell type proportions and overall gene expression, compared to using only representatives. While a sufficient number of representatives can achieve comparable results to the real cohort for straightforward tasks such as cell type marker identification (Supplementary Figs. S17 and

S18), the semi-profiled cohort still presents advantages when conducting detailed single-cell analyses, such as pseudotime (Pearson correlation: 0.809 for semi-profiled vs. 0.545 for representative-only, Supplementary Fig. S18d). It is also crucial to note that the effectiveness of representative data is enhanced by our selection strategies, which include bulk clustering and active learning.

Supplementary Fig. S16: Comparing the semi-profiled COVID-19 cohort, real-profiled cohort, and representative-only cohort. **a-c**, UMAP visualization of the semi-profiled cohort, real-profiled cohort, and representative-only cohort (only representative cells are included) respectively. Different colors represent different cell types and are consistent with **(d)**. **d**, Stacked bar plots visualizing different cohorts' cell type proportions in different COVID-19 severity levels. The semi-profiled cohort shows high similarity with the real-profiled cohort, the Pearson correlations between the semi-profiled cohort and real-profiled cohort in each condition are: Healthy: 0.987, Asymptomatic: 0.970, Mild: 0.996, Moderate: 0.992, Severe: 0.978, Critical: 0.989. Representative-only cohort shows a lower similarity with the real-profiled cohort. Pearson correlations: Healthy: 0.971, Asymptomatic: N/A, Mild: 0.965, Moderate: 0.981, Severe: 0.961, Critical: 0.859. **e**, Joint UMAP showing the similarity between the semi-profiled cohort and the real-profiled cohort. **f**, Joint UMAP the representative-only cohort fails to cover some areas of the real-profiled cohort's UMAP. **g**, Interferon pathway activation pattern comparison between the three cohorts. The representative-only cohort cannot generate results for "Asymptomatic".

Supplementary Fig. S17: Dot plots visualizing the top cell type signature markers' expression patterns in the semi-profiled COVID-19 cohort, real-profiled cohort, and the representative-only cohort.

Supplementary Fig. S18: Downstream single-cell analysis results using real-profiled COVID-19 cohort, semi-profiled cohort, and representative-only cohort. **a**, RRHO plots comparing the similarity between CD4 markers found using real-profiled cohort and semi-profiled cohort (left) and the similarity between CD4 markers found using real-profiled cohort and representative-only cohort. **b**, Comparison of GO enrichment analysis using CD4 genes found by different cohorts (top: real-profiled vs. semi-profiled, bottom: real-profiled vs. representative-only). **c**, Comparing the cell-cell interaction analysis results between the three cohorts. **d**, Pseudotime analysis results between the real-profiled cohort and semi-profiled cohort are more similar (Pearson correlation: 0.809) than that between real-profiled cohort and representative-only cohort (Pearson correlation: 0.545).

At the disease condition level, using only representative cells often fails to cover all disease conditions, which either precludes the study of specific conditions or leads to dramatically worse results compared to using the semi-profiled cohort. In the case of the COVID-19 cohort, we utilized 28 representatives out of 124 total samples for most analyses. These 28 representatives do not cover the 'Asymptomatic' condition, making it challenging to investigate this specific disease condition using single-cell data. For instance, the estimation of cell type

proportions for the 'Asymptomatic' condition cannot be achieved using only representatives, as there is no cells from this conditions were profiled at the single-cell level. shown in Supplementary Fig. S16d. Moreover, even for conditions that are covered by the representatives, the analysis results are often less reliable due to a lack of statistical power or failure to capture the internal variety within each condition. For example, Supplementary Fig. S16 d and g show that the semi-profiled cohort is significantly more similar to the real-profiled cohort than the representative-only version in terms of cell type proportion and pathway activation patterns. Notably, when investigating biomarkers for disease conditions, the semi-profiled dataset demonstrates a stronger similarity to the real-profiled dataset than the representative-only version, as illustrated in Supplementary Fig. S19. For instance, in studying the 'Critical' condition, the top 100 markers identified by the semi-profiled dataset have 68 overlaps (p-value = 5.43×10^{-109}) with the real-profiled ground truth, whereas the representative-only dataset only has 30 overlaps (p-value= 4.31×10^{-31}).

This disparity becomes even more pronounced when the number of representatives is reduced, often due to limited budget constraints in cohort studies. To illustrate this scenario with fewer representatives, we examined the COVID study using just 12 representatives and compared the findings with those from the semi-profiled results (utilizing the same 12 representatives). The results, presented in Supplementary Fig. S20 for the 12 representatives version, demonstrate that even with only 12 representatives, the semi-profiled dataset continues to maintain high similarity in terms of disease marker analysis, while the similarity of the representative-only results declines further.

Supplementary Fig. S19: Evaluating the disease condition markers identified using the semi-profiled COVID-19 cohort and only the 28 representatives by comparing against the real-profiled cohort. Markers are differentially expressed genes between the disease condition group and the healthy group in each cohort. **a**, Markers for “Asymptomatic” conditions identified in the semi-profiled cohort align closely with those from the real-profiled cohort; the representative-only cohort lacks data on “Asymptomatic” patients and hence did not perform this analysis. **b**, “Mild” markers comparison. **c**, “Moderate” markers comparison. **d-e**, Comparisons of “Severe” and “Critical” condition markers, respectively. Markers from the semi-profiled cohort demonstrate a significantly higher overlap with those from the real-profiled cohort compared to the representative-only cohort, reflecting greater similarity in the GO enrichment analysis results (more overlaps in top 10 GO terms).

Supplementary Fig. S20: Comparing disease condition markers identified using three COVID-19 cohorts: the semi-profiled cohort based on 12 representatives, real-profiled cohort, and the only 12 representatives. a, Despite the limited number of representatives, the semi-profiled cohort’s “Asymptomatic” markers closely align with those from the real-profiled cohort. b, “Mild” markers comparison. c-e, Markers for “Moderate,” “Severe,” and “Critical” conditions in the semi-profiled cohort show significantly greater similarity to those in the real-profiled cohort, both in terms of marker overlap and GO enrichment analysis results compared to the representative-only cohort.

At the individual sample level, studying non-representative samples using only single-cell data from representatives is not feasible without *in silico* inference since these samples were not directly profiled. This limitation raises two major concerns: Equity, Diversity, and Inclusion (EDI) and the potential for missed scientific discoveries. Excluding non-representative samples from analysis introduces biases in our understanding of the disease, as these samples are not represented. This could result in the oversight of specific groups that may not be well represented by the selected representatives, leading to unfair representation. Furthermore,

excluding non-representatives is risky as they may possess unique biological traits not found in representatives. However, the scSemiProfiler facilitates the inclusion of single-cell analysis for these non-representatives without additional costs. The generated single-cell data for individual samples closely resemble the target sample, as evidenced by the UMAP visualizations in Supplementary Fig. S2.

Moreover, representing non-representative samples directly using single-cell data from their corresponding representatives (as described in the "selection-only" method in our manuscript) can result in data too divergent from the ground truth to yield meaningful analysis results. For instance, Supplementary Fig. S21 illustrates a non-representative sample with a 'Critical' severity level represented by its representative. This figure shows that the inferred single-cell data has a cell type proportion dramatically more similar to the target sample than to the representative sample. In Supplementary Fig. S21 b, we present 40 target sample biomarkers, differentially expressed genes (DEGs) of this sample against a healthy control. The left 20 DEGs show gene expression more similar to the target sample than the representatives, and the right 20 show where the representatives have more similar patterns than our inferred sample. The visualizations clearly indicate that the inferred sample is significantly more similar to the target than the representative. Even among the least similar 20 genes on the right-hand side, more genes show similar average expressions in the three samples. The Pearson correlations between the semi-profiled non-representative sample and the target sample (0.904) are significantly higher than those between the representative and the target sample (0.568). Importantly, enrichment analysis in Supplementary Fig. S21 c and d reveals that these DEGs are crucially relevant to the immune response, such as "regulation of immune response", "Immune System", "Innate Immune System", "Cytokine Signaling in Immune System" and several terms relevant to MHC (Zhang, Chen et al. 2021), interferon pathway (Gruber 2020, Lee and Shin 2020, Galbraith, Kinning et al. 2022), indicating that using only the representative for analysis could likely lead to incorrect conclusions about the studied disease, since many of those key disease associated immune terms will be missed.

All discussions above have been included in our revised manuscript ("In silico generated single-cell data provides extra information for understanding the studied cohort" section in the Results) to further justify the necessity of our semi-profiling method.

Supplementary Fig. S21: Semi-profiled cohort can provide a much more accurate representation of the ground truth cohort and reveal more biological insights as demonstrated in the COVID-19 dataset. a, The inferred target sample with “Severe” COVID-19 in the semi-profiled cohort has significantly more similar cell type composition with the ground truth than the representative. **b**, The inferred target sample has a significantly more similar disease marker gene expression pattern with the ground truth than the representative. Markers are identified by performing differentially expressed gene analysis between the target ground truth sample and healthy controls. The left 20 are DEGs that show gene expression more similar to the target sample than the representatives, and the right 20 show where the representatives have more similar patterns than the inferred sample. **c-d**, GO and Reactome enrichment analysis using the left 20 DEGs in (b).

Comment 2: The number of samples is important when performing sample-based statistics, such as comparing percentages of cell types, but in this scenario traditional deconvolution methods are sufficient and single-cell resolution is not necessary. In downstream analyses with the three datasets, where is it really important to generate more single cell data within cohorts? This needs to be discussed, also commenting on the inherent increase in data size for downstream sc analysis when adding semi-profiled data, and the obvious increase in resource requirements for processing and management. This is not shown in this paper.

Response: We thank the reviewer for this insightful comment. We agree that for certain tasks, such as calculating the percentage of cell types, traditional deconvolution methods are often sufficient and perform well, as single-cell resolution is not essential in these scenarios. If this

is the primary focus or interest of users, they can opt for conventional deconvolution methods. Nevertheless, we would like to emphasize that even for tasks like cell type percentage quantification, where single-cell resolution may not be necessary, our *scSemiProfiler* pipeline can still offer superior performance, as demonstrated in our benchmarking figures (Fig. 2j, k, l, Fig.4j, k, l, Fig.6j, k, l, Supplementary Fig. S43f, h, i).

In this revision, we have enhanced our discussion to emphasize that the additional resolution provided by single-cell data enables a range of analytical tasks not possible with conventional deconvolution methods. Such tasks include UMAP visualization of single cells, inspection of specific pathways' gene expression, analysis of biomarkers with strong statistical power, interactions between cell types, and pseudotime analysis to explore intra-cluster heterogeneity. These capabilities undoubtedly lead to a deeper understanding of disease mechanisms. Furthermore, it's important to note that the value of our research extends beyond improved resolution. The inclusion of more single-cell data, especially from non-representative samples as generated by our *scSemiProfiler*, provides significant additional biological insights. This is particularly evident in the enhancement of cohort, condition, and patient-level analyses. For a detailed discussion on how the semi-profiled non-representative single-cell data adds substantial value, please refer to Reviewer #3, comment 1.

Regarding the resources required for processing and managing *in silico* generated cells, we believe that although more computational resources are needed, the demands are manageable and justified for most laboratories. Firstly, storing the generated data is a minor issue for most modern computers. The data consists of cell-by-gene count matrices, which are relatively small in size. For instance, the largest COVID-19 cohort's preprocessed real-profiled data (124 samples, 637144 cells, 6030 genes) is only 1.6GB, easily accommodated by most current devices. Secondly, processing a larger single-cell dataset is also feasible. While generating more single-cell data does increase the computational burden, it does not exceed the resources required for processing actual profiled single-cell cohorts commonly reported in literature, such as the COVID-19 study dataset we analyzed. Our runtime and memory curve, shown in Supplementary Fig. S1, support that our method's runtime is reasonable (semi-profile 100 samples with 4000 cells within 4 hours), and the memory usage is comparable to or better than other methods (25GB RAM and 3GB GPU memory assuming 4000 cells per sample, for arbitrary number of samples). The statistics are measured using our Linux server with 8 GPUs (4 Tesla M10 and 4 NVIDIA A16) and two Intel Xeon Platinum 8160 CPUs (each featuring 24 cores and supporting 2 threads per core, resulting in a total of 96 logical CPUs available for computation). Additionally, as highlighted in our previous response to the general comment in the beginning of Reviewer #3's comments, the benefits of having single-cell data for each sample outweigh the acceptable increase in computational resource usage. As per reviewer's suggestion, we have now updated the running time and memory complexity information in Supplementary Fig. S1 and get them referenced in the main text.

Supplementary Fig. S1: Runtime and memory usage. The statistics are measured using our Linux server with 8 GPUs (4 Tesla M10 and 4 NVIDIA A16) and two Intel Xeon Platinum 8160 CPUs (each featuring 24 cores and supporting 2 threads per core, resulting in a total of 96 logical CPUs available for computation). **a**, Estimated runtime of *scSemiProfiler* when 4 representatives are sequenced and all 8 GPUs are used. The x-axis represents the number of target samples in the cohort, and the y-axis indicates time in hours. **b**, Comparison of memory usage among various deconvolution methods. Memory usage was monitored across deconvolution methods when using 1 to 5 samples (each has 4000 cells) as single-cell references for deconvoluting 4 bulk samples. The cell data matrix is sampled from the COVID-19 cohort and contains 6030 gene features. Most deep learning methods have very low memory usage and only increase slightly for reading larger input data matrices. Our method's memory usage does not increase with the number of samples, as different samples are trained using separate models sequentially. Bisque and MuSiC & NNLS exhibit nearly linear memory usage. CIBERSORTx, not being open-source software, offers its service exclusively through a website. Therefore, testing the exact memory usage is not feasible. The website permits a maximum upload of 1GB data per user, approximately equivalent to 42826 cells' gene expression data stored in TXT format, as required by the website.

Comment 3: - What is the significance of the gene sets (scores) in this framework? What is the percentage of final features that are genes and not gene sets? A comparison between using only gene expression (without gene sets), using only gene sets, and using both is needed to understand these points.

Response: Thank you for pointing out the lack of clarification regarding gene sets. Incorporating gene sets has been previously adopted and demonstrated its effectiveness in improving the quality of cell embeddings and distribution learning (Li, Ding and Bar-Joseph 2022). Following the same objective, here we also utilized the gene set score in our framework. To demonstrate its effectiveness, we compared the cell embedding quality generated with/without using gene set scores using an independent dataset (Zeisel, Muñoz-Manchado et al. 2015) that is also employed in the SCVI study. We trained models using cells with only gene expression data and cells augmented with gene set scores separately. Then we used these two models to perform reconstruction task for the dataset and found the one augmented with gene set score achieves lower MSE (19.33 vs. 25.48 Supplementary Fig. S49). The generator VAE in our project primarily reconstructs gene expression profiles using unsupervised learning. This reconstruction demands gene expression data as input. Excluding this data, as would occur if we only used gene sets, would impair the VAE's learning and its predictive accuracy for gene expression values. The training process heavily relies on 'reconstruction loss,' which necessitates real gene expression data for comparison and precision. Using only gene set scores for reconstruction would be inadequate, as they lack the comprehensive expression details essential for recreating original gene expression profiles. Hence, our comparison here strictly

involves the results from gene expression data, with and without gene set scores, to underscore the gene set scores' contributions. The percentage of genes in the final features are: COVID-19: 88.8%, colorectal cancer: 84.5%, iMGL: 89.8%, hamster: 90.0%. The gene set experiment information has been added to the “Representative single-cell profiling and processing” section of the Methods.

Supplementary Fig. S49: Gene set score ablation study. With gene set scores augmented, our generator can perform more accurate reconstructions on the cortex dataset (PMID: 25700174) with lower average MSE (19.33 vs. 25.48, Wilcoxon test p-value: 2.52×10^{-4}).

Comment 4: - The difference between Fig. 2c and 2d (and equivalent in other figures) is not clear. 2c shows only data from representatives? What is shown is better understood in Supp Fig 2...

Response: Thank you for the note. Fig. 2c illustrates a ground-truth cohort with complete real single-cell profiling across all samples, contrasted against a semi-profiled cohort where only selected representatives have undergone single-cell profiling. This comparison validates our tool's ability to approximate a fully profiled dataset. In Fig. 2d, the semi-profiled cohort is depicted with color differentiation to highlight the distinction between actual profiled representatives and in silico generated cells. This emphasizes our method's capacity to extend beyond the representatives, enriching the dataset. Enhancements have been made to both figures and their corresponding explanations in the revised manuscript for greater clarity.

Comment 5: - Plot umaps colored by real/semi-profiled (Fig. 2c), but separately for each of the B clusters (of samples, with 1 representative each).

Response: We appreciate the recommendation to examine our results through multiple lenses. Following your suggestion, we have generated UMAPs for each representative of the B clusters, colored by real/semi-profiled data, and have incorporated these figures (Supplementary Figs. S3, S26, S38) into our manuscript for more detailed analysis.

Comment 6: Also, umap plots colored or split by sample should be included in the supplement to check if the segregation of cell types depends on the sample/sample_cluster and if there are always some cells from representatives.

Response: We are grateful for your input, and in accordance with it, we have prepared UMAP plots that delineate the segregation of cell types by sample, which can now be found in the supplementary section of our updated manuscript (Supplementary Figs. S4, S27, S39). We acknowledge that due to the large number of different samples, the UMAP plots may appear somewhat congested, potentially complicating the assessment of whether cell type segregation is influenced by the sample or cluster. To clarify this, we direct your attention to the new stacked bar plots (Supplementary Figs. S6, S29, S41), which present a clear percentage breakdown of cell types by sample and cluster. Additionally, we prepared UMAP plots of each sample cluster colored by cell type (Supplementary Figs. S5, S28, S40), which shows the cell type distribution in each sample cluster is different. These visuals will further assist in determining if there are consistent representative cells within each cell type. Please refer to the revised figures for this detailed information.

Comment 7: - For evaluation and benchmarking, it is recommended to report CCC (Concordance Correlation Coefficient) as it is a more appropriate metric than RMSE or Pearson correlation alone.

Response: We appreciate your recommendation to utilize the Concordance Correlation Coefficient (CCC) for more accurate benchmarking. Following your advice, we have incorporated the CCC into our benchmarking analysis across all results. The updated benchmarking confirms that our method maintains a considerable advantage over other counterparts in terms of overall deconvolution performance. We have updated all our benchmarking panels in Fig. 2, Fig. 4, Fig. 6, and Supplementary Fig. S43 accordingly.

Comment 8: - Were the geneset scores concatenated with gene expression data also used to calculate losses? Are they treated exactly the same as the genes (like "extra genes")?

Response: Thank you for your comment. Indeed, the geneset scores were concatenated with the gene expression data and factored into the loss calculations, essentially functioning as additional genes. A separate MLP in the encoder network is used for encoding gene set score values to the encoder's hidden layer, and a separate MLP in the decoder is used for generating gene set values from the hidden layer. However, their utilization is specifically tailored to improve the learning of the cell representation and they do not play a role in the computation of bulk tissue losses to guide *in silico* inference. We've made sure to clarify this distinction

within the revised manuscript for transparency and to provide a comprehensive understanding of our approach (please refer to the section “The initial setup of *scSemiProfiler*” and “Pretrain the deep generative learning model for reconstructing the single-cell data of the selected representatives” for corresponding changes).

Comment 9: - Clarify if the training (or part of the training) is independent for each cluster of samples (representative and associated target samples). What about fine tuning?

Response: We recognize the importance of clear communication about our training process. Yes, the training is independent for each of the clusters. Initially, we pre-train a reconstruction model that's tailored to the representative of each cluster. We then fine-tune this model for each associated target sample, employing the target bulk loss to direct the transformation from the representative's profile to that of the target. This process is critical for ensuring the model's accuracy in reflecting the unique characteristics of each target sample (i.e., matching the bulk sequencing). We've included an in-depth explanation of our pre-training and fine-tuning methodology in the revised manuscript. For a thorough understanding of our techniques, please refer to the pertinent section (“Pretrain the deep generative learning model for reconstructing the single-cell data of the selected representatives” and “Fine-tune the deep generative learning model to infer the single-cell measurements for the target samples” in Methods).

Comment 10: - Do plots and results comparing real-profiled vs. semi-profiled represent all available samples? Only representatives for real-profiled and only non-representatives for semi-profiled? Clarification is important in any case.

Response: Thank your request for clarification. Yes, comparing real-profiled vs. semi-profiled means comparing all available samples in the two versions of cohorts, so that we can mimic the in real world situation that only the semi-profiled dataset is available and show we can still get similar results as if the real-profiled dataset is available. In short, these are two distinct approaches to profiling the entire cohort. The first, designated as 'real-profiled', pertains to the actual sequencing of all samples within the cohort, serving as the ground truth. The second, referred to as 'semi-profiled', applies to the cohort in a modified form: it includes actual single-cell profiling for representative samples and computational inference for the remainder. This has been further clarified in the revised manuscript (in the “The semi-profiled COVID-19 single-cell cohort exhibits significant similarity to its real counterpart” section of the Results).

Comment 11: Results should be reported (at least in the Supp) separately for representatives (sc data used for training) and non-representatives (only (pseudo) bulk data was used) in order to assess proper evaluation. Also, in plots where different samples are represented, the ones that were used as representatives should be indicated. For example Fig.S3

Response: We appreciate your suggestions for illustrating the capability of our model to generate new cells beyond those seen in training and for demonstrating how these cells are instrumental in uncovering biological insights. In the updated manuscript, we have delineated results derived from representative samples as well as those from *in silico* generated samples

(Fig. 2d, Fig. 4d, and Fig. 6d). To address the reviewer's inquiry regarding the utility of inferred cells from non-representative samples, we present the following evidence based on comprehensive analyses.

Firstly, the deep generative learning model produces the majority of cells (COVID-19: 74.54%, colorectal cancer: 66.03%, iMGL: 65.96%, hamster: 87.50%)—those inferred for non-representative samples—across all datasets we examined. This is quantified in Supplementary Fig. S22a. The overall positive results underscore the accuracy of the generated cells; should they deviate from real-profiled data, it would adversely affect the aggregate results of single-cell analyses. Furthermore, the comparative analyses presented in Supplementary Figs. S16, S17, S18, S19, S20, and S21 demonstrate the superiority of the semi-profiled dataset over the dataset consisting only of representative cells, substantiating the value of the additional generated cells.

Secondly, the cell type proportions obtained using only the generated non-representative cells closely align with those from the actual profiled datasets (Supplementary Fig. S22b), further affirming the reliability of the inferred cells. The concern that replicating representatives to represent their associated target samples might replicate similar proportions is addressed by our model's detailed generative process, which is designed to maintain the integrity and variety of cell types. Moreover, our analysis in response to Reviewer #3 comment 1, clearly shows that representing target samples using only representatives, without the benefit of *in silico* inference, could introduce biases and potentially lead to inaccurate conclusions.

Supplementary Fig. S22: Overview of the real-profiled cells, representative cells, and in silico inferred cells in the four cohorts we analyzed. a, Pie plots showing that in all semi-profiled cohorts, most cells are inferred using our deep generative model. **b,** Stacked bar plots showing the cell type proportions in the inferred-only cells, representative cells, and the whole real-profiled cohort are very similar.

To further affirm the utility of semi-profiled non-representative cells, we perform another in-depth comparative analysis between these cells and the original real-profiled cells within the COVID-19 dataset. This serves as a robust demonstration of our model's effectiveness in broadening the scope of cellular profiles analyzed. The comparison focuses on cell type distributions and pathway activations, revealing high concordance in cell type percentages and the preservation of key functional attributes such as interferon activation pathways, which are crucial for the immune response to COVID-19. The similarity in these critical biological aspects is substantiated by figures illustrating cell type portages and pathway activations

Supplementary Fig. S23. Moreover, comprehensive downstream analyses—including biomarker identification, pathway enrichment, cell-cell interaction studies, and pseudotime trajectory analyses—using the inferred cells further validate the integrity and biological relevance of our generated data. The results of these analyses Supplementary Fig. S24 showcase the model’s capability to produce data that is not only quantitatively robust but also qualitatively insightful, thus enhancing the dataset with a more representative spectrum of cellular states for advanced biological exploration.

Supplementary Fig. S23: Similarity between the inferred-only dataset (representing the COVID-19 cohort using only in silico inferred cells) and the real-profiled dataset. **a**, Real-profiled COVID-19 cohort UMAP visualization. Colors represent different cell types and are consistent with **(b)** and **(e)**. **b**, UMAP visualization of the inferred-only dataset. **c**, UMAP visualization combining real-profiled cohort and inferred-only cells, showing the deep generative learning model is capable of generating cells similar to the real-profiled cohort for areas not covered well by the representatives. **d**, UMAP visualizing the representatives’ cells and inferred-only cells. **e**, Cell type proportion comparison between the real-profiled dataset and the inferred-only dataset in different COVID-19 disease severity levels. The Pearson correlations are: Healthy: 0.982, Asymptomatic: 0.970, Mild: 0.995, Moderate: 0.975, Severe: 0.963, Critical: 0.970 **f**, Interferon pathway activation pattern comparison.

Supplementary Fig. S24: Downstream analysis using only in silico inferred cells (inferred-only dataset) generate similar results with the real-profiled COVID-19 cohort. a, Dot plots showing nearly identical expression patterns of top cell type signature genes in the inferred -only dataset and real-profiled dataset. **b,** RRHO plot comparing the CD4 markers found using the two versions of datasets. **c,** CD4 cell type signature genes GO enrichment analysis results comparison. **d,** Cell-cell interaction analysis results comparison between the real-profiled dataset and inferred-only dataset. **e,** Pseudotime results comparison using the real-profiled and in silico generated CD4 cells.

Comment 12: - The results of the downstream analyses are compared between real-profiled and semi-profiled sc data. All samples in the cohort were used (original sc data and all semi-profiled from the corresponding (pseudo) bulk data)? In any case, this comparison shows that semi-profiled cells are close to real sc data. But it would be very relevant to compare all these sc-based results between: a) using only real sc data from the representatives; b) using sc from the representatives together with the all sc semi-profiled data. In this way, we can mimic a real scenario where there is a limited budget, so we cannot run as many sc experiments as we would like, and we decided to use scSemiProfiler to fill the gap. Are there more insights when adding the semi-profiled cells to the study? Also comment on the increase in data for downstream sc analysis when adding semi-profiled data, and the obvious increase in resource requirements for processing and management. This is linked to first point.

Response: We are grateful for your insightful feedback and the recognition of the resemblance between our semi-profiled data and the actual data. As noted in our detailed response to Reviewer #3 comment 10, all samples have been included in our analyses. For an in-depth comparison between representative samples and the full semi-profiled cohort, as well as the enhanced insights offered by the in silico generated cells, we direct you to our initial comprehensive response to Reviewer #3's general comment and Comment 1. With regard to the additional resource requirements incurred by the generated cells, please see our elaboration in the response to Reviewer #3 comment 2.

Comment 13: - PAGA analysis in Fig. S8 (and S16, and 7d). In order to properly use graph plots for visual inspection to check the similarity of output graphs, clusters for real and semi-profiled datasets should be placed in the same locations.

Response: We appreciate your constructive feedback on enhancing the quality of our plots. In line with your suggestions, we have revised the PAGA analysis plots in Supplementary Fig. S13, S34, S47d, S48d and Fig. 7d to ensure that clusters from both real and semi-profiled datasets are positioned consistently for accurate visual comparison.

Comment 14: - Fig. S10 (more than S18), especially b), which shows examples of representative real, target real, and target semi-profiled, may be placed somewhere in the main text and figures due to its importance. It shows that semi-profiling is not a "noisy replication of representatives", but is able to correctly generate cells that are similar but different from the corresponding representative, based on the differences between the two (pseudo) masses. This figure could be completed and improved by adding the semi-profiled results after the different training stages (and not only the last one), showing why the following stages improve the results.

Response: Thank you for your positive feedback on the method's performance and the valuable suggestion to enhance the presentation of our results. We have restructured the figures to better illustrate the examples within the main text (please refer to Fig. 2e and f, Fig. 4e and f, Fig. 6e and f). Additionally, we recognize the benefit of displaying the results after different training stages as per the reviewer's suggestion to demonstrate the impact of each phase. These figures

have been created and are included in the supplementary materials (please refer to the Supplementary Fig. S50).

Comment 15: - In Figure 8, the authors show the difference in performance when using an active or a passive learning strategy, through an iterative process to gradually increase the number of representatives. But what if the selection of N representatives was done by Kmeans from the beginning (independent of the results of scSemiProfiler and based only on the differences in the bulk data)? Are these results still worse than the active iterative selection, which has to quantify the differences in $\text{bulkPCA}(D_b) + \text{bulkGenes}(D_{pb}) + \text{scPCA}(D_{sc})$? Discuss this

Response: We thank you for your perceptive observations regarding our representative selection strategy. Further detail on this topic is addressed in our reply to Reviewer #4 minor comment 1. Briefly, our analysis contrasts selection methods employed in a multi-round setting, aiming primarily to conserve resources by enabling users to conclude the process when they deem adequate representation has been reached. In the updated scSemiProfiler, we have introduced the option of selecting a fixed number of representatives using K-Means at the beginning, which we call ‘global mode’. An in-depth comparison of this direct selection in global mode and the iterative selection in active learning mode within the COVID-19 cohort has revealed a significant similarity between the two methods when using an equivalent number of representatives (for further details, please refer to our response to Reviewer #4 minor comment 1). These findings have been incorporated into the revised manuscript (the last paragraph of “The semi-profiled COVID-19 single-cell cohort exhibits significant similarity to its real counterpart” and the last paragraph of “The semi-profiled COVID-19 single-cell cohort proves reliable for single-cell downstream analyses” in the Results).

However, it is important to emphasize that setting a predetermined number of representatives via K-Means (global mode) does not optimize for the minimum budget required to achieve satisfactory semi-profiling quality. Active learning mode has the potential to conclude the process with fewer representatives than a preset number, proving more cost-effective. Moreover, even in global mode, the quality of *in silico* inference using the selected representatives must be evaluated using the strategy described in our response to Reviewer #4, minor comment 1. Should the semi-profiling accuracy in global mode prove unsatisfactory with the predetermined number of representatives, active learning would then be necessary to select additional patients for profiling in subsequent rounds.

Comment 16: - The iMGL dataset is the only one with real bulk data paired with sc data. So results there are more informative and it should be emphasized in the text.

Response: Thank you for your recommendation. In response, we have highlighted the real bulk challenge associated with the iMGL dataset in section “semi-profiling with real bulk measurements yields a dataset nearly identical to the original single-cell data” of the manuscript. We've also expanded the discussion to include the challenges encountered when working with real bulk data, the specific difficulties it poses for deconvolution methods, and the strategies

our tool employs to overcome them. Additionally, to demonstrate our tool's capability with real bulk data deconvolution, we have included a new real bulk dataset from hamster lung, identified after extensive searching within the GEO database. This dataset represents the only other instance we located that aligns closest with our criteria. The average number of cells in each sample is only 1151.4 and three samples have fewer than 1,000 cells, which were supposed to be filtered according to our previous criteria, but we still use it due to lack of available dataset. This small number of cells could potentially limit model's data distribution learning and harm the performance. For comprehensive details about our method's robust performance on this new dataset with real bulk data, we refer you to Reviewer #4 minor comment 3.

Comment 17: - It is not clear what was used for training and inference in the iMGL dataset. There are real bulk and real single cell data available, but it seems that the (unnecessary) pseudobulks obtained from single cell data were used during the process. A better description is needed, and a discussion of the reasons for doing this instead of using only real data.

Response: We appreciate your request for clarification on the training and inference processes used for the iMGL dataset (real bulk data). Yes, we utilized both the real bulk data (for both representative and target samples) and the pseudobulk (for representatives single-cell) to more accurately estimate the target sample's pseudobulk, rather than directly using the real bulk for inference due to potential concerns about noise and dropouts that could lead to discrepancies between bulk data and single-cell data (and its pseudobulk). This approach aims to improve the bulk-loss guidance for the *in silico* inference process. Please refer to Reviewer #1 comment 6 for a more detailed response to a similar query. As per your suggestion, we have updated the "Fine-tune the deep generative learning model to infer the single-cell measurements for the target samples" sections in Methods accordingly to clarify the necessity of our pseudobulk inference strategy for the accurate *in silico* inference of target single-cell data.

Comment 18: - How similar are the expression distributions between real bulk data and pseudobulks generated from sc data? This can be quantified in the iMGL dataset.

Response: Thank you for your comment. In the revised manuscript, we now present a comparison between pseudobulk and real bulk data (please refer to Supplementary Fig. S36). In the iMGL dataset, the average Pearson correlation coefficient between the real bulk and pseudobulk data is 0.88, indicating a high similarity. For the hamster dataset, the similarity is somewhat lower, with an average Pearson correlation of 0.60, which can be attributed to the lesser quality of the single-cell data. Nevertheless, our approach for processing real bulk data proves robust against such technical variability. As illustrated in Supplementary Fig. S36, our method for inferring pseudobulk from the real bulk can precisely emulate the actual pseudobulk from the ground truth single-cell data (Average Pearson correlation with ground truth pseudobulk in iMGL dataset: real bulk 0.880 vs. inferred pseudobulk 0.997, Hamster dataset: real bulk 0.597 vs. inferred pseudobulk 0.923). Given that our model's training is exclusively concerned with pseudobulk data, any technical discrepancies between real bulk and pseudobulk are rendered inconsequential to our model's performance. For a more comprehensive explanation, please see our response to Reviewer #1 comment 6.

Comment 19: - Authors can use other approaches to generate pseudobulks, and check if the distributions are closer to the real ones? Will the results of scSemiProfiler be better on real samples if better pseudobulks are used?

Response: We appreciate your suggestions. We found it is also common to generate pseudobulks by summing the single-cell data and normalized (Chen, Wang et al. 2022), which is essentially the same as our method after library size normalization. Therefore, we do not expect the performance of *scSemiProfiler*'s performance will be affected by this change. Additionally, averaging single-cell data is an unbiased way of connecting real bulk and single-cell data, without detailed information about the difference between single-cell sequencing and bulk sequencing platform and their technical noise. If another pseudobulk generation method has different pseudobulk distribution, then very likely it introduces bias and affect the model in an unexpectable way. Notably, our method for dealing with real bulk data is robust and does not require a similar distribution between the real bulk and pseudobulk data. This is because we estimate the target sample's pseudobulk data in an unbiased manner (see details in our response to Reviewer #1 comment 6) given the real bulk, and let the training and fine-tuning process only cares about the pseudobulk. Thus we cater as much as possible to the single-cell side and bypass the difference between real bulk and pseudobulk. We believe this is a novel technique and can also be applied to other deconvolution methods when dealing with real bulk data. Discussions regarding this has been extended and can be found in "Fine-tune the deep generative learning model to infer the single-cell measurements for the target samples" in the Results of the revised manuscript.

Comment 20: - Fig. S18 for the iMGL dataset would be more interesting if it showed a clear target real-profiled sample that differs more from the representative real-profiled data, as in Fig. S10. In this way, the authors can show that in real data, scSemiProfiler is able to exploit the differences between bulk samples to generate correct cells that are different from those learned from the corresponding representative.

Response: Thank you for your suggestions. It's true that the iMGL dataset samples are quite homogenous, as they all originate from induced pluripotent stem cells (iPSCs), with the original study focusing on investigating subtypes. This dataset also has fewer samples (25) compared to others (COVID-19: 124, colorectal cancer: 112). It is hard to find a better dataset because existing dataset with paired single-cell and bulk data is limited. Nevertheless, we found the results compelling for a couple of reasons: (1) Despite the similarity, there are discernible differences in cell distributions between the target and representative samples, with the semi-profiled version showing more resemblance. We can still see the tendency of the generated cells (yellow) distributed closer to the target ground truth (red) than to the representatives' cells (blue). This indicates that the semi-profiled samples are either denser or sparser in the same areas as their counterparts. (2) These findings underscore *scSemiProfiler*'s sensitivity in detecting subtle differences between cell subtypes—a task that poses a challenge for other deconvolution methods that rely on cell type signature genes. We have expanded upon this discussion in the revised manuscript (please refer to the section "Semi-profiling with real bulk measurements yields a dataset nearly identical to the original single-cell data" in Results).

METHODS:

Comment 21: - Structure of the section must be improved to facilitate comprehension.

Response: Thank you for your suggestions concerning the readability and comprehensibility of the Methods section. We have restructured this section and enhanced it in the following ways: First, we have introduced an overview at the beginning of the Methods to briefly summarize the content and sequence of the entire section, ensuring consistency with our method overview plot (Fig. 1). Second, we have enriched the method sections with more detailed descriptions, as mentioned in our responses to the comments in our response letter. Third, we have organized the pipeline into distinct modules (phases, e.g., Initial Setup, Reconstruction, In-silico Inference, Active Learning), each marked with a descriptive function name (e.g., *IN SILICO INFERENCE*) to aid readers' understanding of each method detail. Finally, we added the *scSemiProfiler* algorithm pseudocode (Supplementary Fig. S51) to the "semi-profiling pipeline stop criteria" subsection, providing a clearer guide for readers to follow the overall pipeline.

Algorithm 1 Semi-profiling using scSemiProfiler

```
1: Input: Budget
2: Output: Semi-profiled single-cell data for the entire cohort
3: Main Routine:
4: Perform bulk sequencing of all samples.
5: Selected representatives  $\leftarrow$  INITIALREPRESENTATIVESELECTION(Bulk Data)
6: Conduct single-cell sequencing on selected representatives.
7: Pretrained models  $\leftarrow$  REPRESENTATIVERECONSTRUCTION(Selected single-cell data, Bulk Data)
8: Inferred single-cell data  $\leftarrow$  INSILICOINFERENCE(All bulk data, Selected single-cell data, Pretrained models)
9: Next representatives  $\leftarrow$  ACTIVELEARNING(Bulk data, Current semi-profiled single-cell cohort)
10: If stopping criteria are not met, repeat from "Conduct single-cell sequencing on selected representatives."
11: return Semi-profiled single-cell data for the entire cohort.

12: procedure INITIALREPRESENTATIVESELECTION(Bulk Data)
13:   Cluster samples based on bulk data.
14:   Select a representative sample from each cluster.
15:   return List of selected representatives.
16: end procedure

17: procedure REPRESENTATIVERECONSTRUCTION(Selected single-cell data, Bulk Data)
18:   Initialize Pretrained models
19:   For each selected representative:
20:     Train a VAE generator to reconstruct single-cell data (Pretrain 1.1).
21:     Jointly train generator and discriminator (Pretrain 1.2).
22:     Enhance generator training in full batch mode with added bulk data loss (Pretrain 2.1).
23:     Similar to Pretrain 2.1, but train discriminator jointly (Pretrain 2.2).
24:   return Pretrained models for selected representatives.
25: end procedure

26: procedure INSILICOINFERENCE(All bulk data, Selected single-cell data, Pretrained models)
27:   Initialize inferred single-cell data
28:   For each non-representative sample:
29:     Select the pretrained model of the closest selected representative.
30:     Fine-tune the model incorporating bulk sample differences.
31:     Conduct mini-stages 1-5 for tuning gene expression accuracy.
32:     Infer single-cell profile for the non-representative sample.
33:   return Inferred single-cell data for all non-representative samples.
34: end procedure

35: procedure ACTIVELEARNING(Bulk data, Current semi-profiled single-cell cohort)
36:   Analyze heterogeneity within clusters using current semi-profiled single-cell cohort and the bulk data
37:   Select next round of representatives to minimize heterogeneity.
38:   Update cluster membership.
39:   return List of next representatives.
40: end procedure
```

Supplementary Fig. S51: Pseudocode of scSemiProfiler 's pipeline.

Comment 22: - The initial setup of scSemiProfiler. Selection of the number of representatives using Kmeans to perform the clustering. How many PCs were used to do it?

Response: Thank you for highlighting the need to specify an important parameter in our data processing methodology. For all dimension reduction tasks in selection of number of representatives using K-Means, we employ a consistent approach to determine the number of principal components (PCs). Specifically, we use 100 PCs when there are more than 100 data samples to minimize information loss. For datasets with fewer than 100 samples, we use a

number of PCs equal to the sample size minus one. This clarification has been added to the “The initial setup of *scSemiProfiler*” section in the Methods of our revised manuscript.

Comment 23: - Representative single cell profiling and processing. Number of highly variable genes and method to select them must be specified as this is a critical starting point. Numbers for "high proportion of mitochondrial reads" are missing. Same for the method to find doublets.

Response: Thank you for reminding us of providing an important piece of information regarding preprocessing. In short, theoretically we believe that as long as enough highly variable genes are selected, the inclusion of the additional genes has a minor impact on the performance of the model as they have a minor impact on any training step, if any. In our experiments, we included more genes chosen by the default SCANPY's (Wolf, Angerer and Theis 2018) `pp.highly_variable_genes` tool to avoid missing important markers. We also reviewed the original study generating the datasets to find important markers and manually added them to the highly variable genes. The highly variable genes are chosen using the `pp.highly_variable_genes` tool in SCANPY. For the colorectal cancer dataset, due to its relatively sparser gene expression, we chose top 4,000 highly variable genes plus a few manually added markers discovered in the original colorectal cancer study (Joanito, Wirapati et al. 2022). For the rest, we chose top 6,000 highly variable genes plus a few markers. The user can decide the number of highly variable genes based on their own dataset and GPU memory or stick with our default value 6,000. The cutoff of the mitochondrial reads and doublets removal are determined using quality control plots generated by SCANPY's `pp.calculate_qc_metrics` tool. For the COVID-19 dataset and the iMGL dataset, these steps were performed by the dataset provider. The COVID-19 dataset performed doublet removal with Scrublet (Wolock, Lopez and Klein 2019) and used 10% as the mitochondrial reads cutoff. For the iMGL dataset, the dataset providers used thresholds of over 1,500 genes detected per cell, fewer than 100,000 reads per cell, and mitochondrial RNA fractions below 20%. For the colorectal cancer dataset, we removed doublets with more than 6000 genes expressed and applied the same mitochondrial reads cutoff 20%. The hamster dataset, characterized by its sparse single-cell data (averaging 1,151.4 cells per sample), bypasses cell, gene, and sample filtering to preserve data integrity. The above-mentioned details have been added to the “Representative single-cell profiling and processing” section of the Methods part of our revised manuscript.

Comment 24: - Fine-tune the deep generative learning model to infer the single cell. Are some weights or subnetworks frozen for pretrain2 with respect to pretrain1? And what about inference (fine-tuning for bulk data)?

Response: Thank you for pointing out a place lack of explanation. In all of these stages, none of the weights or subnetworks are frozen, except in the inference the discriminator network is no-longer involved as mentioned in the paragraph describing inference. This is because we want the model to perform reconstruction and consider the reconstruction loss all the time so that the model does not deviate too aggressively from the representative's data distribution. We have included this information in the revised method “Pretrain the deep generative learning model for reconstructing the single-cell data of the selected representatives” and “Fine-tune the

deep generative learning model to infer the single-cell measurements for the target samples” section.

Comment 25: - In cases such as the COVID-19 and colorectal cancer datasets, where real bulk data is not available, the entire second term simplifies to the pseudobulk of the target. It should not be like this, it should use the target pseudobulk as BT to work in a realistic way. Explain better or discuss

Response: Thanks for pointing out a method description needs to be better explained. We have improved this part in the revised manuscript. In brief, since pseudobulk (average gene expression) is an unbiased piece of information connecting the single-cell and bulk, we inject bulk information into model training by matching the pseudobulk of the generated cells and the target sample. So basically, the information that can be utilized by the model is pseudobulk instead of real bulk. When dealing with real bulk, we estimate the target pseudobulk from real bulk (and pseudobulk of the representative) using a robust and unbiased method (see more details in our answer to Reviewer #1 comment 6 and 7). However, when working with COVID-19 and colorectal cancer datasets, since we already have pseudobulk, we don't have to “infer” it again as the reviewer commented. Instead, we directly use the already existing pseudobulk data. We have updated the method text in section “Fine-tune the deep generative learning model to infer the single-cell measurements for the target samples” in the Methods part accordingly.

Comment 26: - Deep generative learning model architecture. More details are needed. For example: type of activation function for encoder and decoder (ReLU, I think), minibatch size in each training stage, etc. Reference to SCVI is missing.

Response: Thank you for your suggestions of including more relevant details regarding the model. In the revised manuscript, we have included these information and added another reference to SCVI (Lopez, Regier et al. 2018) in this paragraph. We use ReLU activation for our encoder and decoder and LeakyReLU in our discriminator. The minibatch size is 128 for all training except those we explicitly mentioned requiring full batch mode. The hidden layer size of the encoder and decoder is 256 and the latent variable size is 32. We have now added the aforementioned clarification in Method section “Deep generative learning model architecture”.

Comment 27: - Deep Generative Learning Model Training Details. Pretrain1 is described as 2-stage training, first only the AE (generator), and later generator+discriminator. Is this correct? In this case it should be Pretrain1.1, Pretrain1.2 or similar. Same for Pretrain2

Response: Thank you for your suggestions for improving the method description. Yes this is correct. Pretrain 1 and 2 each indeed contains two sub-stages and we now have split them. We have updated the corresponding description in the “Deep generative learning model training details” section in the Methods.

Comment 28: - The explanation of ministage training in fine-tuning for inference is confusing and incomplete. Each ministage corresponds to a threshold option? How are they ordered? Why is the last threshold not 1/16? What exactly does "target bulk loss in loss function is progressively quadrupled" mean, something to do with lambda_BulkT? In general, why minibatches are needed or recommended; what is the point of this strategy?

Response: We thank the reviewer for providing valuable feedback regarding our explanation of model training. Yes each ministage corresponds to a threshold. They are ordered from the largest to the lowest. Initially, there is no threshold, then the largest threshold is used, and the second largest threshold is used, and so on. We did not use 1/16 as our last threshold and used a slightly larger empirical value instead because 1/16 will make the model focus too much on tuning the few values very close to zero. "target bulk loss in loss function is progressively quadrupled" means that in each ministage, lambda_BulkT is four times larger than the previous stage. This is because using a smaller threshold will result in the decrease of the target bulk loss value. We therefore increase the value of lambda_BulkT to make this term (product of lambda_BulkT and target bulk loss) on the same scale as the previous stage. Regarding the necessity of these ministages, they are important for tuning relatively smaller gene expression values. When no threshold is applied, since our bulk loss has the form of square error, larger gradient values are passed to larger gene expression values. In other words, the training will focus too much on tuning the larger gene expression values and lack the attention for the smaller ones. Therefore, we apply this ministage strategy to progressively fine-tune the model, encouraging it to adjust the generated gene expressions from the larger ones to the lower ones. To demonstrate the efficacy of the ministage training strategy, we present the inference performance in Supplementary Fig. S50. This figure illustrates that after ministage training, the cells generated by the model exhibit increased similarity to the cells of the target sample, as compared to those generated prior to this training.

We have fleshed out these details to improve the intuitiveness and readability in the "Deep generative learning model training details" section of the Methods part in our revised manuscript.

Comment 29: - Incrementally select representatives using active learning to improve single-cell inference. Missing values (weights) for lambda_sc and lambda_pb.

Response: Thank you for catching this. The default values for both are 1, which is used for all experiments. This information has been added to the "Incrementally select representatives using active learning to improve single-cell inference" section of the Methods.

Comment 30: - Semi-profiling performance evaluation. "Referring to the comprehensive performance curves shown in (Fig.2d, Fig.4d, Fig.6d, the normalized error..." should be Fig.2f, Fig.4f, Fig.6f.

Response: Thank you for pointing out the mistake. We have corrected this in the revised version.

Reviewer #4 (Remarks to the Author):

This manuscript introduced scSemiProfiler, a novel computational framework that integrated deep generative model with active learning strategy to infer single-cell profiles across large cohorts. The authors carefully evaluated its performance and robustness using three distinct datasets, obtaining consistent results compared to real-profiled single-cell RNA sequencing data, validated by UMAP visualization and downstream analyses. Through these comprehensive evaluations, the authors demonstrated the method's capability to select representative samples for authentic single-cell sequencing, using an active learning module and successfully inferred the remaining samples' single-cell data using a deep generative learning model. Overall, this work is interesting and presents a potentially useful method in facilitating single-cell profiling. I do not have any major concerns, and the manuscript should undergo a minor but essential revision prior to acceptance.

Response: We are heartened by the reviewer's favorable evaluation of our work and their acknowledgment of its contribution to advancing single-cell profiling techniques. It is particularly reassuring to note the absence of any major concerns regarding our manuscript, which underscores the foundational strength of our research. We are thankful for the reviewer's minor yet pivotal suggestions, which have directed our attention to several aspects crucial for enhancing the user experience of our tool. In response, we have carefully addressed each comment in this revision, leading to substantial improvements in the tool's usability and benchmarking.

Minor comment 1:

1. In the overview of their method, the authors outlined the scSemiProfiler pipeline. It looks to me that multiple rounds of single-cell sequencing might be needed to obtain the final perfect results. However, such a procedure might bring a heavy burden to typical labs which budget is always limited, and in multiple rounds of sequencing, batch effects will occur and be difficult to clear. Also, the authors should describe how to experimentally select representative samples prioritized by active learning for sequencing. In my thinking, these representative cells have been used up for the first-round sequencing. Additionally, if only the deep generative module was used, the users would lose the superiority of active learning module. The authors should carefully address these concerns.

Response: We appreciate the reviewer's insightful comments regarding the selection of representatives. The rationale behind employing multiple rounds of single-cell sequencing is to identify the minimal number of profiled single-cell samples necessary for robust *in silico* inference across the entire cohort, thus avoiding over-profiling that could significantly increase the financial burden on the lab. However, we acknowledge the reviewer's concerns that multiple rounds of sequencing can pose challenges for many labs and may lead to batch effects. To accommodate users who prefer all-in-one-batch profiling, we have developed a new function called `scSemiProfiler.inspect_data`. This function allows users to assess data heterogeneity (using Silhouette scores and bulk data visualization) and determine an

approximate number of representative clusters needed for their dataset. Once the number of clusters is established, users can sequence all representatives in one batch to minimize potential batch effects.

Our strategy to experimentally select representative samples prioritized by active learning for sequencing is detailed as follows. Our approach to selecting representative samples for single-cell sequencing employs an iterative active learning process. Initially, an algorithm identifies a set of samples that provide a diverse and informative snapshot of the biological system under study, prioritizing them for the first round of sequencing. After sequencing, the gathered data is used to update and refine our semi-profiling model. If the accuracy of the resulting semi-profiles does not meet our standards (the inferred data is different from the sequenced data from the previous round), the active learning algorithm is re-engaged. It then selects samples as new representatives for subsequent rounds of sequencing based on their expected contribution to improving the model's accuracy. This iterative method ensures that the process does not exhaust the pool of informative samples in the initial round. Instead, it strategically selects a portion of samples at each step, preserving ample candidates for future sequencing. Our active learning design is meant to make efficient, incremental use of the sample pool, improving the model progressively with each round of sequencing. We also included the pseudocode of our method to the “Semi-profiling pipeline stop criteria” section of the Methods to better illustrate the pipeline (Supplementary Fig. S51).

Regarding the reviewer’s concern about the performance of not using active learning for the selection, we give the following explanation: One thing that we did not emphasize enough is that the main purpose of performing active learning is saving budget by allowing the user to stop selecting more representatives at any round and optimizing the performance at the same time. Selecting all representatives in the initial round based on clustering does not necessarily have a significantly worse performance than active learning, but rather losing the potential to minimize the number of samples chosen for single-cell sequencing. For example, based on the initial selection in round 1, we may choose 24 samples to profile, and find that 12 representative samples may deliver similar performance via active learning, which will cut down the sequencing cost substantially.

We have demonstrated the performance of this newly implemented “global” mode, the results were provided in Supplementary Figs S8 and S15. In this experiment using the COVID-19 dataset, we directly selected the same number of representatives as active learning (N=28) based on K-Means clustering of the bulk data and compared the results with the results generated using active learning. The analysis revealed that the “global mode” delivers a slightly worse performance than the “active learning mode”, but overall quite similar performance, but lacks the potential to cut down the representative samples selected for single-cell sequencing. When using *scSemiProfiler* in real-world scenarios, we recommend the users choose between these two modes (global vs. active learning) based on their own needs. If the main concern is to save the budget, then multiple-round selection using active learning is recommended. If the user aims to minimize the batch effects and maximize the convenience, then the global selection mode is more recommended.

Even in the scenario that a global mode is preferred, active learning is still essential to evaluate the quality of the *in silico* inference (semi-profiling). Unlike in the active learning model the *in silico* inference from the previous round of the target samples can be compared with the actual single-cell profiling in the next round to determine the semi-profiling accuracy, the accuracy of *in silico* inference in the global mode cannot be directly examined. To address this limitation, we can split the number of samples selected in global mode (e.g., N=28 will be split into 24 training + 4 test samples). We then only leverage partial samples (e.g., 24 samples) to train the model and *in silico* infer the single-cell data for the remaining samples (e.g., 4 samples). By comparing the *in silico* inferred single-cell data with the true single-cell measurement, we will be able to determine the semi-profiling accuracy using N samples. If not satisfactory, we can then leverage the active learning module to choose the next batch of representatives for better semi-profiling in the next round. In other words, a combination of the two modes can be adopted in real applications. This function has been added to our package published on the GitHub repository and a description of the stop criteria can be found in the “Semi-profiling pipeline stop criteria” section in the Methods of our manuscript

We have included the description of the global selection mode to the “Method overview” section of the Results part and “Global mode representative selection” of the Methods part. The evaluation of the global selection mode has been added to the sections for describing evaluating the model using the COVID-19 dataset (“The semi-profiled COVID-19 single-cell cohort exhibits significant similarity to its real counterpart” and “The semi-profiled COVID-19 single-cell cohort proves reliable for single-cell downstream analyses”) of the Results part respectively in the revised manuscript. We believe this could resolve the concerns mentioned in the reviewer’s comment.

Supplementary Fig. S8: Comparative analyses between real-profiled COVID-19 datasets and the semi-profiled dataset under global mode. **a**, UMAP visualization of the real-profiled COVID-19 cohort. Colors represent cell types and are consistent with **(f)**. **b**, UMAP visualization of the global mode semi-profiled COVID-19 cohort. **c**, Combined UMAP visualization comparing the cell distribution across both datasets. **d**, UMAP visualization of the global mode semi-profiled dataset with different colors representing the representatives' cells selected by the global selection and the in silico inferred cells. **e**, Comparison of the interferon pathway activation pattern in both datasets. **f**, Cell type composition visualization in different COVID-19 disease severity levels. The global mode semi-profiled dataset shows similar cell type proportions with the real-profiled version in each severity level. The Pearson correlations are: Healthy: 0.959, Asymptomatic: 0.968, Mild: 0.991, Moderate: 0.985, Severe: 0.996, Critical: 0.979. **g-i**, Deconvolution performance comparison between semi-profiling using active learning and semi-profiling using global mode.

Supplementary Fig. S15: Comparative analyses of single-cell level downstream analysis tasks using the real-profiled COVID-19 dataset and semi-profiled COVID-19 dataset generated using the global mode. a, Dot plots visualizing the top cell type signature genes' expression in real-profiled data (top) and global mode semi-profiled data (bottom). **b**, RRHO plot comparing the CD4 positive and negative markers in both datasets. **c**, CD4 cell type signature genes GO enrichment analysis results comparison. **d**, Cell-cell interaction analysis results comparison between the real-profiled dataset and the global mode semi-profiled dataset. **e**, Pseudotime results comparison using the real-profiled and global mode semi-profiled CD4 cells.

Minor comment 2. Although the authors successfully validated the performance of scSemiProfiler on three distinct cohorts, showing a considerable consistency between semi-profiled and real-profiled data, I suggest that the authors must perform additional comparisons using more cohorts of different species or tissues to enhance the reliability and transparency of the algorithm.

Response: Thank you for your suggestions for enhancing the reliability and transparency of our method. We have performed another dataset with paired real bulk and single-cell data from

hamsters' lung (Nouailles, Adler et al. 2023). The success of *scSemiProfiler* on this dataset, especially a more significant lead in deconvolution performance over other method further demonstrates the reliability of our tool. See more details regarding this dataset in our response to the next question (Reviewer #4 minor comment 3).

Minor comment 3. From the results, it can be found that the performance of *scSemiProfiler* is not markedly higher than Bisque on the data set of iMGL, compared to other two data sets. Thus, I am curious that whether the results might be attributed to the process of pseudo-bulk data generation. The authors should describe that whether the method of averaging normalized count single-cell data to generate bulk data is a gold-standard method. Furthermore, the authors should clarify that whether only the characteristics of pseudo-bulk data generation from single-cell data has been learnt, in the training step using the COVID-19 and Colorectal Cancer datasets learns. The authors should include more cohorts with real bulk sequencing data and paired single-cell sequencing data for further validation. Considering the data limitation, the authors can also choose to analyze the pseudo-bulk data of iMGL, and then compare it with the real bulk data.

Response: Thank you for your insightful comments. Yes, we acknowledge that the performance of Bisque on the iMGL dataset (with true bulk data) is relatively smaller to *scSemiProfiler* (although *scSemiProfiler*'s performance is still visually better). However, *scSemiProfiler* has significantly better performance in the COVID-19 dataset (CCC p-value 3.95×10^{-16} , Pearson correlation p-value 3.40×10^{-16} , RMSE p-value 1.31×10^{-16}), colorectal cancer dataset (CCC p-value 4.02×10^{-14} , Pearson correlation p-value 4.18×10^{-12} , RMSE p-value 2.81×10^{-13}), and another real bulk hamster dataset(CCC p-value 4.91×10^{-4} , Pearson correlation p-value 4.03×10^{-4} , RMSE p-value 4.91×10^{-4}). compared to the other two datasets. We agreed with the reviewer this may be attributed to the process of pseudo-bulk data generation strategy that was employed. To answer this question, we compared the pseudobulk that we inferred for the target samples based on the representative with the real pseudobulk for the target samples that were actually profiled. We found that the inferred pseudobulk for the target sample better resembles the true pseudobulk of the target sample, compared with the bulk sequencing due to the existence of systematic differences between bulk and single-cell sequencing data. We invite the reviewer to see our response to Reviewer 1 comment 6 for a detailed discussion and comparison of our pseudobulk generation strategy for the target sample. We also want to point out that a similar performance between Bisque and *scSemiProfiler* on real dataset is not a common phenomenon. Besides the iMGL dataset, we also applied the methods to another single-cell dataset from Hamster with true bulk RNA-seq. In this dataset, *scSemiProfiler* shows a much higher performance (Supplementary Fig. S43). This indicates that *scSemiProfiler*, with our pseudobulk generation strategy, delivers superior performance compared to other benchmarked methods.

Here, it is important to clarify that our method does not directly use bulk RNA-seq data to guide the deconvolution of target samples. This approach circumvents the challenge of modeling systemic differences and batch effects typically encountered between single-cell and bulk sequencing, even they are performed on the same sample. Our method entails

inferring the pseudo-bulk data for a target sample by converting the observed pseudo-bulk from our profiled single-cell data of the representative. The bulk sequencing of both the representative and target samples is used solely to calculate a conversion ratio between them, rather than directly supervising the deconvolution process. For instance, if the average expression of a gene (pseudo-bulk) in the profiled single-cell data is 5, and we wish to estimate its average expression in the target sample's single-cell data (marked as x), we utilize bulk profiling to determine this. If bulk sequencing reveals that the representative's expression is 2 and the target's true bulk expression is 4, we do not estimate x directly from this bulk expression. Instead, we use the bulk expression data to calculate a conversion ratio (in this case, 2), which we then apply to the representative's pseudo-bulk data to estimate the target's pseudo-bulk (resulting in $5 \times 2 = 10$). As illustrated in Supplementary Fig. S36, our pseudo-bulk inference strategy produces results that are statistically closer to the true pseudo-bulk of the target samples compared to the expression from bulk sequencing. This substantiates the robustness and efficacy of our approach for pseudo-bulk data generation. In addition, generating pseudobulk expression by aggregating the expression from all cells in the sample (i.e., averaging or summing) is also common practice adopted in many existing studies (Menden, Marouf et al. 2020, Chen, Wang et al. 2022).

The reviewer is correct about the fact that for COVID-19 and colorectal cancer datasets, the model only learns to align single-cell data and pseudobulk data, which means the model deconvolute pseudobulk data into single-cell data, and it generates data with pseudobulk that is similar to the target sample's pseudobulk data. Nevertheless, by accurately converting real bulk data to pseudobulk data space, we managed to let the model maintain its high performance when dealing with pseudobulk data.

In response to the reviewer's concern about the tool's reliability in dealing with real bulk data, we performed both experiments suggested by the reviewer. Firstly, we searched the GEO database and checked all datasets with paired bulk and single-cell data and found one hamster lung dataset (another species and another tissue compared to the datasets we included in the first version) can be used for evaluating our method, although with relatively lower data quality (much lower number of cells in each sample and much fewer samples compared to previous datasets). We performed the same analysis for this dataset and presented the results in Supplementary Figs. S43 and S48. The similarity between the semi-profiled dataset and the real-profiled dataset, and *scSemiProfiler*'s markedly lead over all other deconvolution methods (Supplementary Fig. S43f, h, i) further justifies *scSemiProfiler*'s capability of dealing with datasets with real bulk data, from other species, and other tissue. By only selecting 2 out of 16 samples as representatives (Supplementary Fig. S43d indicates most cells are generated), *scSemiProfiler* successfully semi-profiled a single-cell dataset that is very similar to the real-profiled ground truth. Other outputs from *scSemiProfiler* such as UMAP visualization (Supplementary Fig. S43a, b, c), cell type proportion estimating (Supplementary Fig. S43f, g, h, i) are also very accurate. The downstream analysis shown in Supplementary Fig. S48 demonstrates that for this hamster dataset, the semi-profiled dataset generated by *scSemiProfiler* is still capable of generating reliable analysis results for biomarker discovery, cell-cell interaction, and pseudotime.

Since the cohort studies with both single-cell and bulk sequencing on the same samples are very limited, after extensive searching, we only found the hamster example that we discussed

above. However, the data quality of this dataset is relatively low. The single-cell data in this hamster dataset has significantly fewer cells than the other three datasets. The average number of cells in each sample in the hamster dataset is 1151.4, while in the colorectal cancer dataset is 1914.0, in the COVID-19 dataset is 5138.3, and in the iMGL dataset is 2974.9. Most samples in the hamster dataset could not pass our previous filtering criteria – samples with fewer than 1,000 cells should be filtered. Low number of cells significantly decreases the number of training samples for the deep generative learning model, making it hard for the model to learn the underlying data distribution. Therefore, we followed the reviewer’s suggestion and performed a direct performance comparison of semi-profiling on the iMGL dataset using pseudobulk bulk and real bulk expression data. The results were detailed in Supplementary Figs. S42 and S47. In all the analysis, the pseudobulk version is very similar to the real bulk version, demonstrating that scSemiProfiler’s performance is not affected by using real bulk data. In the semi-profiling results performed using the pseudobulk iMGL dataset, the Supplementary Figs. S42 a, b, c still shows the generated dataset has very similar UMAP visualization as the real-profiled data and Supplementary Figs. S42d shows that most cells are generated by the model. Supplementary Figs. S42e shows the gene set activation pattern between the semi-profiled version and real-profiled version are very similar and this similarity quantified by Pearson correlation is similar to the real bulk version (0.980 vs 0.952). Cell type proportion in each experiment condition shown in Supplementary Figs. S47f is also nearly identical to the real-profiled version. Supplementary Figs. S42g, h, i show the deconvolution performance comparison between scSemiProfiler using real bulk and pseudobulk, which is very similar. The downstream analysis shown in Supplementary Figs. S47a, b, c, d, e are all very similar to the real bulk version shown in Figs. 7. The performance similarity in semi-profiling with real bulk and pseudobulk is also evidence of the effectiveness of our pseudobulk generation strategy. The discussion regarding the hamster dataset has also been added to the “Further validating scSemiProfiler's performance using real bulk data from other species and tissues” section of the Results part of our revised manuscript.

Supplementary Fig. S43: Comparing real-profiled and semi-profiled hamster dataset. **a**, UMAP visualization of cells in the real-profiled hamster cohort, with colors representing cell types. Colors are consistent with (g). **b**, UMAP visualization of the semi-profiled hamster dataset. **c**, Jointly visualize real-profiled and semi-profiled hamster cohorts, showing a high similarity between them. **d**, UMAP of the semi-profiled cohort, with colors representing cells from the 2 representative samples or from the rest 14 inferred non-representative samples. **e**, Error trajectory of scSemiProfiler as more representatives are selected. **f**, **h**, **i**, Deconvolution performance benchmarking using RMSE, CCC, and Pearson correlation respectively. All results show a significant lead of our method over other deconvolution methods. **g**, Stacked bar plots visualizing the cell type proportions in different experiment treatments in the real-profiled and semi-profiled cohorts. The Pearson correlations between the real-profiled and semi-profiled cohorts under different treatments are: aaUntr: 0.934, adeno2x: 0.963, att2x: 0.898, mRNA2x: 0.993, mRNAatt: 0.873.

Supplementary Fig. S48: Comparing the downstream analysis results using the semi-profiled and real-profiled hamster cohorts. a, Dot plots showing similar cell type signature genes expression patterns in the semi-profiled and real-profiled cohorts. **b**, RRHO plot comparing the Trem4+Macrophages markers found using the two versions of datasets. **c**, Trem4+Macrophages cell type signature genes GO enrichment analysis results comparison. **d**, PAGA analysis results generated from the two versions of hamster datasets show high similarity. **e**, Pseudotime results comparison using the real-profiled and in silico generated Trem4+Macrophages

Supplementary Fig. S42: Comparing semi-profiled iMGL dataset generated based on pseudobulk data and real-profiled ground truth. Please note this pseudobulk is not inferred from the real bulk. Instead, it is a direct average of the single-cell data. **a**, Real-profiled iMGL data colored by cell type. Colors are consistent with **(b)** and **(f)**. **b**, Semi-profiled iMGL data based on pseudobulk colored by cell type. **c**, Combined UMAP comparing real-profiled and pseudobulk-based semi-profiled IMGL dataset. **d**, Pseudobulk-based semi-profiled IMGL dataset colored by cell origins (from representative samples or generated by the deep generative model). **e**, “activation of immune response” GO term activation pattern comparison. **f**, Comparing cell type proportions under different experimental conditions between datasets. The Pearson correlation between the real-profiled and pseudobulk-based semi-profiled versions of cell type proportions under different conditions are: iMGL_D0: 0.998, iMGL_D1: 0.872, iMGL_D2: 0.997, iMGL_D3: 0.998, iMGL_D4: 0.995, iMGL_DMSO: 0.998, iMGL_GW30: 0.983, iMGL_GW_300: 0.999, iMGL_T_30: 0.993, iMGL_T_300: 0.999. **g,h,i**, Deconvolution performance comparison between pseudobulk-based and real bulk-based semi-profiling using CCC, RMSE, Pearson correlation respectively.

Supplementary Fig. S47: iMGL real-profiled cohort and pseudobulk-based semi-profiled cohort downstream single-cell analysis results comparison. Please note this pseudobulk is not inferred from the real bulk. Instead, it is a direct average of the single-cell data. **a**, Dot plots visualizing the cell type signature genes. **b**, RRHO plots comparing the cell type biomarkers in two datasets. **c**, C3 Cell type markers GO enrichment analysis results comparison. **d**, PAGA results generated using two versions of datasets comparison. **e**, Pseudotime analysis results comparison.

Minor comment 4: Although the authors compared the performance of scSemiProfiler with CIBERSORTx, Bisque, TAPE, and Scaden, it would be valuable to include more tools for comparisons on deconvolution of bulk data to single-cell data analysis, such as MuSiC, EPIC, and so on. Additionally, analyzing the profiles generated by other tools would highlight the necessity and superiority of scSemiProfiler.

Response: Thank you for your suggestions on improving the benchmarking of the deconvolution task. In the revised version, we have added comparison to three more methods, MuSiC, EPIC, and NNLS. Among the three newly added methods, MuSiC shows the best overall performance. Nevertheless, the new benchmarking results still show that *scSemiProfiler* has a significant overall lead in terms of deconvolution performance (evaluated by CCC, Pearson correlation and RMSE) on all the four datasets we tested (COVID-19 Fig. 2 j, k, l, colorectal cancer Fig. 4 j, k, l, iMGL Fig. 6 j, k, l, hamster Supplementary Fig. S43 f, h, i).

Although MuSiC shows remarkable results on the colorectal cancer dataset, *scSemiProfiler* has overwhelmingly better results on all other three datasets, especially the two real bulk datasets – iMGL (Fig. 6j, k, l a huge gap between our method and these three) and hamster (Supplementary Fig. S43f, h, i, *scSemiProfiler* has around 0.9 average CCC and Pearson correlation while those three methods all have negative CCC and Pearson correlation). It has also been reported in previous studies that MuSiC fails to generate meaningful results on real bulk datasets (negative CCC reported by (Menden, Marouf et al. 2020, Chen, Wang et al. 2022)).

We believe the updated benchmarking results, as well as the fact that *scSemiProfiler* can generate single-cell level outputs that other deconvolution methods cannot generate, further demonstrates the necessity and superiority of *scSemiProfiler*.

Minor Comment 5. Existing tools such as CIBERSORTx offer web and software versions for accessibility, making it easier to use for scientists. The authors only provide a Python package for applications. Is there any possibility that the authors can develop a user-friendly web or software interface to facilitate broader use for biologists?

Response: Thank you for pointing out a good future direction for your method. Unfortunately, unlike CIBERSORTx, our method involves training deep learning models and requires far more extensive computational resources. The lab-server resource in our lab may be limited to support such a service in large-scale at this stage. However, we made our best efforts to provide detailed documentations and tutorials on our GitHub repository. We also provided an example Jupyter notebook that can be applied to any dataset with minor modifications. We also encapsulated all functional modules to facilitate its application and adoption in other studies. We believe these make it convenient for researchers, even those with no extensive coding background, to use our tool. Should users have any potential questions regarding using the tool, we are committed to provide support via the GitHub repository of the program.

Minor Comment 6. Although annotations were provided for each color in Figures 2g, 4g, and 6g, the authors should provide full annotations of their clustering results in Figures 2a, 4a, and 6a.

Response: Thank you for pointing out the issue regarding the legend. We actually used the same legend for the UMAPs (Figs. 2a, 2b, 4a, 4b, 6a, 6b) and the stacked bar plots (Figs. 2g, 4g, 6g in the previous version, which are Figs. 2i, 4i, 6i in the revised version) and mentioned this in the figure caption. In response to your comment, in the revised version, we further emphasized this in the figure captions and text to make it more obvious to the readers.

Taken together, this study provided an interesting method, which might be useful for facilitating single-cell profiling and data analysis.

Reviewer #4 (Remarks on code availability):

Code availability minor comment 1: If possible, an online service should be implemented for a convenient use of their method.

Response: Answered in our response to Reviewer #4 minor comment 5.

We believe that these revisions have significantly improved the manuscript, making it a valuable contribution to single-cell genomics. We appreciate the opportunity to revise our work and thank the reviewers for their constructive feedback.

Thank you for considering our revised manuscript. We look forward to the possibility of our work being published in *Nature Communications*.

Sincerely,

Jun Ding, PhD

Assistant professor
Department of Medicine | School of Computer Science, McGill University
MILA-Quebec AI Institute

References

Chen, Y., et al. (2022). "Deep autoencoder for interpretable tissue-adaptive deconvolution and cell-type-specific gene analysis." *Nature Communications* **13**(1): 6735.

Franzén, O., et al. (2019). "PanglaoDB: a web server for exploration of mouse and human single-cell RNA sequencing data." *Database* **2019**: baz046.

Galbraith, M. D., et al. (2022). "Specialized interferon action in COVID-19." *Proceedings of the National Academy of Sciences* **119**(11): e2116730119.

Gruber, C. (2020). "Impaired interferon signature in severe COVID-19." *Nature Reviews Immunology* **20**(6): 353-353.

Hao, Y., et al. (2021). "Integrated analysis of multimodal single-cell data." *Cell* **184**(13): 3573-3587. e3529.

Hasanaj, E., et al. (2022). "Interactive single-cell data analysis using Cellar." *Nature Communications* **13**(1): 1998.

Hicks, S. C., et al. (2018). "Missing data and technical variability in single-cell RNA-sequencing experiments." *Biostatistics* **19**(4): 562-578.

- Hou, W., et al. (2020). "A systematic evaluation of single-cell RNA-sequencing imputation methods." Genome biology **21**: 1-30.
- Hu, C., et al. (2023). "CellMarker 2.0: an updated database of manually curated cell markers in human/mouse and web tools based on scRNA-seq data." Nucleic Acids Research **51**(D1): D870-D876.
- Joanito, I., et al. (2022). "Single-cell and bulk transcriptome sequencing identifies two epithelial tumor cell states and refines the consensus molecular classification of colorectal cancer." Nature genetics **54**(7): 963-975.
- Kharchenko, P. V., et al. (2014). "Bayesian approach to single-cell differential expression analysis." Nature methods **11**(7): 740-742.
- Krogh, A. and J. Hertz (1991). "A simple weight decay can improve generalization." Advances in neural information processing systems **4**.
- Lawrence, I. and K. Lin (1989). "A concordance correlation coefficient to evaluate reproducibility." Biometrics: 255-268.
- Lee, J. S. and E.-C. Shin (2020). "The type I interferon response in COVID-19: implications for treatment." Nature Reviews Immunology **20**(10): 585-586.
- Li, D., et al. (2022). "Unsupervised cell functional annotation for single-cell RNA-seq." Genome Research **32**(9): 1765-1775.
- Lopez, R., et al. (2018). "Deep generative modeling for single-cell transcriptomics." Nature methods **15**(12): 1053-1058.
- Loshchilov, I. and F. Hutter (2017). "Decoupled weight decay regularization." arXiv preprint arXiv:1711.05101.
- Maden, S. K., et al. (2023). "Challenges and opportunities to computationally deconvolve heterogeneous tissue with varying cell sizes using single-cell RNA-sequencing datasets." Genome biology **24**(1): 288.
- Menden, K., et al. (2020). "Deep learning-based cell composition analysis from tissue expression profiles." Science advances **6**(30): eaba2619.
- Momeni, K., et al. (2023). "Unraveling the complexity: understanding the deconvolutions of RNA-seq data." Translational Medicine Communications **8**(1): 21.
- Nouailles, G., et al. (2023). "Live-attenuated vaccine sCPD9 elicits superior mucosal and systemic immunity to SARS-CoV-2 variants in hamsters." Nature Microbiology **8**(5): 860-874.
- Steen, C. B., et al. (2020). "Profiling cell type abundance and expression in bulk tissues with CIBERSORTx." Stem Cell Transcriptional Networks: Methods and Protocols: 135-157.
- Wolf, F. A., et al. (2018). "SCANPY: large-scale single-cell gene expression data analysis." Genome biology **19**: 1-5.
- Wolock, S. L., et al. (2019). "Scrublet: computational identification of cell doublets in single-cell transcriptomic data." Cell systems **8**(4): 281-291. e289.

Zeisel, A., et al. (2015). "Cell types in the mouse cortex and hippocampus revealed by single-cell RNA-seq." Science **347**(6226): 1138-1142.

Zhang, X., et al. (2019). "CellMarker: a manually curated resource of cell markers in human and mouse." Nucleic Acids Research **47**(D1): D721-D728.

Zhang, Y., et al. (2021). "The ORF8 protein of SARS-CoV-2 mediates immune evasion through down-regulating MHC-I." Proceedings of the National Academy of Sciences **118**(23): e2024202118.

Reviewer #1 (Remarks to the Author):

The authors have made an exceptional effort to address all reviewers' concerns and comments. I find their revision to be meticulously considered and expertly presented. I have no further comments.

Reviewer #2 (Remarks to the Author):

I would like to congratulate the authors for all the work done to review the manuscript. All my concerns have been addressed.

Reviewer #3 (Remarks to the Author):

The authors have successfully addressed all my questions and concerns in a rigorous and comprehensive manner.

Reviewer #4 (Remarks to the Author):

I carefully read the revised manuscript and their response letter. The authors did a lot of jobs to address all my concerns. I agree that the manuscript can be accepted in current form.

Reviewer #4 (Remarks on code availability):

The code is OK.